# Near-Optimal Second-Order Guarantees for Model-Based Adversarial Imitation Learning

**Shangzhe Li**
UNC Chapel Hill
Chapel Hill, NC, USA
shangzhe@unc.edu

**Dongruo Zhou**
Indiana University Bloomington
Bloomington, IN, USA
dz13@iu.edu

**Weitong Zhang**
UNC Chapel Hill
Chapel Hill, NC, USA
weitongz@unc.edu

## Abstract

We study online adversarial imitation learning (AIL), where an agent learns from offline expert demonstrations and interacts with the environment online without access to rewards. Despite strong empirical results, the benefits of online interaction and the impact of stochasticity remain poorly understood. We address these gaps by introducing a model-based AIL algorithm (`MB-AIL`) and establish its horizon-free, second-order sample-complexity guarantees under general function approximations for both expert data and reward-free interactions. These second-order bounds provide an instance-dependent result that can scale with the variance of returns under the relevant policies and therefore tighten as the system approaches determinism. Together with second-order, information-theoretic lower bounds on a newly constructed hard-instance family, we show that `MB-AIL` attains minimax-optimal sample complexity for online interaction (up to logarithmic factors) with limited expert demonstrations and matches the lower bound for expert demonstrations in terms of the dependence on horizon $H$, precision $\epsilon$ and the policy variance $\sigma^2$. Experiments further validate our theoretical findings and demonstrate that a practical implementation of `MB-AIL` matches or surpasses the sample efficiency of existing methods.

## 1 Introduction

Imitation learning (IL) focuses on learning a policy that mimics expert behavior for sequential decision-making using demonstrations. Unlike reinforcement learning (RL), IL learns from expert trajectories without the reward signals. This advantage has drawn substantial attention to IL, with demonstrated successes across diverse real-world applications, including robot learning (Chi et al., 2024; Shi et al., 2023) and autonomous driving (Couto & Antonelo, 2021; Cheng et al., 2024) where the reward signal is hard to design or implicit to learn.

Among the many variants of imitation learning, methods largely fall into two families: behavioral cloning (BC) and adversarial imitation learning (AIL). In particular, BC applies supervised learning to fit the expert policy directly from demonstrations (Florence et al., 2022; Chi et al., 2024), while AIL employs an adversarial framework to align the state-action distributions of expert and learner policies (Ho & Ermon, 2016; Garg et al., 2021). Importantly, AIL can also leverage online, reward-free interactions during training. This online interaction will allow the agent to collect the environment dynamics information in order to better align with the behavioral policy. To explain empirical findings that AIL often outperforms BC in low expert demonstration regimes (Ho & Ermon, 2016; Garg et al., 2021), Xu et al. (2022) analyzed this phenomenon under a specific MDP structure, relating it to the expert sample complexity results of Rajaraman et al. (2020). Naturally, we have the following question:

*What is the tight characterization of the benefits of online interaction in imitation learning?*

In order to obtain a tight characterization, two aspects need to be addressed. First, a line of research has extended the *model-based framework* in RL (Sun et al., 2019a; Wang et al., 2024) to AIL, both theoretically (Xu et al., 2023) and empirically (Baram et al., 2016; Kolev et al., 2024; Yin et al., 2022; Li et al., 2025), partially demonstrating that model-based approaches can yield superior sample efficiency. However, existing theoretical analyses for model-based IL have not yet clarified why such

approaches are beneficial to AIL under the general setting. Second, it has been observed that the *stochasticity* of the experts plays an important role in achieving tight sample complexity. Existing works analyze BC and AIL under deterministic versus stochastic experts and have established a gap: for deterministic experts, Rajaraman et al. (2020); Foster et al. (2024) obtain an $\widetilde{\mathcal{O}}(\epsilon^{-1})$ expert sample complexity, whereas for stochastic experts, existing results typically yield $\widetilde{\mathcal{O}}(\epsilon^{-2})$ (Foster et al., 2024; Xu et al., 2024). Nevertheless, a complete understanding of how stochasticity affects the sample complexity of AIL remains open.

In this paper, we answer this question affirmatively by presenting a model-based AIL algorithm (`MB-AIL`) with a corresponding horizon-free, second-order analysis in general function approximation to understand the benefits of online interaction and the impact of stochasticity to the sample complexity to both expert demonstration and online interaction. A second-order bound scales with the variance of the total return and can be small as the system approaches determinism, providing a powerful tool for us to demonstrate the effect of stochasticity. Combining with a lower bound on the minimax sample complexity for both expert and online interaction, we understand the benefit and barrier of the online interaction statically. In particular, our contributions can be listed as follows:

- We present a model-based adversarial imitation learning algorithm (`MB-AIL`) inspired by the fact that the policy class $\Pi$ can be split into the reward class $\mathcal{R}$ and the model class $\mathcal{P}$. `MB-AIL` decomposes this reward learning and model learning procedure by estimating the reward adversarially using expert demonstration using a no-regret algorithm and learn the model using a simple MLE estimator for the model and an optimistic exploration strategy for online interaction.

- We present a second-order, horizon-free bound for `MB-AIL` under general function approximation on sample complexity for both of the expert demonstration and online interactions. This result reveals the benefit of the online interaction and the impact of the variance to the sample complexity. At the core of our analysis is to decompose the regret into the error of reward and the error of the model estimation. Then two fine-grained variance-depend analysis is conducted in both of the part for a horizon-free second-order regret analysis in model-based RL.

- We present a information-theoretic lower bound for both of the expert demonstration and online interaction on a newly designed hard instance for imitation learning. The lower bound reveals that the expert demonstration can help with estimating the reward while the online interaction is tied with learning the transition kernel. This lower bound also suggests that the `MB-AIL` is minimax-optimal in terms of the online interaction up to log factors when given limited expert demonstrations. While `MB-AIL` leaves only a $\log |\mathcal{R}|$ gap in terms of the expert demonstrations.

Besides the theoretical advancements in understanding the online interaction in imitation learning, we also present a practical implementation of the proposed `MB-AIL` algorithm to validate our theoretical findings. `MB-AIL` is also reported with better sample complexity in a various of environment.

**Notation.** In this paper, we use plain letters such as $x$ to denote scalars, lowercase bold letters such as $\mathbf{x}$ to denote vectors, and uppercase bold letters such as $\mathbf{X}$ to denote matrices. Functions are denoted by bold symbols such as $\boldsymbol{f}$. Sets and classes are denoted by the calligraphic font such as $\mathcal{F}$. For a vector $\mathbf{x}$, $\|\mathbf{x}\|_2$ denotes its norm $\ell_2$. For a positive integer $N$, we use $[N]$ to denote $\{1, 2, \ldots, N\}$. We use standard asymptotic notations, including $O(\cdot), \Omega(\cdot), \Theta(\cdot)$, and $\widetilde{O}(\cdot), \widetilde{\Omega}(\cdot), \widetilde{\Theta}(\cdot)$ which will hide logarithmic factors. For two distribution $p$ and $q$ ,we define the Hellinger distance as $\mathbb{H}^2(p, q) = \int \left(\sqrt{\mathrm{d}p/\mathrm{d}w} - \sqrt{\mathrm{d}q/\mathrm{d}w}\right)^2 \mathrm{d}w$. We denote the $d$-dimensional binary set by $\mathbb{B}^d = \{\pm 1\}^d$ that is, the set of all $d$-dimensional vectors whose entries are either $+1$ or $-1$.

## 2 RELATED WORKS

Below we provide a comprehensive review of the related work on theoretical understanding of imitation learning. We defer a more detailed discussion on the empirical implementation on imitation learning, theoretical background of reinforcement learning into Appendix A.

**Theoretical Understanding in Imitation Learning.** Imitation learning can be interpreted as a variant of RL in which the agent learns from expert demonstrations instead of reward signals. Early works (Abbeel & Ng, 2004; Syed & Schapire, 2007; Sun et al., 2019b; Rajaraman et al., 2020; Chen et al., 2020) assumes either known transition dynamics or access to an exploratory data distribution. More recent studies consider the unknown transitions and provide theoretical guarantees on the

Table 1: **Summary of Existing Results.** We summarize existing upper and lower bound results on imitation learning and compare them with our results. *Stoch. Expt?* indicates that if the presented analysis can deal with the stochastic expert or is for deterministic expert only. *Demonstration* indicates the number of expert demonstration trajectories required for achieving $\epsilon$-optimal policy. *Interaction* indicates the rounds of online interaction which is not required for the algorithms marked with 'Offline BC'. Algorithms that may require an online interaction without specifying a bound is left with '–'. The sample complexity is evaluated under the bounded cumulative reward $\sum_{h=1}^{H} r_h \leq 1$ and the results with bounded reward assumption $r_h \leq 1$ are translated by letting $\epsilon \leftarrow H\epsilon$ in the original results and marked with (*). Dimension $d_E$ and model class $\log|\Pi|$, $\log|\mathcal{R}|$, $\log|\mathcal{Q}|$, $\log|\mathcal{P}|$ in general function approximations inherits their original definitions in the paper, and it usually corresponds to the dimension of linear function when reduced to linear function approximations. Variance $\sigma^2$ refers to the variance of the collected return, as defined in detail in the paper, and is always bounded by $H^2$. In contrast, $\sigma_E^2$ denotes the variance of the collected return under the expert policy. Our results are highlights in boldface and in cyan background.

| Setting | Algorithm | Stoch. Expt? | Demonstration | Interaction |
|---------|-----------|:---:|:---:|:---:|
| Tabular MDPs | Rajaraman et al. (2020) | $\times$ | $\widetilde{\mathcal{O}}(H\|\mathcal{S}\|\epsilon^{-1})^*$ | Offline BC |
| | TV-AIL (Xu et al., 2022) * | $\checkmark$ | $\widetilde{\mathcal{O}}(\|\mathcal{S}\|\epsilon^{-2})^*$ | – |
| | MB-TAIL (Xu et al., 2023) | $\times$ | $\widetilde{\mathcal{O}}(H^{1/2}\|\mathcal{S}\|\epsilon^{-1})^*$ | $\widetilde{\mathcal{O}}(H\|\mathcal{S}\|^2\|\mathcal{A}\|\epsilon^{-2})^*$ |
| | Lower Bound (Rajaraman et al., 2020) | $\checkmark$ | $\widetilde{\Omega}(H\|\mathcal{S}\|\epsilon^{-1})^*$ | Offline BC |
| | Lower Bound (Xu et al., 2022) † | $\checkmark$ | $\widetilde{\Omega}(\|\mathcal{S}\|\epsilon^{-2})^*$ | – |
| Linear MDPs Linear Mix. MDPs | BRIG (Viano et al., 2024) | $\checkmark$ | $\widetilde{\mathcal{O}}(d\epsilon^{-2})^*$ | $\widetilde{\mathcal{O}}(H^2 d^3 \epsilon^{-2})^*$ |
| | OGAIL (Liu et al., 2021) | $\checkmark$ | $\widetilde{\mathcal{O}}(Hd^2\epsilon^{-2})^*$ | $\widetilde{\mathcal{O}}(H^2 d^3 \epsilon^{-2})^*$ |
| General | Foster et al. (2024) | $\checkmark$ | $\widetilde{\mathcal{O}}(\sigma_E^2 \log|\Pi|\epsilon^{-2})$ | Offline BC |
| | Foster et al. (2024) | $\times$ | $\widetilde{\mathcal{O}}(\log|\Pi|\epsilon^{-1})$ | Offline BC |
| | OPT-AIL (Xu et al., 2024) | $\checkmark$ | $\widetilde{\mathcal{O}}(\log|\mathcal{R}|\epsilon^{-2})^*$ | $\widetilde{\mathcal{O}}(H^2 d_E \log(|\mathcal{R}||\mathcal{Q}|)\epsilon^{-2})^*$ |
| Function | **MB-AIL (Ours)** | $\checkmark$ | $\widetilde{\mathcal{O}}(\sigma_E^2 \log(|\mathcal{R}|)\epsilon^{-2})$ | $\widetilde{\mathcal{O}}(\sigma^2 d_E \log(|\mathcal{P}|)\epsilon^{-2})$ |
| | Lower Bound (Foster et al., 2024) | $\times$ | $\widetilde{\Omega}(\epsilon^{-1})^*$ | Offline BC |
| Approximation | Lower Bound (Foster et al., 2024) | $\checkmark$ | $\widetilde{\Omega}(\sigma_E^2 \epsilon^{-2})$ | Offline BC |
| | **Lower Bound (Ours)** | $\checkmark$ | $\widetilde{\Omega}(\sigma_E^2 \epsilon^{-2})$ | $\widetilde{\Omega}(\sigma^2 (\log|\mathcal{P}|)^2 \exp(-N)\epsilon^{-2})^{\ddagger}$ |

sample complexity of expert data under various structural assumptions. For example, Shani et al. (2022) provide an $\mathcal{O}(\sqrt{K})$ regret upper bound for on-policy AIL, while Chen et al. (2024) provide a convergence guarantee for off-policy AIL in tabular MDPs. Moulin et al. (2025b) investigates the state-only imitation learning setting under discounted linear MDPs and provides an expert sampling complexity upper bound $\widetilde{\mathcal{O}}(W_{\max}^2 (1 - \gamma)^{-2}\epsilon^{-2})$ and an interaction complexity upper bound $\widetilde{\mathcal{O}}(d^3(1 - \gamma)^{-4.5}\epsilon^{-2})$. Xu et al. (2023) study the tabular MDP setting with a deterministic expert policy, obtaining an expert sample complexity of $\widetilde{\mathcal{O}}(H^{3/2}|\mathcal{S}|/\epsilon)$ and interaction complexity of $\widetilde{\mathcal{O}}(H^3|\mathcal{S}|^2|\mathcal{A}|/\epsilon)$. Viano et al. (2024) obtain $\widetilde{\mathcal{O}}(H^2 d/\epsilon^2)$ expert sample complexity with the same $\widetilde{\mathcal{O}}(H^4 d^3/\epsilon^2)$ interaction complexity for linear MDPs. Liu et al. (2021) establish $\widetilde{\mathcal{O}}(H^3 d^2/\epsilon^2)$ expert sample complexity and $\widetilde{\mathcal{O}}(H^4 d^3/\epsilon^2)$ interaction complexity for linear mixture MDPs. For general function approximation, Xu et al. (2024) provide bounds of $\widetilde{\mathcal{O}}(H^2 \log(\mathcal{N}(\mathcal{R}))/\epsilon^2)$ on expert sampling complexity and $\widetilde{\mathcal{O}}((H^4 d_{\text{GEC}} \log(\mathcal{N}(\mathcal{R})\mathcal{N}(\mathcal{Q})) + H^2)/\epsilon^2)$ on interaction complexity.

**Information-Theoretic Lower Bounds for Imitation Learning.** Several information-theoretic minimax lower bounds have been established for the number of expert policies. In the tabular MDP with a deterministic expert, Rajaraman et al. (2020) prove a lower bound of $\widetilde{\Omega}(|\mathcal{S}|H^2/\epsilon)$. Foster et al. (2024) further derive lower bounds of $\widetilde{\Omega}(H/\epsilon)$ for deterministic experts and $\widetilde{\Omega}(\epsilon^{-2})$ for stochastic experts. Similar lower bounds has been studied in Xu et al. (2022) with a minimax-optimal algorithm under a special class of MDPs. However, none of these information-theoretic lower bound targets for the sample complexity of online interaction. Moulin et al. (2025b) provides a lower bound for expert sampling complexity $\Omega(W_{\max}^2 (1 - \gamma)^{-2}\epsilon^{-2})$ and a lower bound for interaction complexity $\Omega(d(1 - \gamma)^{-2}\epsilon^{-2})$ for all imitation learning algorithms under discounted linear MDPs. We record part of these existing results in Table 1.

---

*TV-AIL considers a special class of tabular MDP referred to RBAS-MDP.

†The lower bound in Xu et al. (2022) is only for the TV-AIL algorithm thus not minimax lower bound.

‡ $N$ is the number of expert demonstrations and can be small.

## 3 PRELIMINARIES

**Time-homogeneous Episodic MDPs.** We consider the time-homogeneous episodic Markov decision processes (MDPs) denoted by $\mathcal{M} = (\mathcal{S}, \mathcal{A}, H, P^\star, r)$ by convention. Here, $\mathcal{S}$ and $\mathcal{A}$ are state and action spaces, $H$ is the length of each episode, $P^\star : \mathcal{S} \times \mathcal{A} \to \Delta(\mathcal{S})$ is the transition probability from state $s$ to $s'$ with action $a$. $\mathcal{R} \ni r : \mathcal{S} \times \mathcal{A} \to [0, 1]$ is the reward function. For any policy $\pi$ and reward $r$, the state-action value function $Q_{h;P^\star;r}^\pi(s, a)$ on stage $h$ is defined as:

$$Q_{h;P^\star;r}^\pi(s, a) = \mathbb{E}\left[\sum_{t=h}^H r(s_t, a_t) \Big| s_h = s, a_h = a, s_{t+1} \sim P^\star(\cdot|s_t, a_t), a_{t+1} \sim \pi(s_{t+1})\right],$$

and the value function is $V_{h;P^\star;r}^\pi(s) = Q_{h;P^\star;r}^\pi(s, \pi_h(s))$. The optimal values are defined by:

$$V_{h;P^\star;r}^\star(s) = \max_\pi V_{h;P^\star;r}^\pi(s); Q_{h;P^\star;r}^\star(s, a) = \max_\pi Q_{h;P^\star;r}^\pi(s, a).$$

For any value function $V : \mathcal{S} \to \mathbb{R}$ and stage $h \in [H]$, we define the first-order Bellman operator associated with policy $\pi$, reward function $r$, and transition model $P^\star$ as

$$(\mathcal{T}_h V_{h+1;P^\star;r}^\pi)(s) = r(s, a) + \mathbb{E}_{s' \sim P^\star(\cdot|s,a)}\left[V_{h+1;P^\star;r}^\pi(s')\right] := r(s, a) + [P^\star V_{h+1;P^\star;r}^\pi](s, a),$$

The conditional variance of the value function under state-action pair $(s, a)$ is defined as

$$[\mathbb{V}_{P^\star} V](s, a) = [P^\star V^2](s, a) - \left([P^\star V](s, a)\right)^2.$$

The variance of the cumulative reward collected by policy $\pi$ under reward function $r$ is defined by:

$$\text{VaR}_r^\pi = \mathbb{E}\left[\left(\sum_{h=1}^H r(s_h, a_h)\right)^2\right] - \left(V_{1;P^\star;r}^\pi(s_1)\right)^2,$$

the expectation is taken with respect to the trajectory distribution induced by the policy $\pi$ and transition kernel $P^\star$, i.e., $a_h \sim \pi_h(\cdot \mid s_h)$ and $s_{h+1} \sim P^\star(\cdot \mid s_h, a_h)$. In this paper, we assume the initial state is fixed at $s_1$. This can be generalized to a distribution of the initial state w.l.o.g..

**Online Adversarial Imitation Learning.** In this paper we consider the online adversarial imitation learning with offline demonstration. To be specific, the agent is given a *offline dataset* $\mathcal{D}_E = \{s_h^i, a_h^i\}_{h \in [H]}^{i \in [N]}$ containing $N$ trajectories rolled out by the *expert policy* $\pi^E$. The agent is allowed to *online interact* with the environment. Throughout the offline dataset and online interactions, the agent is performing in a *reward-free* paradigm such that it has no access to the collected reward in either offline demonstrations and online explorations.

The goal of the adversarial imitation learning (AIL) is to output a policy $\pi$ such that the policy $\pi$ and $\pi^E$ have similar behavior over the reward functions, usually defined by the typical AIL objective by

$$\min_\pi \max_{r \in \mathcal{R}} \mathbb{E}_s[V_{1;P^\star;r}^{\pi^E}(s) - V_{1;P^\star;r}^\pi(s)], \tag{3.1}$$

where the expectation is taken over the initial state distribution. In the online setting, we further define the regret of this AIL objective over $K$ rounds by

$$\text{Regret}(K) = \max_{r \in \mathcal{R}} \sum_{k=1}^K \mathbb{E}_s\left[V_{1;P^\star;r}^{\pi^E}(s) - V_{1;P^\star;r}^{\pi^k}(s)\right]. \tag{3.2}$$

**Model-based reinforcement learning with general function approximation.** We consider the model-based RL with general function approximation. In particular, we assume the function class of the transition kernel as $P \in \mathcal{P}$ and define the $\ell_p$ Eluder dimension of a function class

**Definition 3.1** ($\ell_p$ Eluder dimension, Russo & Roy 2013)**.** $\text{DE}_p(\mathcal{G}, \mathcal{X}, \epsilon)$ is the Eluder dimension for $\mathcal{X}$ with function class $\Phi$, when the longest *epsilon*-independent sequence $x_1, \cdots, x_L \subset \mathcal{X}$ enjoys the length $L$ less than $\text{DE}_p(\mathcal{G}, \mathcal{X}, \epsilon)$. In other words, there exists function $g \in \mathcal{G}$ such that for all $t \leq \text{DE}_p(\mathcal{G}, \mathcal{X}, \epsilon)$ we have $\sum_{l=1}^{t-1} |g(x_l)|^p \leq \epsilon^p$ and $|g(x_t)|^p \geq \epsilon^p$. Through out this paper, we work with the $\ell_1$ Eluder dimension defined by $d_E = \text{DE}_1(\mathcal{G}, \mathcal{S} \times \mathcal{A}, \epsilon)$ defined over the Hellinger distance class $\mathcal{G} : \{(s, a) \to \mathbb{H}^2(P^\star(s, a) \parallel P(s, a)) : P \in \mathcal{P}\}$ and $\epsilon = 1/(KH)$.

For the analysis under general function approximation, we leverage the following definitions on covering and bracketing numbers

---

**Algorithm 1** Model-based Adversarial Imitation Learning (`MB-AIL`)

---

**Input:** Number of iterations $K$, confidence radius $\beta$, expert dataset $\mathcal{D}_E$, $|\mathcal{D}_E| = N$
**Input:** Policy class $\Pi$, reward class $\mathcal{R}$, model class $\mathcal{P}$
1: Let $\pi^0(\cdot|s) = \text{Unif.}(\mathcal{A})$, $r^0(s,a) = 0$ for all $(s,a) \in \mathcal{S} \times \mathcal{A}$, $\mathcal{D}_0 = \emptyset$
2: **for** $k = 1, 2, \ldots, K$ **do**
3:     Collect $\tau_{k-1} = \{s_h^{k-1}, a_h^{k-1}\}_h$ using policy $\pi_{k-1}$, update $\mathcal{D}_k = \mathcal{D}_{k-1} \cup \{\tau_{k-1}\}$
4:     Compose the value difference loss $\mathcal{L}_{k-1}(r) = \sum_h r(s_h^{k-1}, a_h^{k-1}) - \frac{1}{N} \sum_{n,h} r(s_h^n, a_h^n)$
5:     Obtain reward function $r^k$ by running a no-regret algorithm to solve the online optimization problem with observed loss $\{\mathcal{L}_i(r)\}_{i=0}^{k-1}$ up to an optimization error tolerance $\epsilon_{\text{opt}}^r$
6:     Construct a version space $\widehat{\mathcal{P}}^k$:

$$\widehat{\mathcal{P}}^k = \left\{ P \in \mathcal{P} : \sum_{(s,a,s') \in \mathcal{D}^k} \log P(s'|s,a) \geq \max_{\widetilde{P} \in \mathcal{P}} \sum_{(s,a,s') \in \mathcal{D}^k} \log \widetilde{P}(s'|s,a) - \beta \right\}$$

7:     Set $(\pi^k, P^k) \leftarrow \text{argmax}_{\pi \in \Pi, P \in \widehat{\mathcal{P}}^k} V_{1;P;r^k}^\pi(s_1)$.
8: **end for**
**Output:** $\bar{\pi} = \text{Unif}(\{\pi^k\}_{k=1}^K)$

---

**Definition 3.2** ($\epsilon$-covering number). For a function class $\mathcal{F} \subseteq (\mathcal{X} \to \mathbb{R})$, the $\epsilon$-covering number of $\mathcal{F}$, denoted $\mathcal{N}_\epsilon(\mathcal{F})$, is the smallest integer $n \in \mathbb{N}$ such that there exists a subset $\mathcal{F}' \subseteq \mathcal{F}$ of size $|\mathcal{F}'| = n$ such that for all $f \in \mathcal{F}$, there exists $f' \in \mathcal{F}'$ with $\sup_{x \in \mathcal{X}} |f(x) - f'(x)| \leq \epsilon$.

**Definition 3.3** (Bracketing number, van de Geer 2000). Let $\mathcal{G}$ be a function class $\mathcal{X} \to \mathbb{R}$. Given two functions $l, u$ such that $l(x) \leq u(x)$ for all $x \in \mathcal{X}$, the bracket $[l, u]$ is defined as the set of functions $g \in \mathcal{G}$ satisfying $l(x) \leq g(x) \leq u(x)$ for all $x \in \mathcal{X}$. We call $[l, u]$ an $\epsilon$-bracket if $\|u - l\| \leq \epsilon$. The $\epsilon$-bracketing number of $\mathcal{G}$ with respect to a norm $\|\cdot\|$, denoted $\mathcal{N}_{[]}(\epsilon, \mathcal{G}, \|\cdot\|)$, is the minimum number of $\epsilon$-brackets needed to cover $\mathcal{G}$.

We build our analysis based on the bounded total reward assumption.

**Assumption 3.4** (Bounded cumulative rewards). For any reward function $r \in \mathcal{R}$, the cumulative reward collected by any possible trajectory $\{(s_h, a_h)\}_h$ is bounded by $0 \leq \sum_h r(s_h, a_h) \leq 1$.

Up to rescaling a factor $H$, Assumption 3.4 generalizes the standard reward scale assumption where $r_h \in [0, 1]$ for all $h \in [H]$. This assumption also ensures that the value function $V_{h;P^\star;r}^\pi(s)$ and action-value function $Q_{h;P^\star;r}^\pi(s,a)$ belong to the interval $[0, 1]$.

We also make the realizability assumption for the reward, policy and the transition kernel.

**Assumption 3.5.** The ground-truth reward function, transition dynamics, and optimal policy are *realizable*, i.e., $r^\star \in \mathcal{R}, P^\star \in \mathcal{P}, \pi^\star \in \Pi$.

These assumptions are common in the literature on model-based reinforcement learning and imitation learning, as they ensure that the learning problem is well-specified.

# 4 MODEL-BASED ADVERSARIAL IMITATION LEARNING

In this section, we introduce our proposed algorithm Model Based Adversarial Imitation Learning (`MB-AIL`). The full procedure is presented in Algorithm 1. The algorithm takes as input an expert dataset $\mathcal{D}_E$ consisting of multiple expert trajectories; three function classes $\Pi$, $\mathcal{R}$, and $\mathcal{P}$ for policy, reward, and transition model function approximations, respectively; and various hyperparameters, including the confidence radius for the version space, and the number of iterations. The algorithm begins by initializing the policy with a uniform distribution and setting the reward function to zero. In each iteration, it first collects a new trajectory $\tau_{k-1}$ using the current policy $\pi_{k-1}$ described in Line 3. Based on our hypothesis that the learning the expert policy $\pi^E$ within the policy class $\Pi$ can be decomposed into two procedure in learning the reward $r \in \mathcal{R}$ and learning the model $P \in \mathcal{P}$ separately, which is described as follows.

**Procedure A. Adversarial Reward Learning for $r \in \mathcal{R}$.** According to the adversarial online imitation learning objective in Eq. 3.1, the reward function must be optimized at each iteration $k$

to maximize the value gap between the expert policy $\pi^E$ and the behavioral policies $\pi^i$ generated in previous iterations $i < k$. Therefore, the reward optimization aims to maximize the adversarial objective $\max_r V_{1,P^\star,r}^{\pi^E} - V_{1,P^\star,r}^{\pi^k}$ by minimizing the following empirical loss $\mathcal{L}_{k-1}(r)$ with respect to the reward function $r \in \mathcal{R}$

$$\mathcal{L}_{k-1}(r) = \sum_h r(s_h^{k-1}, a_h^{k-1}) - \frac{1}{N} \sum_{n,h} r(s_h^n, a_h^n) := \widehat{V}_{1,P^\star,r}^{\pi^{k-1}}(s_1) - \widehat{V}_{1,P^\star,r}^{\pi^E}(s_1), \qquad (4.1)$$

where we denote $\widehat{V}(\cdot)$ as the empirical estimation of $V(\cdot)$. Since the series of loss function $\{\mathcal{L}_i\}_{i=0}^{k-1}$ is collected by an adversarial policy $\{\pi_i\}_{i=0}^{k-1}$ that is trying to maximize $V_{1,P^\star,r_k}^{\pi^k}(s_1)$, an no regret online optimization must be called in Line 5 to output the reward $r_k$. Over the $K$ rounds, the optimal solution for this reward learning is denoted by $\max_{r \in \mathcal{R}} \sum_k \left( \widehat{V}_{1,P^\star,r}^{\pi^k}(s_1) - \widehat{V}_{1,P^\star,r}^{\pi^E}(s_1) \right)$ and thus the optimization error is denoted by

$$\epsilon_{\mathrm{opt}}^r := \frac{1}{K} \max_{r \in \mathcal{R}} \sum_k \left( \widehat{V}_{1,P^\star,r_k}^{\pi^k}(s_1) - \widehat{V}_{1,P^\star,r_k}^{\pi^E}(s_1) \right) - \left( \widehat{V}_{1,P^\star,r}^{\pi^k}(s_1) - \widehat{V}_{1,P^\star,r}^{\pi^E}(s_1) \right). \qquad (4.2)$$

We note that Follow-the-Regularized-Leader (Shalev-Shwartz & Singer, 2007) can be used as a practical example of the no-regret online optimization algorithm and can obtain the optimization error $\epsilon_{\mathrm{opt}}^r = \mathcal{O}(\sqrt{K})$ optimization error in this adversarial reward learning procedure. Such an optimization-based reward learning method is also leveraged in Xu et al. (2024) where they assume the optimization is based on the individual reward function in each step $\mathcal{R}_h$. Ours by contrast assumes the reward function is time-homogeneous and therefore the optimization is based on all reward function $r \in \mathcal{R}$. Despite this difference, we note that the major difference of our approach is that MB-AIL is a model-based approach and is provably more efficient to capture the online interactions with carefully designed learning and analysis, as we will describe in detail below.

**Procedure B. Model and Policy Learning for $P \in \mathcal{P}$.** In parallel with learning the reward, in Line 6, we rely on an simple maximum likelihood estimator to learn the transition kernel information $P$ within a version space controlled by parameter $\beta$ described by

$$\widehat{\mathcal{P}}^k = \left\{ P \in \mathcal{P} : \sum_{(s,a,s') \in \mathcal{D}^k} \log P(s'|s,a) \geq \max_{\widetilde{P} \in \mathcal{P}} \sum_{(s,a,s') \in \mathcal{D}^k} \log \widetilde{P}(s'|s,a) - \beta \right\},$$

where the candidate transition kernel is of at most $\beta$ smaller than the 'optimal' version $\widetilde{P}$ in terms of their log likelihood. In practice, this procedure described in Line 6 can be mimic by training a series of world models by maximizing their likelihood.

Finally, with the obtained reward $r_k \in \mathcal{R}$ and the model $\widehat{\mathcal{P}}^k \subset \mathcal{P}$, the updated policy is obtained by the policy that obtains the maximum cumulative reward with the optimistic estimation over $P \in \widehat{\mathcal{P}}^k$, as described in Line 7 in Algorithm 1. We note that similar optimistic-based MLE approach has been widely applied in the theoretical literature on model-based RL Liu et al. (2023); Zhan et al. (2022); Wang et al. (2024) but with a fixed reward function $r$. However, in MB-AIL, the ground truth reward is unknown so that the policy optimization should be based on the current reward $r_k$ estimated in Line 5. We also note that this procedure can be efficiently implemented by ensembling the estimation of several models in practice Ye et al. (2023); Zhang et al. (2024a); Janner et al. (2019).

## 5 THEORETICAL ANALYSIS

We present a detailed theoretical analysis of Algorithm 1. In particular, we establish a horizon-free, second order bound under general function approximation for both expert demonstrations and the online interactions. We further present a information-theoretic minimax lower bound to statistically understand the boundary of the benefit for online interactions and the expert dataset, Based on the intuition that the policy class $\Pi$ can be decomposed into the reward $\mathcal{R}$ and the model $\mathcal{P}$.

### 5.1 UPPER BOUNDS FOR MODEL-BASED ADVERSARIAL IMITATION LEARNING (MB-AIL)

In this subsection, we establish an upper bound on the average regret of our proposed algorithm. We first start with the regret analysis for the online adversarial imitation learning.

**Theorem 5.1.** For any $\delta \in (0,1)$, let $\beta = 7 \log \left( K \mathcal{N}_{\mathcal{P}} / \delta \right)$. Under Assumptions 3.4 and 3.5, if FTRL (Shalev-Shwartz & Singer, 2007) is employed as the no-regret algorithm in Line 5 in

Algorithm 1, the averaged adversarial imitation learning regret defined in (3.2) for `MB-AIL` satisfies:

$$\frac{\text{Regret}(K)}{K} = \frac{1}{K} \max_{r \in \mathcal{R}} \sum_{k=1}^{K} \left[ V_{1;P^\star;r}^{\pi^E}(s) - V_{1;P^\star;r}^{\pi^k}(s) \right]$$

$$\leq \widetilde{\mathcal{O}} \left( \frac{1}{K} \sqrt{\sum_{k=1}^{K} \text{VaR}_k \, d_E \log \frac{\mathcal{N}_\mathcal{P}}{\delta}} + \frac{1}{K} \sqrt{\sum_{k=1}^{K} \text{VaR}_k \log \frac{\mathcal{N}_\mathcal{R}}{\delta}} + \frac{d_E \log \frac{\mathcal{N}_\mathcal{P}}{\delta} + \log \frac{\mathcal{N}_\mathcal{R}}{\delta}}{K} \right)$$

$$+ \widetilde{\mathcal{O}} \left( \sqrt{\frac{1}{N} \text{VaR}_E \log \frac{\mathcal{N}_\mathcal{R}}{\delta}} + \frac{\log \frac{\mathcal{N}_\mathcal{R}}{\delta}}{N} + \frac{1}{\sqrt{K}} \right),$$

where $\widetilde{\mathcal{O}}(\cdot)$ hides the polynominal logarithmic factors on $K, H, N$. $d_E$ is the eluder dimension defined based on Hellinger distance as described in Definition 3.1. We denote $\text{VaR}_k = \max_{r \in \mathcal{R}} \text{VaR}_r^{\pi^k}$ on the maximum variance of the cumulative return for policy $\pi^k$, $\text{VaR}_E = \max_{r \in \mathcal{R}} \text{VaR}_r^{\pi^k}$ is the short hand for the maximum variance collected by the expert policy $\pi^E$. $\mathcal{N}_\mathcal{P}$ is the bracketing number for the model class defined in Definition 3.3 while $\mathcal{N}_\mathcal{R}$ is the covering number for the reward class in Definition 3.2

The regret result in Theorem 5.1 can be immediate translated to the sample complexity bound for a mixed policy as stated in the following corollary.

**Corollary 5.2.** Let $\sigma^2 := \max_{\pi \in \Pi, r \in \mathcal{R}} \text{Var}_r^\pi$, where $\Pi$ is the policy class induced by the planning oracle $(\pi, P) = \arg\max_{\pi, P} V_{1,P,r}^\pi(s_1)$ used in Line 7 of Algorithm 1. For the mixed policy $\bar{\pi} = \text{Unif}\left(\{\pi^k\}_{k=1}^K\right)$ output by Algorithm 1, for any $\delta \in [0, 1]$, with probability at least $1 - \delta$, Algorithm 1 returns an $\epsilon$−optimal imitator, i.e., $\max_{r \in \mathcal{R}} \left[ V_{1,P^\star,r}^{\pi^E}(s_1) - V_{1,P^\star,r}^{\bar{\pi}}(s_1) \right] \leq \epsilon$ with expert and interaction sample complexity as:

$$K = \widetilde{\mathcal{O}} \left( \frac{1 + d_E \sigma^2 \log \frac{\mathcal{N}_\mathcal{P}}{\delta} + \sigma^2 \log \frac{\mathcal{N}_\mathcal{R}}{\delta}}{\epsilon^2} + \frac{d_E \log \frac{\mathcal{N}_\mathcal{P}}{\delta} + \log \frac{\mathcal{N}_\mathcal{R}}{\delta}}{\epsilon} \right), \quad N = \widetilde{\mathcal{O}} \left( \frac{\text{VaR}_E \log \frac{\mathcal{N}_\mathcal{R}}{\delta}}{\epsilon^2} + \frac{\log \frac{\mathcal{N}_\mathcal{R}}{\delta}}{\epsilon} \right).$$

**Remark 5.3** (Expert Sample Complexity). Corollary 5.2 reveals an $\widetilde{\mathcal{O}}\left(\text{VaR}_E \log(\mathcal{N}_\mathcal{R})\epsilon^{-2}\right)$ sample complexity for expert demonstrations when $\text{VaR}_E > 0$. When the reward is a $d_R$-linear as assumed in Viano et al. (2024), this matches the Viano et al. (2024); Xu et al. (2024) as $\widetilde{\mathcal{O}}(d_R \epsilon^{-2})$ with the *second-order* information $\text{VaR}_E \leq 1$. We also improve the $Hd$ factor compared with Liu et al. (2021) results for model-based AIL on linear mixture MDPs and makes this *horizon-free*.

**Remark 5.4** (Online Interaction Complexity). When $\sigma > 0$, Corollary 5.2 suggests that `MB-AIL` enjoys an $\widetilde{\mathcal{O}}\left(\sigma^2 \left(d_E \log \mathcal{N}_\mathcal{P} + \log(\mathcal{N}_\mathcal{R})\right) \epsilon^{-2}\right)$ interaction complexity. When reducing to linear mixture MDPs with $\mathcal{N}_\mathcal{P} = \widetilde{\mathcal{O}}(d)$ and $\mathcal{N}_\mathcal{R} = \widetilde{\mathcal{O}}(d)$, this $\widetilde{\mathcal{O}}(d^2 \sigma^2 \epsilon^{-2})$ sample complexity improves Liu et al. (2021) by an $H^2 d$ factor and obtains the *horizon-free* result with *second order* information. We note that the additional $1/\epsilon^2$ sample complexity comes from the adaption of the no-regret algorithm (FTRL) with $\epsilon_{\text{opt}}^r = \widetilde{\mathcal{O}}(1/\sqrt{K})$ and is dominated when $\mathcal{N}_\mathcal{P}$ or $\mathcal{N}_\mathcal{R}$ is large.

Besides directly compare our work with existing results, the second-order analysis reveals some interesting results regarding the deterministic expert or deterministic model.

**Remark 5.5** (Deterministic Model and Policy). When both the model class $\mathcal{P}$ and the policy class $\Pi$ only contain deterministic transition kernel and deterministic policies, it's easy to verify that $\sigma^2 = 0$ and $\text{VaR}_E = 0$. In such a case, the online interaction sample complexity is significantly reduced to $\widetilde{\mathcal{O}}(\epsilon^{-2} + (d_E \log \mathcal{N}_P + \mathcal{N}_\mathcal{R})\epsilon^{-1})$. If we relax the AIL gap to be $\max_{r \in \mathcal{R}} \left[ V_{1,P^\star,r}^{\pi^E}(s_1) - V_{1,P^\star,r}^{\bar{\pi}}(s_1) \right] \leq \epsilon + \epsilon_{\text{opt}}^r$ to accommodate the optimization error $\epsilon_{\text{opt}}^r$, an $\frac{1}{\epsilon}$-order sample complexity is built, which suggests that a deterministic model will benefit online interaction. Similarly, an $\mathcal{O}(\frac{\log \mathcal{N}_\mathcal{R}}{\epsilon})$ sample complexity is obtained.

Remark 5.5 indicates an $\epsilon^{-1}$ sample complexity in deterministic systems (disregarding the $\epsilon_{\text{opt}}^r$ term) and an $\epsilon^{-2}$ rate in stochastic systems, consistent with Foster et al. (2024). Crucially, however, the meaning of "stochasticity" and the mechanism behind the $\epsilon^{-2}$ dependence differ *fundamentally* between behavioral cloning and adversarial imitation learning as we would like to elaborate more.

**Remark 5.6** (Behavioral Cloning (BC) v.s. Adversarial Imitation Learning (AIL)). One fundamental difference between (offline) behavioral cloning (BC) and (online) adversarial imitation learning (AIL) is the target of estimation. BC fits the expert policy $\pi^E \in \Pi$ directly from demonstrations, whereas MB-AIL searches over rewards $r \in \mathcal{R}$ and learns the transition kernel $P \in \mathcal{P}$ before planning. Consequently, when additional structure constrains $\mathcal{P}$ or $\mathcal{R}$, AIL's effective hypothesis space can be much smaller than $\Pi$, making AIL statistically easier than BC. This distinction is reflected in the rates: for stochastic policies, our result depends on a sample complexity of $\widetilde{\mathcal{O}}(\sigma^2 \frac{\log \mathcal{N}_{\mathcal{R}}}{\epsilon^2})$, whereas Foster et al. (2024) obtain $\widetilde{\mathcal{O}}(\sigma^2 \frac{\log |\Pi|}{\epsilon^2})$. Moreover, Moulin et al. (2025a) provide an upper bound that depends only on the function class for $Q$-functions in the offline imitation setting, obtaining $\mathcal{O}(d(1-\gamma)^{-4}\epsilon^{-2})$ for linear $Q$-functions and $\mathcal{O}(\log \mathcal{N}_Q (1-\gamma)^{-8}\epsilon^{-4})$ for general function approximation under discounted MDPs. However, in general $\mathcal{N}_Q > \mathcal{N}_{\mathcal{R}}$, since multiple distinct $Q$-functions can correspond to the same reward function. Therefore, in particular, when $\mathcal{N}_{\mathcal{R}}$ is small, our analysis indicates that AIL needs fewer expert demonstrations with the help of online interaction.

**Remark 5.7** (Stochasticity in BC vs. AIL). As summarized in Table 2, BC and AIL react differently to stochasticity—especially when the expert policy $\pi^E$ is deterministic while the dynamics $P$ are stochastic (highlighted in red in Table 2). In this regime, Foster et al. (2024) obtain a

Table 2: Expert sample complexity under different kinds of stochasticity between BC (Foster et al., 2024) and MB-AIL (**ours**).

| Model $P$ \ Policy $\pi^E$ | Stochastic. | Deterministic. |
|---|---|---|
| Stochastic. | BC: $\sigma^2 \log \|\Pi\| \epsilon^{-2}$
AIL: $\sigma^2 \log \mathcal{N}_{\mathcal{R}} \epsilon^{-2}$ | BC: $\log \|\Pi\| \epsilon^{-1}$
AIL: $\sigma^2 \log \mathcal{N}_{\mathcal{R}} \epsilon^{-2}$ |
| Deterministic. | BC: $\sigma^2 \log \|\Pi\| \epsilon^{-2}$
AIL: $\sigma^2 \log \mathcal{N}_{\mathcal{R}} \epsilon^{-2}$ | BC: $\log \|\Pi\| \epsilon^{-1}$
AIL: $\log \mathcal{N}_{\mathcal{R}} \epsilon^{-1}$ |

sample complexity of $\widetilde{\mathcal{O}}(\log |\Pi| \epsilon^{-1})$ for BC with deterministic experts, whereas MB-AIL yields $\widetilde{\mathcal{O}}(\log \mathcal{N}_{\mathcal{R}} \epsilon^{-2})$. We interpret this as a fundamental gap between BC and AIL: BC only needs to match actions and is largely insensitive to transition stochasticity, whereas AIL must match the expert's occupancy measure, whose estimation necessarily reflects the stochasticity of $P$. Additionally, Rajaraman et al. (2020) report an $\widetilde{\mathcal{O}}(1/\epsilon)$ expert-sample complexity in tabular MDPs as a tabular-specific artifact noted by Foster et al. (2024). Outside this specific tabular result, we found that the $\epsilon^{-1}$ (deterministic) vs. $\epsilon^{-2}$ (stochastic) rates remain consistent across imitation-learning analyses.

Taken together, Remark 5.6 and Remark 5.7 described a clear theoretical separation between BC and AIL in terms of expert sample complexity: AIL is preferable when the reward class $\mathcal{R}$ is highly structured (Remark 5.6); in the extreme case with $|\mathcal{R}| = 1$, AIL reduces to standard RL and no expert data are needed. Conversely, BC is preferable when the expert policy class $\Pi$ is simple and deterministic while the transition dynamics are highly stochastic and complex (Remark 5.7).

## 5.2 MINIMAX LOWER BOUNDS FOR IMITATION LEARNING

In this subsection we present minimax lower bound on the sample complexity of the offline demonstrations $N = |\mathcal{D}_E|$ and the online interactions $K$. We start from constructing a hard instance as described below.

**Hard instance.** As illustrated in Figure 1, we consider an $(H+1)$-step time-homogeneous MDP. For a reward dimension $d_R$, kernel dimension $d_P$, and horizon $H$, The state has two components $\mathcal{S} = \widetilde{\mathcal{S}} \times \{0, 1\}$, where the first component $\widetilde{\mathcal{S}} = \mathbb{B}^{d_R} \cup \{\mathbf{s}_0\}$ and the second component is 0 if and only if the state is the initial state, we write the state as $\boxed{\mathbf{s}, 0}$ for these two components. The action space $\mathcal{A} = \mathbb{B}^{d_P} \times \{-1, +1\}$ concentration concatenation of two action where $\mathbf{a}_p \in \mathbb{B}^{d_P}$ is the first part and $a_R \in \{\pm 1\}$ as the second part. The agent always starts from a state $\boxed{\mathbf{s}, 0}$ where $\mathbf{s} \sim \text{Unif.}(\mathbb{B}^d_R)$. For the first step $h = 1$, the agent will either proceed to the same but absorbing

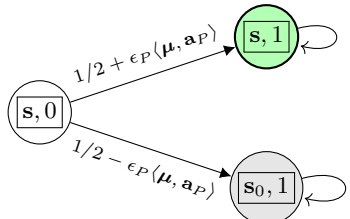

Figure 1: **Structure of the Hard Instance.** The reward can only be observed in Green states. The absorbing fail state is shown in gray.

state $\boxed{\mathbf{s}, 1}$ or fall into the special state $\boxed{\mathbf{s}_0, 1}$. For the rest of the steps $h \in (1, H]$, the agent will stay in their state $\boxed{\cdot, 1}$. The reward is only accessed at these regular absorbing state $\boxed{\mathbf{s}, 1}$. The MDP is parameterized with a parameter $\boldsymbol{\theta} \in \mathbb{M}^{d_R} \subseteq \mathbb{B}^{d_R}$ for reward and $\boldsymbol{\mu} \in \mathbb{B}^{d_P}$ for model, where $\mathbb{M}^{d_R}$ is a special *almost orthogonal* class of binary vectors defined in Appendix D.2. For a detailed

Table 3: **MuJoCo Results on Cumulative Rewards.** We conduct experiments on the Hopper, Walker2d, and Humanoid environments, evaluating cumulative rewards and comparing our approach with BC, GAIL, and OPT-AIL baselines. Our method achieves comparable performance on Hopper and Walker2d, while significantly outperforming all baselines on Humanoid. Results are averaged over three random seeds.

| Environment | Expert | BC | GAIL | OPT-AIL | MB-AIL (Ours) |
|---|---|---|---|---|---|
| Hopper | 3609.2 | 2856.7±42.5 | 3211.9±38.3 | 3438.6 ± 21.1 | **3451.3 ± 15.5** |
| Walker2d | 4636.5 | 2697.3 ± 97.2 | 3776.5 ± 64.3 | **4238.4± 39.2** | 4169.7 ± 48.3 |
| Humanoid | 5884.6 | 342.5 ± 23.8 | 1614.4 ± 118.1 | 2014.4 ± 342.2 | **5816.4 ± 15.2** |

formulation of the transition dynamics $P(\mathbf{s}'|\mathbf{s}, \mathbf{a}; \boldsymbol{\mu})$, the reward function $r(\mathbf{s}, \mathbf{a}; \boldsymbol{\theta})$, and the policy class $\pi(\mathbf{a}|\mathbf{s}; \widehat{\boldsymbol{\mu}}, \widehat{\boldsymbol{\theta}})$, we refer the reader to Appendix D.1.

Based on this construction, we are ready to present the lower bound for sample complexity.

**Theorem 5.8.** Given the hard instance described in Figure 1, for all $0 < \epsilon < \min\{\frac{1}{4d_P}, \frac{1}{4d_R}\}$, $d_R > 48$, and $\sigma^2 := \max_{\pi \in \Pi, r \in \mathcal{R}} \mathrm{Var}_{\pi, r} < 1$, any (online or offline) imitation learning algorithm that guarantees producing an $\epsilon$-suboptimal policy with probability at least $\frac{1}{2}$ under the averaged form of the regret defined in Eq. 3.2 by using $N$ expert demonstrations and $K$ iterations requires at least:

$$N = \Omega\left(\max\left\{\sigma^2/\epsilon^2, \sigma^2 \log^2 |\mathcal{R}|\right\}\right)$$

$$K = \Omega\left(\max\left\{\sigma^2 \log^2 |\mathcal{P}| \exp(-N)/\epsilon^2, \sigma^2 \epsilon^2 \log^2 |\mathcal{R}| \log^4 |\mathcal{P}| \exp(-N/\log^2 |\mathcal{P}|), 1\right\}\right),$$

with variance of the expert policy $\sigma_E^2 = \max_{r \in \mathcal{R}} \mathrm{Var}_{\pi_E, r}$ is bounded by $\frac{1}{2}\sigma^2 \leq \sigma_E^2 \leq \sigma^2$. This suggests that $N = \Omega(\sigma_E^2 \epsilon^{-2})$ and $K = \Omega(\sigma^2 \log^2 |\mathcal{P}| \exp(-N)\epsilon^{-2})$.

The detailed proof of Theorem 5.8 is deferred into Appendix D.

**Remark 5.9.** In Theorem 5.8, $K$ indexes the policies, which indicates that the number of online interactions is $K - 1$ since $\pi_K$ is learned from the offline dataset together with the previous $K - 1$ rounds of interaction. Specially, the purely offline setting corresponds to $K = 1$. More discussions about this interaction complexity in the offline setting is deferred to Appendix B. For stochastic policies, Foster et al. (2024) suggests an $\Omega(\sigma_E^2 \epsilon^{-2})$ expert demonstration. In contrast, Theorem 5.8 presents a fine-grained analysis on the sample complexity for both expert demonstration and online interaction: When online interaction is allowed, our results matches Foster et al. (2024) with an $\Omega(\sigma_E^2 \epsilon^{-2})$ expert sample complicity with an additional $\Omega(\log^2 |\mathcal{P}|\sigma^2 \exp(-N)\epsilon^{-2})$ lower bound for online interaction. This result implies that in the practical AIL case where the expert demonstration is limited and is much smaller than the model class with $N \ll \log^2 |\mathcal{P}|$, the online interaction conducted in `MB-AIL` can effectively assist the policy learning as $K = \Omega(\log^2 |\mathcal{P}|\sigma^2 \epsilon^{-2})$ with an minimax optimal sample complexity.

**Remark 5.10.** Together with Theorem 5.1, Theorem 5.8 shows that `MB-AIL` is minimax-optimal w.r.t. online interaction $K$ with only a logarithmic gap when given a small number of expert demonstrations $N$, and is within a $\log |\mathcal{R}|$ factor in its dependence on $N$. We hypothesize that removing this $\log |\mathcal{R}|$ gap may be intrinsically difficult as similar gaps ($\widetilde{\mathcal{O}}(\log |\Pi|/\epsilon^2)$ vs. $\widetilde{\Omega}(\epsilon^{-2})$) persist in Foster et al. (2024). Specifically, Xu et al. (2022) establish an $\Omega(|\mathcal{S}|\epsilon^{-2})$ lower bound for their TV-AIL algorithm in tabular MDPs, yet a *minimax* lower bound that explicitly exhibits a $\log |\Pi|$ or $\log |\mathcal{R}|$ dependence remains open even in the tabular setting.

## 6 EMPIRICAL RESULTS

We implement `MB-AIL` with deep neural networks and evaluate its performance on three standard MuJoCo benchmarks (Brockman et al., 2016), covering three environments (Hopper, Walker2d and Humanoid). We compare its performance against existing offline and online imitation learning baselines in terms of episode rewards and sample efficiency, showing that our practical algorithm matches or even surpasses existing approaches while highlighting the superior sample efficiency of our model-based approach. Details of the practical implementation are provided in Appendix E, the MuJoCo experimental results are presented in Section 6.1, and the ablation studies on network sizes are discussed in Section 6.2.

## 6.1 MuJoCo Experiments

We evaluate the proposed practical algorithm on several MuJoCo environments (Brockman et al., 2016), including Hopper, Walker2d, and Humanoid. For each task, we use 64 expert trajectories as demonstrations for training. We compare our method against Behavioral Cloning (BC), GAIL (Ho & Ermon, 2016), and OPT-AIL (Xu et al., 2024). The cumulative reward results are reported in Table 3. Our method achieves performance comparable to the best baselines on Hopper and Walker2d, while significantly outper-

Table 4: **MuJoCo Results on Interaction Complexity.** We compare our interaction complexity with OPT-AIL and show that our method requires significantly fewer steps to reach optimal performance.

| Environments | OPT-AIL | MB-AIL (Ours) |
|---|---|---|
| Hopper | $\sim$210K | $\sim$60K |
| Walker2d | $\sim$320K | $\sim$120K |
| Humanoid | $\sim$220K | $\sim$90K |

forming all compared methods on the more challenging Humanoid task, demonstrating its stronger capability in complex, high-dimensional control settings. All results are averaged over three random seeds. In addition, we compare the interaction complexity of our method with the model-free baseline OPT-AIL (Xu et al., 2024) in Table 4, which highlights the improved interaction efficiency of our model-based approach.

## 6.2 Ablation Studies on the Network Sizes

In this section, we conduct an ablation study on network capacity to empirically examine the claim in Remark 5.6 that $\log |\mathcal{R}| < \log |\Pi|$. Specifically, we vary both the depth and the hidden-layer dimensions of the reward and policy networks while keeping all other components fixed. In particular, we fix the policy network to a 2-layer MLP with 1024 hidden units and vary the reward network in Table 5 (a) and fix the reward network to a 2-layer MLP with 256 hidden units and vary the policy network in Table 5 (b). The results indicate that the reward function can be learned effectively with a relatively small network, whereas reducing the capacity of the policy network leads to a substantial degradation in performance. These findings provide empirical support for the claim that, in

Table 5: **Network architecture ablation.** in changing the network architecture for the policy network and reward network.

(a). Changing the reward network

| Reward Network | Episode Rewards |
|---|---|
| 2-layer, 256 hidden units | $4169.7 \pm 48.3$ |
| 2-layer, 128 hidden units | $4109.5 \pm 39.6$ |
| 1-layer, 256 hidden units | $4125.1 \pm 28.2$ |

(b). Changing the policy network

| Policy Network | Episode Rewards |
|---|---|
| 2-layer, 1024 hidden units | $4169.7 \pm 48.3$ |
| 2-layer, 256 hidden units | $704.9 \pm 112.5$ |

typical settings, the complexity of the reward class is significantly smaller than that of the policy class. The detailed ablation results are summarized in Table 5.

## 6.3 Additional Experiments

In addition to the experiments on MuJoCo, we conduct empirical analyses in GridWorld environments to examine the effects of different reward classes $\mathcal{R}$, environment stochasticity, impact of the model misspecification, and the benefits of online exploration. The corresponding results are presented in Appendix F, showing that a smaller reward class can lead to improved performance for online imitation learning algorithm, thereby supporting our theoretical analysis in Theorem 5.1.

## 7 Conclusion

We introduced `MB-AIL`, a model-based adversarial imitation learning algorithm, and established sharp statistical guarantees. We proved horizon-free, second-order upper bounds under general function approximation and complementary information-theoretic lower bounds, showing that `MB-AIL` is minimax-optimal in its use of online interaction when the expert demo is limited. We also show that `MB-AIL` is optimal within a $\log |\mathcal{R}|$ factor in its dependence on expert demonstrations. Experiments have been conducted to validate our theoretical claims and demonstrate that the `MB-AIL` matches or exceeds the sample efficiency of strong baselines across diverse environments. Our analysis clarifies how online interaction and system stochasticity impact the sample complexity of AIL and delineates regimes where AIL can outperform behavioral cloning.

REPRODUCIBILITY STATEMENT

We have made significant efforts to ensure the reproducibility of our results. The experiment details are documented in Appendix G, 6.1 and F. Our source code is provided to facilitate faithful reproduction of our experiments in the supplementary materials.

ETHICS STATEMENT

We have carefully reviewed the Code of Ethics and find that our work does not raise any significant ethical concerns. Our research does not involve human subjects, sensitive data, or potentially harmful applications. We believe our methodology and contributions align with principles of fairness, transparency, and research integrity.

ACKNOWLEDGMENTS

We thank the anonymous reviewers for their helpful comments. This research was supported by WZ's startup funding provided by the School of Data Science and Society at UNC Chapel Hill.

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

CONTENTS OF APPENDICES

# A  ADDITIONAL RELATED WORKS

**Variance-aware Second-order Analysis in Reinforcement Learning.** Variance-dependent (or second-order) bounds are instance-dependent guarantees that scale with the variance of the return. They have been studied in various settings, including tabular MDPs (Zanette & Brunskill, 2019; Zhou et al., 2023; Zhang et al., 2024c; Talebi & Maillard, 2018), linear mixture MDPs (Zhao et al., 2023), low-rank MDPs (Wang et al., 2023), and general function approximation (Wang et al., 2024). In imitation learning, Foster et al. (2024) also established variance-dependent upper and lower bounds for stochastic experts.

**RL with General Function Approximation** A large body of theoretical work has investigated reinforcement learning and imitation learning under general function approximation. To characterize the theoretical limits and better understand practical RL/IL algorithms, researchers have proposed a variety of statistical complexity measures, including Bellman Rank (Jiang et al., 2017), Witness Rank (Sun et al., 2019a), Eluder Dimension (Russo & Roy, 2013), Bellman Eluder Dimension (Jin et al., 2021), Decision Estimation Coefficient (DEC) (Foster et al., 2021), Admissible Bellman Characterization (Chen et al., 2022), Generalized Eluder dimension (Agarwal et al., 2023), and Generalized Eluder Coefficient (GEC) (Zhong et al., 2022), among others. For imitation learning, related works have considered general function approximation in the context of behavior cloning (Foster et al., 2024) and optimization-based adversarial imitation learning (Xu et al., 2024).

**Model-based Reinforcement Learning** Many works have conducted theoretical analyses of model-based reinforcement learning. Prior studies have examined model-based frameworks with rich function approximation in both online RL (Sun et al., 2019a; Foster et al., 2021; Song & Sun, 2021; Zhan et al., 2022; Wang et al., 2024) and offline RL (Wang et al., 2024; Uehara & Sun, 2021). In particular, Wang et al. (2024) derived a nearly horizon-free second-order bound for model-based RL in both online and offline settings, where the upper bound depends only logarithmically on $H$. In addition to theoretical analysis, numerous studies have investigated practical algorithms for model-based RL (Janner et al., 2019; Yu et al., 2020; Feinberg et al., 2018; Zhang et al., 2024b).

# B  ADDITIONAL DISCUSSIONS FOR THEOREM 5.8

**Extension to Purely Offline Behavioral Cloning** Our lower bound is originally established for the online setting. When extending it to the purely offline seeting, this corresponds to the regret with $K = 1$ since

$$\text{SubOpt}(\pi^1) = \text{Regret}(K = 1) = \max_{r \in \mathcal{R}} \mathbb{E}_s[V_{1;P^\star;r}^{\pi^E}(s) - V_{1;P^\star;r}^{\pi^1}(s)],$$

since $\pi^1$ only depends on the offline demonstrations. As a results, the online interactions $K$ by definition suffers from a **trivial lower bound** $K = \Omega(1)$ for purely offline setting. The full version of the lower bound for online interaction presented in Theorem 5.8 is therefore $\Omega(\max\{1, \sigma^2 \cdot \log^2 |\mathcal{P}| \exp(-N)\epsilon^{-2}\})$.

As a result, when the offline interaction $N$ is significantly large (which corresponds to the behavior cloning setting, precisely $N = \Omega(\log \frac{\sigma^2 \log^2 |\mathcal{P}|}{\epsilon^2}) = \Omega(\log \frac{\sigma \log |\mathcal{P}|}{\epsilon}))$, the lower bound becomes trivial since

$$K = \Omega(\max\{1, \sigma^2 \cdot \log^2 |\mathcal{P}| \exp(-N)\epsilon^{-2}\}) = \Omega(1).$$

We can further highlight that when the reward class is limited, Theorem 5.8 together with Theorem 5.1 suggests that it would be possible to obtain a smaller $N$ and complete the estimation on model $P$ through online interactions, which goes beyond Foster et al. (2024).

**Expert Policy Variance vs. Maximum Policy Variance** Foster et al. (2024) derive a variance-aware lower bound that depends only on the expert policy's variance. In contrast, our lower bound generally depends on the *maximum* variance over all policies in the policy class. In the construction used for Theorem 5.8, this implies that the expert variance $\sigma_E^2$ must be of the same order as this maximum variance, i.e., a lower bound on the expert variance is required, as stated in the following lemma:

**Lemma B.1.** By the construction of the hard instance for Theorem 5.8, the variance of the expert policy $\sigma_E^2 = \max_{r \in \mathcal{R}} \text{VaR}_{\pi^E, r}$ is lower bounded by the MDP parameter $\tau$:

$$\sigma_E^2 \geq \tfrac{1}{2}\tau^2.$$

*Proof.* Given the definition of the expert policy variance:

$$\sigma_E^2 = \max_{r \in \mathcal{R}} \text{VaR}_{\pi^E, r} = \max_r \mathbb{E}\left[\left(\sum_{h=1}^H r(\mathbf{s}_h, \mathbf{a}_h)\right)^2\right] - \left(\mathbb{E}[V_{1;P^\star;r}^{\pi^E}(\mathbf{s})]\right)^2,$$

by Eq. D.3 and the definition of the reward in the constructed hard instance in Appendix D.1, let the parameterization of the reward be $\widetilde{\boldsymbol{\theta}}$ and the expert policy be $\pi(\mathbf{a}|\mathbf{s}; \boldsymbol{\mu}^*, \boldsymbol{\theta}^*)$, we can have:

$$\mathbb{E}\left[\left(\sum_{h=1}^H r(\mathbf{s}_h, \mathbf{a}_h)\right)^2\right] = \left(\tfrac{1}{2} + \epsilon_\pi \epsilon_P d_R \langle \boldsymbol{\mu}^*, \boldsymbol{\mu}^* \rangle\right) \tau^2 \mathbb{E}\left(\tfrac{1}{4} + \tfrac{a_R}{2d_R}\langle \widetilde{\boldsymbol{\theta}}, \mathbf{s} \rangle + \tfrac{1}{4d_R^2}\langle \widetilde{\boldsymbol{\theta}}, \mathbf{s} \rangle^2\right),$$

$$\left(\mathbb{E}[V_{1;P^\star;r}^{\pi^E}(\mathbf{s})]\right)^2 = \left[\left(\tfrac{1}{2} + \epsilon_\pi \epsilon_P d_R \langle \boldsymbol{\mu}^*, \boldsymbol{\mu}^* \rangle\right) \cdot \left(\tfrac{1}{2}\tau + \tfrac{\epsilon_\pi \tau}{d_R}\langle \widetilde{\boldsymbol{\theta}}, \mathbf{s} \rangle \langle \boldsymbol{\theta}^*, \mathbf{s} \rangle\right)\right]^2.$$

By simple algebra, since we have $0 \leq \epsilon_p \leq 1/(2d_P)$ and $0 \leq \epsilon_\pi \leq 1/d_R$, we can derive the lower bound for $\sigma_E^2$:

$$\begin{aligned}
\sigma_E^2 &= \max_{r \in \mathcal{R}} \mathbb{E}\left[\left(\sum_{h=1}^H r(\mathbf{s}_h, \mathbf{a}_h)\right)^2\right] - \left(\mathbb{E}[V_{1;P^\star;r}^{\pi^E}(\mathbf{s})]\right)^2 \\
&\geq \left(\tfrac{1}{2} + \epsilon_\pi \epsilon_P d_R \langle \boldsymbol{\mu}^*, \boldsymbol{\mu}^* \rangle\right)\tau^2 \\
&\geq \tfrac{1}{2}\tau^2,
\end{aligned}$$

where the first inequality is given by:

$$\begin{aligned}
\max_r -\left(\mathbb{E}[V_{1;P^\star;r}^{\pi}(\mathbf{s})]\right)^2 &= -\min_{r \in \mathcal{R}}\left(\mathbb{E}[V_{1;P^\star;r}^{\pi}(\mathbf{s})]\right)^2 \\
&= -\min_{\widetilde{\boldsymbol{\theta}}}\left[\left(\tfrac{1}{2} + \epsilon_\pi \epsilon_P d_R \langle \boldsymbol{\mu}^*, \boldsymbol{\mu}^* \rangle\right) \cdot \left(\tfrac{1}{2}\tau + \tfrac{\epsilon_\pi \tau}{d_R}\langle \widetilde{\boldsymbol{\theta}}, \mathbf{s} \rangle \langle \boldsymbol{\theta}^*, \mathbf{s} \rangle\right)\right]^2 \\
&\geq -\min_{\widetilde{\boldsymbol{\theta}}}\left[\left(\tfrac{1}{2} + \tfrac{1}{2d_P d_R} d_R \langle \boldsymbol{\mu}^*, \boldsymbol{\mu}^* \rangle\right) \cdot \left(\tfrac{1}{2}\tau + \tfrac{\tau}{d_R^2}\langle \widetilde{\boldsymbol{\theta}}, \mathbf{s} \rangle \langle \boldsymbol{\theta}^*, \mathbf{s} \rangle\right)\right]^2 \\
&= 0,
\end{aligned}$$

and

$$\begin{aligned}
\max_r \mathbb{E}\left[\left(\sum_{h=1}^H r(\mathbf{s}_h, \mathbf{a}_h)\right)^2\right] &= \max_{\widetilde{\boldsymbol{\theta}}}\left(\tfrac{1}{2} + \epsilon_\pi \epsilon_P d_R \langle \boldsymbol{\mu}^*, \boldsymbol{\mu}^* \rangle\right)\tau^2 \mathbb{E}\left(\tfrac{1}{4} + \tfrac{a_R}{2d_R}\langle \widetilde{\boldsymbol{\theta}}, \mathbf{s} \rangle + \tfrac{1}{4d_R^2}\langle \widetilde{\boldsymbol{\theta}}, \mathbf{s} \rangle^2\right) \\
&= \left(\tfrac{1}{2} + \epsilon_\pi \epsilon_P d_R \langle \boldsymbol{\mu}^*, \boldsymbol{\mu}^* \rangle\right)\tau^2,
\end{aligned}$$

which concludes the proof. □

Since we show in the proof of Theorem 5.8 that $\text{Var}_{\pi, r} \leq \tau^2$ for all $\pi \in \Pi$ and $r \in \mathcal{R}$, the expert variance $\sigma_E^2$ cannot be significantly smaller than $\sigma^2 := \max_{\pi \in \Pi, r \in \mathcal{R}} \text{Var}_{\pi, r}$. In particular, $\sigma_E^2$ must be of the same order as $\sigma^2$. This ensures that our construction is consistent with the result in Foster et al. (2024).

## C   PROOF OF THE UPPER BOUNDS

### C.1   UPPER BOUND WITH FINITE FUNCTION CLASS

We first aim to upper bound the average regret of our proposed algorithm under finite function classes, i.e., when both $\mathcal{P}$ and $\mathcal{R}$ are finite. Formally, this upper bound can be stated in the following theorem:

**Theorem C.1.** For any $\delta \in (0, 1)$, let $\beta = 4\log(K|\mathcal{P}|/\delta)$. Under Assumptions 3.4 and 3.5, if FTRL (Shalev-Shwartz & Singer, 2007) is employed as the no-regret algorithm, and $\mathcal{R}$ and $\mathcal{P}$ are finite function classes, the average adversarial imitation learning regret defined in 3.1 satisfies:

$$(1/K)\text{Regret}(K) = \frac{1}{K}\max_{r \in \mathcal{R}}\sum_{k=1}^K \left[V_{1;P^\star;r}^{\pi^E}(s) - V_{1;P^\star;r}^{\pi^k}(s)\right]$$

$$\leq \widetilde{\mathcal{O}}\left(\frac{1}{K}\sqrt{\sum_{k=1}^{K}\mathrm{VaR}_{\pi,k}\, d_E \log\frac{|\mathcal{P}|}{\delta}} + \frac{1}{K}d_E \log\frac{|\mathcal{P}|}{\delta} + \frac{1}{\sqrt{K}} + \frac{\log(|\mathcal{R}|/\delta)}{K}\right)$$

$$+ \widetilde{\mathcal{O}}\left(\frac{\log(|\mathcal{R}|/\delta)}{N} + \sqrt{\frac{1}{K}\sum_{k=1}^{K}\mathrm{VaR}_{\pi,k}\,\log(|\mathcal{R}|/\delta)} + \sqrt{\frac{1}{N}\mathrm{VaR}_{\pi^E}\,\log(|\mathcal{R}|/\delta)}\right),$$

where $\widetilde{\mathcal{O}}(\cdot)$ hides the polynominal logarithmic factors on $K, H, N$. $d_E$ is the eluder dimension defined based on Hellinger distance as described in Definition 3.1. We denote $\mathrm{VaR}_k = \max_{r \in \mathcal{R}} \mathrm{VaR}_r^{\pi^k}$ on the maximum variance of the cumulative return for policy $\pi^k$, $\mathrm{VaR}_E = \max_{r \in \mathcal{R}} \mathrm{VaR}_r^{\pi^k}$ is the short hand for the maximum variance collected by the expert policy $\pi^E$.

## C.2 Proof of Theorem C.1

According to the definition of average regret in Eq. 3.2, we can decompose it into two components—reward learning and policy learning—and analyze them separately:

$$\frac{1}{K}\mathrm{Regret}(K) = \max_{\widetilde{r}\in\mathcal{R}}\frac{1}{K}\sum_{k=1}^{K}\left(V_{1;P^\star;\widetilde{r}}^{\pi^E} - V_{1;P^\star;\widetilde{r}}^{\pi^k}\right)$$

$$= \max_{\widetilde{r}\in\mathcal{R}}\frac{1}{K}\sum_{k=1}^{K}\underbrace{\left(V_{1;P^\star;\widetilde{r}}^{\pi^E} - V_{1;P^\star;\widetilde{r}}^{\pi^k} - (V_{1;P^\star;r^k}^{\pi^E} - V_{1;P^\star;r^k}^{\pi^k})\right)}_{\text{T1: reward error}}$$

$$+ \underbrace{\frac{1}{K}\sum_{k=1}^{K}\left(V_{1;P^\star;r^k}^{\pi^E} - V_{1;P^\star;r^k}^{\pi^k}\right)}_{\text{T2: policy error}}.$$

This regret decomposition is standard in AIL analysis. Since the learned reward is not required to exactly match the ground-truth reward function $r^\star$, it is natural to define the reward error as the difference in value gaps between the expert and behavioral distributions. We provide upper bounds for the reward and policy errors under finite function classes in the following lemmas:

**Lemma C.2** (Reward Error Upper Bound). For any $\delta \in (0,1)$, using FTRL (Shalev-Shwartz & Singer, 2007) as the no-regret algorithm, given a finite reward function class $\mathcal{R}$, with probability $1 - \delta$, the average regret for reward optimization is bounded:

$$\max_{\widetilde{r}\in\mathcal{R}}\frac{1}{K}\sum_{k=1}^{K}V_{1;P^\star;\widetilde{r}}^{\pi^E} - V_{1;P^\star;\widetilde{r}}^{\pi^k} - (V_{1;P^\star;r^k}^{\pi^E} - V_{1;P^\star;r^k}^{\pi^k})$$

$$\lesssim \mathcal{O}\left(\frac{1}{\sqrt{K}} + \sqrt{\frac{1}{N}\left(\max_{r\in\mathcal{R}}\mathrm{VaR}_{\pi^E;r}\right)\log(|\mathcal{R}|/\delta)} + \frac{\log(|\mathcal{R}|/\delta)}{N}\right.$$

$$\left. + \sqrt{\frac{1}{K}\sum_{k=1}^{K}\max_{r\in\mathcal{R}}\mathrm{VaR}_{\pi^k;r}\log(|\mathcal{R}|/\delta)} + \frac{\log(|\mathcal{R}|/\delta)}{K}\right),$$

where $\mathrm{VaR}_{\pi;r}$ denotes the variance of accumulative rewards over a horizon $H$.

**Lemma C.3** (Policy Error Upper Bound). For any $\delta \in (0,1)$, with a finite model class $\mathcal{P}$, let $\beta = 4\log(K|\mathcal{P}|/\delta)$. With probability $1 - \delta$, the average regret of the policy optimization is bounded:

$$\frac{1}{K}\sum_{k=1}^{K}\left(V_{1;P^\star;r^k}^{\pi^E} - V_{1;P^\star;r^k}^{\pi^k}\right)$$

$$\lesssim \mathcal{O}\left(\frac{1}{K}\sqrt{\sum_{k=1}^{K}\max_{r\in\mathcal{R}}\mathrm{VaR}_{\pi^k,r}\cdot d_E \log(KH|\mathcal{P}|/\delta)\log(KH)}\right.$$

$$+ \frac{1}{K} d_E \log(KH|\mathcal{P}|/\delta) \log(KH)\Bigg),$$

where $d_E = DE_1(\mathcal{G}, \mathcal{S} \times \mathcal{A}, 1/KH)$ is the Eluder dimension and $\mathrm{VaR}_{\pi,k} = \max_{r \in \mathcal{R}} \mathrm{VaR}_{\pi^k, r}$ is the maximum variance for $\pi^k$.

In the proof steps of Lemma C.2, the reward error T1 is decomposed into an optimization error and a value estimation error. The optimization error, stemming from the no-regret algorithm in Line 5 of Algorithm 1 and related to $\epsilon_{\mathrm{opt}}^r$ in Eq. 4.2, is bounded by $\mathcal{O}(1/\sqrt{K})$ when using FTRL (Shalev-Shwartz & Singer, 2007). The value estimation error, which represents the difference between the empirical value $\widehat{V}_{1;P^\star;r}^{\pi^k}$ and the true value function $V_{1;P^\star;r}^{\pi^k}$, is bounded using standard Bernstein-style techniques and union bounds for a finite reward class. For an infinite reward class, as detailed in Appendix C.5, we establish the bound by constructing a $\rho$-cover using the covering number from Definition 3.2, which enables the use of union bounds followed by a Bernstein-style analysis. We present the analysis for the finite reward class in the proof of Lemma C.2, and the analysis for the infinite reward function class in Lemma C.14.

Regarding the policy error T2, we generally follow the proof steps in Wang et al. (2024) but with a changing $r^k$ during iterations instead of a fixed $r^\star$. Overall, the bound is constructed by first applying the standard MLE analysis on the estimation of transition model and a careful analysis on training-to-testing distribution transfer via Eluder dimension adopted in prior work (Wang et al., 2024). After this analysis, we can obtain a variance-aware upper bound for the policy error with a logarithmic dependency on the horizon $H$. We provide the detailed analysis in the proof of Lemma C.3 for the finite model class, and in Lemma C.15 for the infinite model class.

Finally, by combining Lemma C.2 and Lemma C.3, we complete the proof of Theorem C.1, which is the upper bound with finite function classes. For the results under general function approximation in Theorem 5.1, we provide the detailed analysis in Appendix C.5.

### C.3 UPPER BOUND FOR THE REWARD ERROR

#### C.3.1 TECHNICAL LEMMAS

In this subsection, we present several technical lemmas that are instrumental in deriving the bound on the reward error.

**Lemma C.4** (Bernstein Inequality, Bernstein 1924)**.** Let $X_1, X_2, \ldots, X_n$ be i.i.d. random variables, with mean $\mu = \mathbb{E}[X_i]$, empirical mean $\widehat{\mu} = \frac{1}{n} \sum_{i=1}^n X_i$, and variance $\sigma^2 = \mathrm{Var}(X_i)$. Then, for any $\delta \in (0, 1)$, with probability at least $1 - \delta$, the following inequality holds:

$$|\widehat{\mu} - \mu| \leq \sqrt{\frac{2\sigma^2 \log(2/\delta)}{n}} + \frac{2 \log(2/\delta)}{3n}.$$

**Lemma C.5** (Azuma-Bernstein Inequality, Azuma 1967)**.** Let $(X_k)_{k=1}^n$ be a martingale difference sequence with respect to filtration $(\mathcal{F}_k)_{k=0}^n$, i.e., $\mathbb{E}[X_k \mid \mathcal{F}_{k-1}] = 0$ almost surely, and suppose the increments satisfy

$$|X_k| \leq b \quad \text{almost surely for all } k,$$

and the conditional variances satisfy

$$\sigma_k^2 := \mathbb{E}[X_k^2 \mid \mathcal{F}_{k-1}].$$

Define the total conditional variance

$$V_n := \sum_{k=1}^n \sigma_k^2.$$

Then, for any $\delta \in (0, 1)$, with probability at least $1 - \delta$,

$$\left| \sum_{k=1}^n X_k \right| \leq \sqrt{2V_n \log\left(\frac{2}{\delta}\right)} + \frac{b}{3} \log\left(\frac{2}{\delta}\right).$$

### C.3.2 PROOF OF LEMMA C.2

The reward error can be decomposed into two components: the optimization error of the no-regret algorithm and the estimation error of the value function. We begin by providing a formal description on optimization error introduced in Eq. 4.2:

**Definition C.6** (Xu et al. (2024)). Given a sequence of policies $\{\pi^k\}_{k=1}^K$, suppose a no-regret reward optimization algorithm sequentially outputs reward functions $r^1, r^2, \ldots, r^K$. The reward optimization error $\epsilon_{\text{opt}}^r$ is defined as:

$$\epsilon_{\text{opt}}^r := \frac{1}{K} \max_{r \in \mathcal{R}} \sum_{k=1}^K \left( \widehat{V}_{1;P^\star;r^k}^{\pi^k} - \widehat{V}_{1;P^\star;r^k}^{\pi^E} - \left( \widehat{V}_{1;P^\star;r}^{\pi^k} - \widehat{V}_{1;P^\star;r}^{\pi^E} \right) \right).$$

This error can be bounded by $O(1/\sqrt{K})$ when using the Follow-the-Regularized-Leader (FTRL) algorithm (Shalev-Shwartz & Singer, 2007) as the no-regret learner. We then define the empirical value estimates under both the learned and expert policies to facilitate bounding the value estimation error:

**Definition C.7** (Xu et al. (2024)). The empirical value of policy $\pi^k$ under reward function $r \in \mathcal{R}$ is defined as:

$$\widehat{V}_{1;P^\star;r}^{\pi^k} := \sum_{h=1}^H r(s_h^k, a_h^k),$$

where $\{(s_h^k, a_h^k)\}_{h=1}^H$ is a trajectory sampled from policy $\pi^k$.
For the expert policy $\pi^E$, given an expert dataset $\mathcal{D}_E = \{\tau_1, \tau_2, \ldots, \tau_N\}$, the empirical value is defined as:

$$\widehat{V}_{1;P^\star;r}^{\pi^E} := \frac{1}{N} \sum_{\tau \in \mathcal{D}_E} \sum_{h=1}^H r(s_h(\tau), a_h(\tau)).$$

With Lemma C.4, C.5 and Definition C.6, C.7, we're now ready to demonstrate the proof for Lemma C.2.

*Proof of Lemma C.2.* We first apply the following decomposition to separate the reward error into a reward optimization component and a value estimation component, as described in Appendix C.2:

$$\max_{\widetilde{r} \in \mathcal{R}} \frac{1}{K} \sum_{k=1}^K V_{1;P^\star;\widetilde{r}}^{\pi^E} - V_{1;P^\star;\widetilde{r}}^{\pi^k} - (V_{1;P^\star;r^k}^{\pi^E} - V_{1;P^\star;r^k}^{\pi^k})$$

$$= \underbrace{\max_{\widetilde{r} \in \mathcal{R}} \frac{1}{K} \sum_{k=1}^K \left( \widehat{V}_{1;P^\star;\widetilde{r}}^{\pi^E} - \widehat{V}_{1;P^\star;\widetilde{r}}^{\pi^k} - (\widehat{V}_{1;P^\star;r^k}^{\pi^E} - \widehat{V}_{1;P^\star;r^k}^{\pi^k}) \right)}_{\text{T3: optimization error}}$$

$$+ \underbrace{\max_{\widetilde{r} \in \mathcal{R}} V_{1;P^\star;\widetilde{r}}^{\pi^E} - \widehat{V}_{1;P^\star;\widetilde{r}}^{\pi^E} + \frac{1}{K} \sum_{k=1}^K \widehat{V}_{1;r^k;P^\star}^{\pi^E} - V_{1;r^k;P^\star}^{\pi^E} + \widehat{V}_{1;P^\star;\widetilde{r}}^{\pi^k} - V_{1;P^\star;\widetilde{r}}^{\pi^k} + V_{1;P^\star;r^k}^{\pi^k} - \widehat{V}_{1;P^\star;r^k}^{\pi^k}}_{\text{T4: estimation error}}.$$

Term T3 corresponds directly to the optimization error of the no-regret algorithm, as defined in Definition C.6:

$$\text{T3} = \max_{\widetilde{r} \in \mathcal{R}} \frac{1}{K} \sum_{k=1}^K \widehat{V}_{1;P^\star;\widetilde{r}}^{\pi^E} - \widehat{V}_{1;P^\star;\widetilde{r}}^{\pi^k} - (\widehat{V}_{1;P^\star;r^k}^{\pi^E} - \widehat{V}_{1;P^\star;r^k}^{\pi^k})$$

$$= \frac{1}{K} \max_{\widetilde{r} \in \mathcal{R}} \sum_{k=1}^K \widehat{V}_{1;P^\star;r^k}^{\pi^k} - \widehat{V}_{1;P^\star;r^k}^{\pi^E} - (\widehat{V}_{1;P^\star;\widetilde{r}}^{\pi^k} - \widehat{V}_{1;P^\star;\widetilde{r}}^{\pi^E})$$

$$= \epsilon_{\text{opt}}^r.$$

Having bounded the optimization error, we now turn to bounding the estimation error:

$$\text{T4} = \max_{\widetilde{r} \in \mathcal{R}} \frac{1}{K} \sum_{k=1}^{K} \left( V_{1;P^\star;\widetilde{r}}^{\pi^E} - \widehat{V}_{1;P^\star;\widetilde{r}}^{\pi^E} \right) + \left( \widehat{V}_{1;P^\star;r^k}^{\pi^E} - V_{1;P^\star;r^k}^{\pi^E} \right)$$
$$+ \left( \widehat{V}_{1;P^\star;\widetilde{r}}^{\pi^k} - V_{1;P^\star;\widetilde{r}}^{\pi^k} \right) + \left( V_{1;P^\star;r^k}^{\pi^k} - \widehat{V}_{1;P^\star;r^k}^{\pi^k} \right).$$

We aim to derive a variance-aware bound by providing an analysis using a Bernstein-style inequality. Following the definition of variances in Section 3, we first focus on the estimation error incurred when approximating $V_{1;P^\star;r}^{\pi^E}$ with its empirical counterpart $\widehat{V}_{1;P^\star;r}^{\pi^E}$. By applying Bernstein's inequality, we obtain that, for any fixed reward function $r$, the estimation error is bounded with probability at least $1 - \delta$:

$$\left| \widehat{V}_{1;P^\star;r}^{\pi^E} - V_{1;P^\star;r}^{\pi^E} \right| = \left| \frac{1}{N} \sum_{\tau \in \mathcal{D}_E} \sum_{h=1}^{H} r(s_h(\tau), a_h(\tau)) - \mathbb{E}_{\pi^E} \left[ \sum_{h=1}^{H} r(s_h, a_h) \right] \right|$$
$$\leq \sqrt{\frac{2 \sum_{\tau \in \mathcal{D}_E} \mathbb{V}_{\pi^E} \left[ \sum_{h=1}^{H} r(s_h(\tau), a_h(\tau)) \right] \log(2/\delta)}{N}} + \frac{2 \log(2/\delta)}{3N}$$
$$= \sqrt{\frac{2}{N} \text{VaR}_{\pi^E;r} \log(2/\delta)} + \frac{2 \log(2/\delta)}{3N}. \tag{C.1}$$

Since here we only consider finite reward class $\mathcal{R}$, by stardard union bound:

$$\left| V_{1;P^\star;r}^{\pi^E} - V_{1;P^\star;r}^{\pi^E} \right| \leq \sqrt{\frac{2}{N} \text{VaR}_{\pi^E;r} \log(2|\mathcal{R}|/\delta)} + \frac{2 \log(2|\mathcal{R}|/\delta)}{3N}$$
$$\leq \sqrt{\frac{2}{N} \text{VaR}_{\pi^E;r} \log(2|\mathcal{R}|)/\delta)} + \frac{2 \log(2|\mathcal{R}|)/\delta)}{3N}.$$

Therefore, we can bound the first two terms in the estimation error:

$$\max_{\widetilde{r} \in \mathcal{R}} \frac{1}{K} \sum_{k=1}^{K} \left( V_{1;P^\star;\widetilde{r}}^{\pi^E} - \widehat{V}_{1;P^\star;\widetilde{r}}^{\pi^E} \right) + \left( \widehat{V}_{1;P^\star;r^k}^{\pi^E} - V_{1;P^\star;r^k}^{\pi^E} \right)$$
$$\leq 2 \sqrt{\frac{2}{N} \left( \max_{r \in \mathcal{R}} \text{VaR}_{\pi^E;r} \right) \log(2|\mathcal{R}|/\delta)} + \frac{4 \log(2|\mathcal{R}|/\delta)}{3N}. \tag{C.2}$$

We then focus on the value difference with respect to $\pi^k$ for any $r \in \mathcal{R}$ using Azuma-Bernstein's inequality, with probability at least $1 - \delta$:

$$\frac{1}{K} \sum_{k=1}^{K} \left| \widehat{V}_{1;P^\star;r}^{\pi^k} - V_{1;P^\star;r}^{\pi^k} \right| = \frac{1}{K} \sum_{k=1}^{K} \left| \sum_{h=1}^{H} r(s_h, a_h) - \mathbb{E}_{\pi^k} \left[ \sum_{h=1}^{H} r(s_h, a_h) \right] \right|$$
$$\leq \sqrt{\frac{2}{K} \sum_{k=1}^{K} \mathbb{V}_{\pi^k} \left[ \sum_{h=1}^{H} r(s_h, a_h) \right] \log(2/\delta)} + \frac{2 \log(2/\delta)}{3K}$$
$$= \sqrt{\frac{2}{K} \sum_{k=1}^{K} \text{VaR}_{\pi^k;r} \log(2/\delta)} + \frac{2 \log(2/\delta)}{3K}.$$

Following the analysis of the terms involving $\pi^E$, we apply the union bound:

$$\frac{1}{K} \sum_{k=1}^{K} \left| V_{1;P^\star;r}^{\pi^k} - V_{1;P^\star;r}^{\pi^k} \right| \leq \sqrt{\frac{2}{K} \sum_{k=1}^{K} \text{VaR}_{\pi^k;r} \log(2|\mathcal{R}|/\delta)} + \frac{2 \log(2|\mathcal{R}|/\delta)}{3K}.$$

In this case, the terms associated with $\pi^k$ are bounded by Bernstein-style variance-aware upper bounds, with probability at least $1 - \delta$:

$$\max_{\widetilde{r} \in \mathcal{R}} \frac{1}{K} \sum_{k=1}^K \left( \widehat{V}_{1;P^\star;\widetilde{r}}^{\pi^k} - V_{1;P^\star;\widetilde{r}}^{\pi^k} \right) + \left( V_{1;P^\star;r^k}^{\pi^k} - \widehat{V}_{1;P^\star;r^k}^{\pi^k} \right)$$

$$\leq 2\sqrt{\frac{2}{K} \sum_{k=1}^K (\max_{r \in \mathcal{R}} \mathrm{VaR}_{\pi^k;r}) \log(2|\mathcal{R}|/\delta)} + \frac{4\log(2|\mathcal{R}|/\delta)}{3K}. \tag{C.3}$$

Then finally by union bound, using two high-probability inequalities: Eq. C.2, and Eq. C.3 with probability at least $1 - \delta$:

$$\max_{\widetilde{r} \in \mathcal{R}} \frac{1}{K} \sum_{k=1}^K \left( V_{1;P^\star;\widetilde{r}}^{\pi^E} - \widehat{V}_{1;P^\star;\widetilde{r}}^{\pi^E} \right) + \left( \widehat{V}_{1;P^\star;r^k}^{\pi^E} - V_{1;P^\star;r^k}^{\pi^E} \right)$$

$$\leq 2\sqrt{\frac{2}{N} \left( \max_{r \in \mathcal{R}} \mathrm{VaR}_{\pi^E;r} \right) \log(4|\mathcal{R}|/\delta)} + \frac{4\log(4|\mathcal{R}|/\delta)}{3N},$$

$$\max_{\widetilde{r} \in \mathcal{R}} \frac{1}{K} \sum_{k=1}^K \left( \widehat{V}_{1;P^\star;\widetilde{r}}^{\pi^k} - V_{1;P^\star;\widetilde{r}}^{\pi^k} \right) + \left( V_{1;P^\star;r^k}^{\pi^k} - \widehat{V}_{1;P^\star;r^k}^{\pi^k} \right)$$

$$\leq 2\sqrt{\frac{2}{K} \sum_{k=1}^K (\max_{r \in \mathcal{R}} \mathrm{VaR}_{\pi^k;r}) \log(4|\mathcal{R}|/\delta)} + \frac{4\log(4|\mathcal{R}|/\delta)}{3K}.$$

Combining two high-probability inequalities, with probability $1 - \delta$:

$$\mathrm{T1} \leq \epsilon_{\mathrm{opt}}^r + 2\sqrt{\frac{2}{N} \left( \max_{r \in \mathcal{R}} \mathrm{VaR}_{\pi^E;r} \right) \log(4|\mathcal{R}|/\delta)} + \frac{4\log(4|\mathcal{R}|/\delta)}{3N}$$

$$+ 2\sqrt{\frac{2}{K} \sum_{k=1}^K (\max_{r \in \mathcal{R}} \mathrm{VaR}_{\pi^k;r}) \log(4|\mathcal{R}|/\delta)} + \frac{4\log(4|\mathcal{R}|/\delta)}{3K}$$

$$\lesssim \mathcal{O}\left( \frac{1}{\sqrt{K}} + \sqrt{\frac{1}{N} \left( \max_{r \in \mathcal{R}} \mathrm{VaR}_{\pi^E;r} \right) \log(|\mathcal{R}|/\delta)} + \frac{\log(|\mathcal{R}|/\delta)}{N} \right.$$

$$\left. + \sqrt{\frac{1}{K} \sum_{k=1}^K (\max_{r \in \mathcal{R}} \mathrm{VaR}_{\pi^k;r}) \log(|\mathcal{R}|/\delta)} + \frac{\log(|\mathcal{R}|/\delta)}{K} \right).$$

when incorporating FTRL as the no-regret algorithm for reward optimization, providing an $O(1/\sqrt{K})$ bound for $\epsilon_{\mathrm{opt}}^r$.

$\square$

## C.4 Upper Bound for the Policy Error

### C.4.1 Technical Lemmas

In this subsection, we present several technical lemmas that are essential for analyzing the policy error bound. Our analysis is conducted under the following function class:

$$\mathcal{G} = \left\{ (s,a) \mapsto \mathbb{H}^2 \left( P^\star(s,a) \,\|\, P(s,a) \right) : P \in \mathcal{P} \right\},$$

where $\mathbb{H}^2(\cdot\|\cdot)$ denotes the squared Hellinger distance and $\mathcal{P}$ is the model function class. We now state a series of key lemmas used in the subsequent analysis.

**Lemma C.8** (New Eluder Pigeon Lemma, Wang et al. 2024). Let the event $\mathcal{E}$ be:

$$\mathcal{E} : \forall k \in [K], \quad \sum_{i=1}^{k-1} \sum_{h=1}^H \mathbb{H}^2(P^k(s_h^i, a_h^i) \| P^\star(s_h^i, a_h^i)) \leq \eta.$$

Then under event $\mathcal{E}$, there exists a set $\mathcal{K} \in [K]$ such that:

- We have $|\mathcal{K}| \leq 13 \log^2(4\eta KH) \cdot DE_1(\mathcal{G}, \mathcal{S} \times \mathcal{A}, 1/(8\eta KH))$.

- We have

$$\sum_{k \in [K] \setminus \mathcal{K}} \sum_{h=1}^{H} \mathbb{H}^2(P^k(s_h^k, a_h^k) \| P^\star(s_h^k, a_h^k)) \leq DE_1(\mathcal{G}, \mathcal{S} \times \mathcal{A}, 1/KH)(2 + 7\eta \log(KH)) + 1.$$

**Lemma C.9** (Bounded Hellinger Distance, Wang et al. (2024)). By standard MLE generalization bound and the realizability assumption, by running Algorithm 1, it exists:

- $P^\star \in \widehat{\mathcal{P}}^k$.

- With probability $1 - \delta$, $\sum_{i=1}^{k-1} \sum_{h=1}^{H} \mathbb{H}^2(P^k(s_h^i, a_h^i) \| P^\star(s_h^i, a_h^i)) \leq 22 \log(K|\mathcal{P}|/\delta)$.

**Lemma C.10** (Bounded Sum of Mean Value Differences, Wang et al. 2024). Given Lemma C.9 happens under high probability, we have:

$$\sum_{k \in [K] \setminus \mathcal{K}} \sum_{h=0}^{H-1} \left| \mathbb{E}_{s' \sim P^k(s_h^k, a_h^k)} V_{h+1; P^k; r^k}^{\pi^k}(s') - \mathbb{E}_{s' \sim P^\star(s_h^k, a_h^k)} V_{h+1; P^k; r^k}^{\pi^k}(s') \right|$$

$$\lesssim \sqrt{\sum_{k \in [K] \setminus \mathcal{K}} \sum_{h=0}^{H-1} \left[ (\mathbb{V}_{P^\star} V_{h+1; P^k; r^k}^{\pi^k})(s_h^k, a_h^k) \right] \cdot DE_1(\mathcal{G}, \mathcal{S} \times \mathcal{A}, 1/KH) \log(K|\mathcal{P}|/\delta) \log(KH)}$$

$$+ DE_1(\mathcal{G}, \mathcal{S} \times \mathcal{A}, 1/KH) \log(K|\mathcal{P}|/\delta) \log(KH).$$

**Lemma C.11** (Variance Conversion Lemma, Wang et al. 2024). Let $d_E = DE_1(\mathcal{G}, \mathcal{S} \times \mathcal{A}, 1/KH)$. If the following events happen with high probability:

- $\forall k \in [K]: P^\star \in \widehat{\mathcal{P}}$, and $\sum_{i=1}^{k-1} \sum_{h=1}^{H} \mathbb{H}^2(P^k(s_h^i, a_h^i) \| P^\star(s_h^i, a_h^i)) \leq 22 \log(K|\mathcal{P}|/\delta)$.

- $\forall m \in [0, \lceil \log_2(\frac{KH}{G}) \rceil]: C_m \lesssim 2^m G + \sqrt{\log(1/\delta) \cdot C_{m+1}} + \log(1/\delta)$. $G$ and $C_m$ are defined as:

$$G := \sqrt{\sum_{k \in [K] \setminus \mathcal{K}} \sum_{h=0}^{H-1} \left[ (\mathbb{V}_{P^\star} V_{h+1; P^k; r^k}^{\pi^k})(s_h^k, a_h^k) \right] \cdot d_E \log(K|\mathcal{P}|/\delta) \log(KH)}$$

$$+ d_E \log(K|\mathcal{P}|/\delta) \log(KH),$$

$$C_m := \sum_{k \in [K] \setminus \mathcal{K}} \sum_{h=0}^{H-1} \left[ \left( \mathbb{V}_{P^\star} (V_{h+1; P^k; r^k}^{\pi^k})^{2^m} \right)(s_h^k, a_h^k) \right].$$

we have:

$$\sum_{k \in [K] \setminus \mathcal{K}} \sum_{h=0}^{H-1} \left[ \left( \mathbb{V}_{P^\star} V_{h+1; P^k; r^k}^{\pi^k} \right)(s_h^k, a_h^k) \right]$$

$$\leq O\left( \sum_{k \in [K] \setminus \mathcal{K}} \sum_{h=0}^{H-1} \left[ \left( \mathbb{V}_{P^\star} V_{h+1; P^\star; r^k}^{\pi^k} \right)(s_h^k, a_h^k) \right] + d_E \log(K|\mathcal{P}|/\delta) \log(KH) \right).$$

**Lemma C.12** (Generalization Bound of MLE for Infinite Model Classes, Wang et al. 2024). Let $\mathcal{X}$ be the context (feature) space and $\mathcal{Y}$ be the label space. Suppose we are given a dataset $\mathcal{D} = \{(x_i, y_i)\}_{i \in [n]}$ generated from a martingale process, where for each $i = 1, 2, \ldots, n$, the input $x_i$ is drawn from a distribution $\mathcal{D}_i(x_{1:i-1}, y_{1:i-1})$, and the output $y_i$ is sampled from the conditional distribution $p(\cdot \mid x_i)$.

Let the true data-generating distribution be denoted as $f^\star(x, y) = p(y \mid x)$, and assume the model class $\mathcal{F} : \mathcal{X} \times \mathcal{Y} \to \Delta(\mathbb{R})$ is realizable, i.e., $f^\star \in \mathcal{F}$. Let $\mathcal{F}$ be potentially infinite. Fix any $\delta \in (0, 1)$, and define

$$\beta = \log\left( \frac{\mathcal{N}_{[]}\left( (n|\mathcal{Y}|)^{-1}, \mathcal{F}, \| \cdot \|_\infty \right)}{\delta} \right),$$

where $\mathcal{N}_{[]}\left((n|\mathcal{Y}|)^{-1}, \mathcal{F}, \|\cdot\|_\infty\right)$ denotes the bracketing number as defined in Definition 3.3.

Let the version space be defined as

$$\widehat{\mathcal{F}} = \left\{ f \in \mathcal{F} : \sum_{i=1}^{n} \log f(x_i, y_i) \geq \max_{\widetilde{f} \in \mathcal{F}} \sum_{i=1}^{n} \log \widetilde{f}(x_i, y_i) - 7\beta \right\}.$$

Then, with probability at least $1 - \delta$, the following statements hold:

(1) The true distribution lies in the version space, i.e., $f^\star \in \widehat{\mathcal{F}}$.

(2) Every function in the version space is close to the true distribution in Hellinger distance:

$$\sum_{i=1}^{n} \mathbb{E}_{x \sim \mathcal{D}_i} \left[ \mathbb{H}^2 \left( f(x, \cdot) \,\|\, f^\star(x, \cdot) \right) \right] \leq 28\beta, \quad \forall f \in \widehat{\mathcal{F}}.$$

**Lemma C.13.** For any $\delta \in (0, 1)$, by applying Bellman equation recursively and with Lemma C.10, with probability $1 - \delta$:

$$\sum_{k \in [K] \setminus \mathcal{K}} \left( V_{1;P^k;r^k}^{\pi^k} - \sum_{h=0}^{H-1} r^k(s_h^k, a_h^k) \right)$$

$$\lesssim \sqrt{\sum_{k \in [K] \setminus \mathcal{K}} \sum_{h=0}^{H-1} (\mathbb{V}_{P^\star} V_{h+1;P^k;r^k}^{\pi^k})(s_h^k, a_h^k) \log(1/\delta) + \log(1/\delta)}$$

$$+ \sqrt{\sum_{k \in [K] \setminus \mathcal{K}} \sum_{h=0}^{H-1} \left[ (\mathbb{V}_{P^\star} V_{h+1;P^k;r^k}^{\pi^k})(s_h^k, a_h^k) \right] \cdot DE_1(\mathcal{G}, \mathcal{S} \times \mathcal{A}, 1/KH) \log(K|\mathcal{P}|/\delta) \log(KH)}$$

$$+ DE_1(\mathcal{G}, \mathcal{S} \times \mathcal{A}, 1/KH) \log(K|\mathcal{P}|/\delta) \log(KH),$$

where $(\mathbb{V}_{P^\star}[V_{h+1;P^k;r^k}^{\pi^k}])(s_h^k, a_h^k)$ is the variance of the value function with respect to the true transition model $P^\star$, and $DE_1(\mathcal{G}, \mathcal{S} \times \mathcal{A}, 1/KH)$ is the Eluder dimension.

*Proof of Lemma C.13.*

$$\sum_{k \in [K] \setminus \mathcal{K}} \left( V_{1;P^k;r^k}^{\pi^k} - \sum_{h=0}^{H-1} r^k(s_h^k, a_h^k) \right)$$

$$\leq \sum_{k \in [K] \setminus \mathcal{K}} \sum_{h=0}^{H-1} \left[ \mathbb{E}_{s' \sim P^\star(s_h^k, a_h^k)} V_{h+1;P^k;r^k}^{\pi^k}(s') - V_{h+1;P^k;r^k}^{\pi^k}(s_{h+1}^k) \right]$$

$$+ \sum_{k \in [K] \setminus \mathcal{K}} \sum_{h=0}^{H-1} \left| \mathbb{E}_{s' \sim P^k(s_h^k, a_h^k)} V_{h+1;P^k;r^k}^{\pi^k}(s') - \mathbb{E}_{s' \sim P^\star(s_h^k, a_h^k)} V_{h+1;P^k;r^k}^{\pi^k}(s') \right|$$

$$\leq \sqrt{2 \sum_{k \in [K] \setminus \mathcal{K}} \sum_{h=0}^{H-1} (\mathbb{V}_{P^\star} V_{h+1;P^k;r^k}^{\pi^k})(s_h^k, a_h^k) \log(1/\delta) + \frac{2}{3} \log(1/\delta)}$$

$$+ \sum_{k \in [K] \setminus \mathcal{K}} \sum_{h=0}^{H-1} \left| \mathbb{E}_{s' \sim P^k(s_h^k, a_h^k)} V_{h+1;P^k;r^k}^{\pi^k}(s') - \mathbb{E}_{s' \sim P^\star(s_h^k, a_h^k)} V_{h+1;P^k;r^k}^{\pi^k}(s') \right|$$

$$\lesssim \sqrt{\sum_{k \in [K] \setminus \mathcal{K}} \sum_{h=0}^{H-1} (\mathbb{V}_{P^\star} V_{h+1;P^k;r^k}^{\pi^k})(s_h^k, a_h^k) \log(1/\delta) + \log(1/\delta)}$$

$$+ \sqrt{\sum_{k \in [K] \setminus \mathcal{K}} \sum_{h=0}^{H-1} \left[ (\mathbb{V}_{P^\star} V_{h+1;P^k;r^k}^{\pi^k})(s_h^k, a_h^k) \right] \cdot DE_1(\mathcal{G}, \mathcal{S} \times \mathcal{A}, 1/KH) \log(K|\mathcal{P}|/\delta) \log(KH)}$$

$$+ DE_1(\mathcal{G}, \mathcal{S} \times \mathcal{A}, 1/KH) \log(K|\mathcal{P}|/\delta) \log(KH).$$

$\square$

### C.4.2 PROOF OF LEMMA C.3

*Proof of Lemma C.3.* Our proof for bounding the policy error generally follows the proof steps in (Wang et al., 2024) but with $r^k$ instead of a fixed $r^\star$. By Line 6 of Algorithm 1, the policy error can be bounded by the optimism guarantee, and with $\mathcal{K} \subseteq [K]$ and Lemma C.8:

$$\text{T2} = \frac{1}{K} \sum_{k=1}^{K} \left( V_{1;P^\star;r^k}^{\pi^E} - V_{1;P^\star;r^k}^{\pi^k} \right)$$

$$= \frac{1}{K} \sum_{k=1}^{K} V_{1;P^\star;r^k}^{\pi^E} - \frac{1}{K} \sum_{k=1}^{K} \sum_{h=1}^{H} r^k(s_h^k, a_h^k) + \frac{1}{K} \sum_{k=1}^{K} \sum_{h=1}^{H} r^k(s_h^k, a_h^k) - \frac{1}{K} \sum_{k=1}^{K} V_{1;P^\star;r^k}^{\pi^k}$$

$$\lesssim \frac{1}{K} |\mathcal{K}| + \frac{1}{K} \sum_{k \in [K] \setminus \mathcal{K}} \left( V_{1;P^k;r^k}^{\pi^k} - \sum_{h=0}^{H-1} r^k(s_h^k, a_h^k) \right) + \sqrt{\frac{2 \sum_{k=1}^{K} \text{VaR}_{\pi^k, r^k} \log(1/\delta)}{K}} + \frac{2 \log(1/\delta)}{3K}$$

$$\lesssim \frac{1}{K} \log^2(4\eta KH) \cdot DE_1(\mathcal{G}, \mathcal{S} \times \mathcal{A}, 1/(8\eta KH)) + \frac{1}{K} \sum_{k \in [K] \setminus \mathcal{K}} \left( V_{1;P^k;r^k}^{\pi^k} - \sum_{h=0}^{H-1} r^k(s_h^k, a_h^k) \right)$$

$$+ \sqrt{\frac{2 \sum_{k=1}^{K} \text{VaR}_{\pi^k, r^k} \log(1/\delta)}{K}} + \frac{2 \log(1/\delta)}{3K}.$$

By Lemma C.9 and probability at least $1 - \delta$, set $\eta := 22 \log(5K|\mathcal{P}|/\delta)$, we have:

$$\text{T2} \lesssim \frac{1}{K} \log^2(\log(K|\mathcal{P}|/\delta)KH) \cdot DE_1(\mathcal{G}, \mathcal{S} \times \mathcal{A}, 1/(\log(K|\mathcal{P}|/\delta)KH))$$

$$+ \frac{1}{K} \sum_{k \in [K] \setminus \mathcal{K}} \left( V_{1;P^k;r^k}^{\pi^k} - \sum_{h=0}^{H-1} r^k(s_h^k, a_h^k) \right) + \sqrt{\frac{\sum_{k=1}^{K} \text{VaR}_{\pi^k, r^k} \log(1/\delta)}{K}} + \frac{\log(1/\delta)}{K}.$$

$$\text{(C.4)}$$

We now bound the second term on the RHS of Eq. C.4. By Lemma C.13, this term admits a variance-aware upper bound that depends on the Eluder dimension. Substituting this result, together with Lemma C.11, back into Eq. C.4 yields:

$$\text{T2} \lesssim \frac{1}{K} \log^2(\log(K|\mathcal{P}|/\delta)KH) \cdot DE_1(\mathcal{G}, \mathcal{S} \times \mathcal{A}, 1/(\log(K|\mathcal{P}|/\delta)KH))$$

$$+ \sqrt{\frac{\sum_{k=1}^{K} \text{VaR}_{\pi^k, r^k} \log(1/\delta)}{K}} + \frac{1}{K} \sqrt{\sum_{k \in [K] \setminus \mathcal{K}} \sum_{h=0}^{H-1} (\mathbb{V}_{P^\star} V_{h+1;P^k;r^k}^{\pi^k})(s_h^k, a_h^k) \log(1/\delta)}$$

$$+ \frac{1}{K} \sqrt{\sum_{k \in [K] \setminus \mathcal{K}} \sum_{h=0}^{H-1} \left[ (\mathbb{V}_{P^\star} V_{h+1;P^k;r^k}^{\pi^k})(s_h^k, a_h^k) \right] \cdot d_E \log(K|\mathcal{P}|/\delta) \log(KH)} + \frac{\log(1/\delta)}{K}$$

$$\lesssim \frac{1}{K} \log^2(\log(K|\mathcal{P}|/\delta)KH) \cdot DE_1(\mathcal{G}, \mathcal{S} \times \mathcal{A}, 1/(\log(K|\mathcal{P}|/\delta)KH))$$

$$+ \frac{1}{K} \sqrt{\left( \sum_{k \in [K] \setminus \mathcal{K}} \sum_{h=0}^{H-1} \left[ \left( \mathbb{V}_{P^\star} V_{h+1;P^\star;r^k}^{\pi^k} \right)(s_h^k, a_h^k) \right] + d_E \log(K|\mathcal{P}|/\delta) \log(KH) \right) \log(1/\delta)}$$

$$+ \frac{1}{K} \sqrt{\sum_{k \in [K] \setminus \mathcal{K}} \sum_{h=0}^{H-1} \left[ (\mathbb{V}_{P^\star} V_{h+1;P^k;r^k}^{\pi^k})(s_h^k, a_h^k) \right] \cdot d_E \log(K|\mathcal{P}|/\delta) \log(KH)}$$

$$+ \frac{\log(1/\delta)}{K} + \sqrt{\frac{\sum_{k=1}^{K} \text{VaR}_{\pi^k, r^k} \log(1/\delta)}{K}}$$

$$\lesssim \frac{1}{K} \log^2(\log(K|\mathcal{P}|/\delta)KH) \cdot DE_1(\mathcal{G}, \mathcal{S} \times \mathcal{A}, 1/(\log(K|\mathcal{P}|/\delta)KH))$$

$$+ \frac{1}{K} \sqrt{\left( \sum_{k \in [K] \setminus \mathcal{K}} \sum_{h=0}^{H-1} \left[ \left( \mathbb{V}_{P^\star} V_{h+1;P^\star;r^k}^{\pi^k} \right)(s_h^k, a_h^k) \right] + d_E \log(K|\mathcal{P}|/\delta) \log(KH) \right) \log(1/\delta)}$$

$$+ \frac{1}{K} \sqrt{\left( \sum_{k \in [K] \setminus \mathcal{K}} \sum_{h=0}^{H-1} \left[ \left( \mathbb{V}_{P^\star} V_{h+1;P^\star;r^k}^{\pi^k} \right)(s_h^k, a_h^k) \right] + d_E \log(K|\mathcal{P}|/\delta) \log(KH) \right) \log(1/\delta)}$$

$$\times \sqrt{d_E \log(K|\mathcal{P}|/\delta) \log(KH)} + \frac{\log(1/\delta)}{K} + \sqrt{\frac{\sum_{k=1}^{K} \mathrm{VaR}_{\pi^k, r^k} \log(1/\delta)}{K}}$$

$$\le \mathcal{O}\left( \frac{1}{K} \sqrt{\sum_{k=1}^{K} \mathrm{VaR}_{\pi^k, r^k} \cdot d_E \cdot \log(K|\mathcal{P}|/\delta) \log(KH)} \right.$$

$$\left. + \frac{1}{K} d_E \cdot \log(K|\mathcal{P}|/\delta) \log(KH) \right)$$

$$\le \mathcal{O}\left( \frac{1}{K} \sqrt{\sum_{k=1}^{K} \max_{r \in \mathcal{R}} \mathrm{VaR}_{\pi^k, r} \cdot d_E \cdot \log(K|\mathcal{P}|/\delta) \log(KH)} \right.$$

$$\left. + \frac{1}{K} d_E \cdot \log(K|\mathcal{P}|/\delta) \log(KH) \right),$$

where $d_E = DE_1(\mathcal{G}, \mathcal{S} \times \mathcal{A}, 1/KH)$. Replacing $\delta$ with $\delta/(5KH)$ concludes the proof. $\qquad\square$

### C.5 UPPER BOUNDS UNDER GENERAL FUNCTION APPROXIMATION

In this section, we extend the our upper bound result to the general function approximation, providing the proof for Theorem 5.1. The proof steps generally follow the proof steps for Theorem C.1, but with covering number analysis for the reward function class $\mathcal{R}$ and bracketing number analysis for the model function class $\mathcal{P}$. The covering number is defined in Definition 3.2 and the bracketing number is defined in Definition 3.3. We present the lemmas for the upper bounds of reward and policy error under general function approximation as follows:

**Lemma C.14** (Reward Error Upper Bound with Infinite Reward Class). For any $\delta \in (0, 1)$, using FTRL (Shalev-Shwartz & Singer, 2007) as the no-regret algorithm, given an infinite reward function class $\mathcal{R}$, with probability $1 - \delta$, the average regret for reward optimization is bounded:

$$\max_{\widetilde{r} \in \mathcal{R}} \frac{1}{K} \sum_{k=1}^{K} V_{1;P^\star;\widetilde{r}}^{\pi^E} - V_{1;P^\star;\widetilde{r}}^{\pi^k} - (V_{1;P^\star;r^k}^{\pi^E} - V_{1;P^\star;r^k}^{\pi^k})$$

$$\lesssim \mathcal{O}\left( \frac{1}{\sqrt{K}} + \frac{\log(\mathcal{N}_{\mathcal{R}}/\delta)}{K} + \frac{\log(\mathcal{N}_{\mathcal{R}}/\delta)}{N} + \sqrt{\frac{1}{K} \sum_{k=1}^{K} \max_{r \in \mathcal{R}} \mathrm{VaR}_{\pi^k; r} \log(\mathcal{N}_{\mathcal{R}}/\delta)} \right.$$

$$\left. + \sqrt{\frac{1}{N} \max_{r \in \mathcal{R}} \mathrm{VaR}_{\pi^E; r} \log(\mathcal{N}_{\mathcal{R}}/\delta)} \right)$$

where $\mathrm{VaR}_{\pi; r}$ denotes the variance of accumulative rewards over a horizon $H$ under $\pi$ and reward function $r$. $\mathcal{N}_{\mathcal{R}} = \max\{\mathcal{N}_{1/KH}(\mathcal{R}), \mathcal{N}_{1/NH}(\mathcal{R})\}$ is the covering number for reward function class $\mathcal{R}$.

**Lemma C.15** (Policy Error Upper Bound). For any $\delta \in (0, 1)$, with an infinite model class $\mathcal{P}$, let $\beta = 7 \log(K \mathcal{N}_{\mathcal{P}}/\delta)$. With probability $1 - \delta$, the average regret of the policy optimization is bounded:

$$\frac{1}{K} \sum_{k=1}^{K} \left( V_{1;P^\star;r^k}^{\pi^E} - V_{1;P^\star;r^k}^{\pi^k} \right)$$

$$\lesssim \mathcal{O}\left(\frac{1}{K}\sqrt{\sum_{k=1}^{K}\max_{r\in\mathcal{R}}\mathrm{VaR}_{\pi^k,r}\cdot d_E\log(KH\mathcal{N}_{\mathcal{P}}/\delta)\log(KH)}\right.$$

$$\left.+\frac{1}{K}d_E\log(KH\mathcal{N}_{\mathcal{P}}/\delta)\log(KH)\right),$$

where $d_E = DE_1(\mathcal{G}, \mathcal{S}\times\mathcal{A}, 1/KH)$ is the Eluder dimension, $\mathrm{VaR}_{\pi,k} = \max_{r\in\mathcal{R}}\mathrm{VaR}_{\pi^k,r}$ is the maximum variance for $\pi^k$ and $\mathcal{N}_{\mathcal{P}} = \mathcal{N}_{[]}\left((KH|\mathcal{S}|)^{-1}, \mathcal{P}, \|\cdot\|_{\infty}\right)$ is the bracketing number.

By combining Lemmas C.14 and C.15, we complete the proof of Theorem 5.1 under general function approximation.

### C.5.1 UPPER BOUND FOR THE REWARD ERROR

*Proof of Lemma C.14.* We first revisit the bound on the reward estimation error under a general function approximation setting. Our goal is to refine the upper bound established in Lemma C.2 by expressing it in terms of the covering number $\mathcal{N}_{\epsilon}(\mathcal{R})$ rather than the cardinality $|\mathcal{R}|$. From the regret decomposition used in the proof of Lemma C.2, the reward optimization error remains unchanged, as it depends solely on $\epsilon_{\mathrm{opt}}^r$ and is independent of the reward function class. Therefore, the focus shifts to revisiting the estimation error:

$$\mathrm{T4} = \max_{\widetilde{r}\in\mathcal{R}}\frac{1}{K}\sum_{k=1}^{K}\left(V_{1;P^\star;\widetilde{r}}^{\pi^E} - \widehat{V}_{1;P^\star;\widetilde{r}}^{\pi^E}\right) + \left(\widehat{V}_{1;P^\star;r^k}^{\pi^E} - V_{1;P^\star;r^k}^{\pi^E}\right)$$

$$+ \left(\widehat{V}_{1;P^\star;\widetilde{r}}^{\pi^k} - V_{1;P^\star;\widetilde{r}}^{\pi^k}\right) + \left(V_{1;P^\star;r^k}^{\pi^k} - \widehat{V}_{1;P^\star;r^k}^{\pi^k}\right)$$

By applying Bernstein's inequality, we obtain that, for any fixed reward function $r$, the estimation error is bounded with probability at least $1 - \delta$:

$$\left|\widehat{V}_{1;P^\star;r}^{\pi^E} - V_{1;P^\star;r}^{\pi^E}\right| = \left|\frac{1}{N}\sum_{\tau\in\mathcal{D}_E}\sum_{h=1}^{H}r(s_h(\tau), a_h(\tau)) - \mathbb{E}_{\pi^E}\left[\sum_{h=1}^{H}r(s_h, a_h)\right]\right|$$

$$\leq \sqrt{\frac{2\sum_{\tau\in\mathcal{D}_E}\mathbb{V}_{\pi^E}[\sum_{h=1}^{H}r(s_h(\tau), a_h(\tau))]\log(2/\delta)}{N}} + \frac{2\log(2/\delta)}{3N}$$

$$= \sqrt{\frac{2}{N}\mathrm{VaR}_{\pi^E;r}\log(2/\delta)} + \frac{2\log(2/\delta)}{3N} \tag{C.5}$$

Unlike the analysis for finite function classes, we cannot directly apply a union bound over the cardinality $|\mathcal{R}|$ since the reward class $\mathcal{R}$ is now infinite. To address this, we construct a $\rho$-cover of the reward space, denoted $(\mathcal{R})\rho$. Then, for all $\widehat{r}\in(\mathcal{R})\rho$, we can apply the union bound:

$$\left|\widehat{V}_{1;P^\star;\widehat{r}}^{\pi^E} - V_{1;P^\star;\widehat{r}}^{\pi^E}\right| \leq \sqrt{\frac{2}{N}\mathrm{VaR}_{\pi^E;\widehat{r}}\log(2|(\mathcal{R})_\rho|/\delta)} + \frac{2\log(2|(\mathcal{R})_\rho|/\delta)}{3N}$$

According to the definition of $\rho$-cover, for any $r\in\mathcal{R}$, there exists $\widehat{r}\in(\mathcal{R})_\rho$ that satisfies $\max_{(s,a)\in\mathcal{S}\times\mathcal{A}}|\widehat{r}(s,a) - r(s,a)| \leq \rho$. Therefore, it exists:

$$|\widehat{V}_{1;P^\star;r}^{\pi^E} - \widehat{V}_{1;P^\star;\widehat{r}}^{\pi^E}| \leq \frac{1}{N}\sum_{\tau\in\mathcal{D}_E}\sum_{h=1}^{H}\left|r(s_h(\tau), a_h(\tau)) - \widehat{r}(s_h(\tau), a_h(\tau))\right| \leq H\rho,$$

$$|V_{1;P^\star;r}^{\pi^E} - V_{1;P^\star;\widehat{r}}^{\pi^E}| \leq \mathbb{E}_{\pi^E}\left[\sum_{h=1}^{H}\left|r(s_h(\tau), a_h(\tau)) - \widehat{r}(s_h(\tau), a_h(\tau))\right|\right] \leq H\rho.$$

Therefore, the value estimation difference of any reward function $r$ can be upper bounded by that of its representative $\widehat{r}$ within the $\rho$-cover of the reward function class:

$$\left|\widehat{V}_{1;P^\star;r}^{\pi^E} - V_{1;P^\star;r}^{\pi^E}\right| \leq \left|\widehat{V}_{1;P^\star;\widehat{r}}^{\pi^E} - V_{1;P^\star;\widehat{r}}^{\pi^E}\right| + 2H\rho$$

$$\leq \sqrt{\frac{2}{N} \mathrm{VaR}_{\pi^E;\widehat{r}} \log(2|(\mathcal{R})_\rho|/\delta)} + \frac{2\log(2|(\mathcal{R})_\rho|/\delta)}{3N} + 2H\rho \qquad \text{(C.6)}$$

For $\mathrm{VaR}_{\pi^E,\widehat{r}}$, we have:

$$\mathrm{VaR}_{\pi^E,\widehat{r}} = \mathbb{E}_{\tau \sim \mathcal{D}_E}\Big(\sum_{h=1}^{H} \widehat{r}(s_h(\tau), a_h(\tau)) - \mathbb{E}_{\pi^E} \sum_{h=1}^{H} \widehat{r}(s_h, a_h)\Big)^2$$

$$\leq \mathbb{E}_{\tau \sim \mathcal{D}_E}\Big(\sum_{h=1}^{H} r(s_h(\tau), a_h(\tau)) - \mathbb{E}_{\pi^E} \sum_{h=1}^{H} r(s_h, a_h)\Big)^2 + \rho^2 H^2$$

$$= \mathrm{VaR}_{\pi^E,r} + \rho^2 H^2$$

We can further bound Eq. C.6 with $\mathrm{VaR}_{\pi^E,r}$:

$$\left|\widehat{V}_{1;P^\star;r}^{\pi^E} - V_{1;P^\star;r}^{\pi^E}\right| \leq \sqrt{\frac{2}{N}(\mathrm{VaR}_{\pi^E;r} + \rho^2 H^2)\log(2|(\mathcal{R})_\rho|)/\delta)} + \frac{2\log(2(|\mathcal{R}|)_\rho)/\delta)}{3N} + 2H\rho$$

$$\text{(C.7)}$$

By plugging in $\widetilde{r}$ and $r^k$ and selecting $\rho := 1/NH$:

$$\max_{\widetilde{r} \in \mathcal{R}} \frac{1}{K} \sum_{k=1}^{K} \left(V_{1;P^\star;\widetilde{r}}^{\pi^E} - \widehat{V}_{1;P^\star;\widetilde{r}}^{\pi^E}\right) + \left(\widehat{V}_{1;P^\star;r^k}^{\pi^E} - V_{1;P^\star;r^k}^{\pi^E}\right)$$

$$\leq 2\sqrt{\frac{2}{N}\Big(\max_{r \in \mathcal{R}} \mathrm{VaR}_{\pi^E;r} + \rho^2 H^2\Big)\log(2|(\mathcal{R})_\rho|)/\delta)} + \frac{4\log(2|(\mathcal{R})_\rho|/\delta)}{3N} + 4H\rho$$

$$= 2\sqrt{\frac{2}{N}\Big(\max_{r \in \mathcal{R}} \mathrm{VaR}_{\pi^E;r} + \frac{1}{N^2}\Big)\log(2|(\mathcal{R})_{1/NH}|/\delta)} + \frac{4\log(2|(\mathcal{R})_{1/NH}|/\delta)}{3N} + \frac{4}{N} \qquad \text{(C.8)}$$

We then focus on the value difference with respect to $\pi^k$ for any $r \in \mathcal{R}$ using Azuma-Bernstein's inequality, with probability at least $1 - \delta$:

$$\frac{1}{K} \sum_{k=1}^{K} \left|\widehat{V}_{1;P^\star;r}^{\pi^k} - V_{1;P^\star;r}^{\pi^k}\right| = \frac{1}{K} \sum_{k=1}^{K} \left|\sum_{h=1}^{H} r(s_h, a_h) - \mathbb{E}_{\pi^k}\Big[\sum_{h=1}^{H} r(s_h, a_h)\Big]\right|$$

$$\leq \sqrt{\frac{2}{K} \sum_{k=1}^{K} \mathbb{V}_{\pi^k}\Big[\sum_{h=1}^{H} r(s_h, a_h)\Big] \log(1/\delta)} + \frac{2\log(1/\delta)}{3K}$$

$$= \sqrt{\frac{2}{K} \sum_{k=1}^{K} \mathrm{VaR}_{\pi^k;r} \log(1/\delta)} + \frac{2\log(1/\delta)}{3K}$$

Following the analysis for terms related to $\pi^E$, given a $\rho$-cover $(\mathcal{R})_\rho$, for all $\widehat{r} \in (\mathcal{R})_\rho$, by union bound:

$$\frac{1}{K} \sum_{k=1}^{K} \left|\widehat{V}_{1;P^\star;\widehat{r}}^{\pi^k} - V_{1;P^\star;\widehat{r}}^{\pi^k}\right| \leq \sqrt{\frac{2}{K} \sum_{k=1}^{K} \mathrm{VaR}_{\pi^k;\widehat{r}} \log(2|(\mathcal{R})_\rho|/\delta)} + \frac{2\log(2|(\mathcal{R})_\rho|/\delta)}{3K}$$

By the definition of $\rho$-cover, it exists:

$$|\widehat{V}_{1;P^\star;r}^{\pi^k} - \widehat{V}_{1;P^\star;\widehat{r}}^{\pi^k}| \leq \mathbb{E}_{\pi^k}\left[\sum_{h=1}^{H} \left|r(s_h^k, a_h^k) - \widehat{r}(s_h^k, a_h^k)\right|\right] \leq H\rho,$$

$$|V_{1;P^\star;r}^{\pi^k} - V_{1;P^\star;\widehat{r}}^{\pi^k}| \leq \mathbb{E}_{\pi^k}\left[\sum_{h=1}^{H} \left|r(s_h^k, a_h^k) - \widehat{r}(s_h^k, a_h^k)\right|\right] \leq H\rho.$$

Therefore, the average value estimation difference of any reward function $r$ can be upper bounded by that of its representative $\widehat{r}$ within the $\rho$-cover of the reward function class:

$$\frac{1}{K}\sum_{k=1}^{K}\left|\widehat{V}_{1;P^\star;r}^{\pi^k} - V_{1;P^\star;r}^{\pi^k}\right| \le \frac{1}{K}\sum_{k=1}^{K}\left|\widehat{V}_{1;P^\star;\widehat{r}}^{\pi^k} - V_{1;P^\star;\widehat{r}}^{\pi^k}\right| + 2H\rho$$

$$\le \sqrt{\frac{2}{K}\sum_{k=1}^{K}\mathrm{VaR}_{\pi^k;\widehat{r}}\log(2|(\mathcal{R})_\rho|/\delta)} + \frac{2\log(2|(\mathcal{R})_\rho|/\delta)}{3K} + 2H\rho$$

Similar to the analysis for the terms related to $\pi^E$, we also have the variance bound:

$$\mathrm{VaR}_{\pi^k,\widehat{r}} \le \mathrm{VaR}_{\pi^k,r} + \rho^2 H^2$$

Therefore, we can finally upper bound the rest of the terms in the estimation error:

$$\frac{1}{K}\sum_{k=1}^{K}\left|\widehat{V}_{1;P^\star;r}^{\pi^k} - V_{1;P^\star;r}^{\pi^k}\right|$$

$$\le \sqrt{\frac{2}{K}\sum_{k=1}^{K}(\mathrm{VaR}_{\pi^k;r} + \rho^2 H^2)\log(2|(\mathcal{R})_\rho|/\delta)} + \frac{2\log(2|(\mathcal{R})_\rho|/\delta)}{3K} + 2H\rho$$

Thus, the last two terms related to $\pi^k$ is upper bounded, selecting $\rho := 1/KH$, with probability at least $1-\delta$:

$$\max_{\widetilde{r}\in\mathcal{R}}\ \frac{1}{K}\sum_{k=1}^{K}\left(\widehat{V}_{1;P^\star;\widetilde{r}}^{\pi^k} - V_{1;P^\star;\widetilde{r}}^{\pi^k}\right) + \left(V_{1;P^\star;r^k}^{\pi^k} - \widehat{V}_{1;P^\star;r^k}^{\pi^k}\right)$$

$$\le 2\sqrt{\frac{2}{K}\sum_{k=1}^{K}(\max_{r\in\mathcal{R}}\mathrm{VaR}_{\pi^k;r} + \rho^2 H^2)\log(2|(\mathcal{R})_\rho|/\delta)} + \frac{4\log(2|(\mathcal{R})_\rho|/\delta)}{3K} + 4H\rho \tag{C.9}$$

$$= 2\sqrt{\frac{2}{K}\sum_{k=1}^{K}(\max_{r\in\mathcal{R}}\mathrm{VaR}_{\pi^k;r} + \frac{1}{K^2})\log(2|(\mathcal{R})_{1/KH}|/\delta)} + \frac{4\log(2|(\mathcal{R})_{1/KH}|/\delta)}{3K} + \frac{4}{K} \tag{C.10}$$

Then finally by union bound, using two high-probability inequalities: Eq. C.8, and Eq. C.9 with probability at least $1-\delta$:

$$\max_{\widetilde{r}\in\mathcal{R}}\ \frac{1}{K}\sum_{k=1}^{K}\left(V_{1;P^\star;\widetilde{r}}^{\pi^E} - \widehat{V}_{1;P^\star;\widetilde{r}}^{\pi^E}\right) + \left(\widehat{V}_{1;P^\star;r^k}^{\pi^E} - V_{1;P^\star;r^k}^{\pi^E}\right)$$

$$\le 2\sqrt{\frac{2}{N}\left(\max_{r\in\mathcal{R}}\mathrm{VaR}_{\pi^E;r} + \frac{1}{N^2}\right)\log(4|(\mathcal{R})_{1/NH}|/\delta)} + \frac{4\log(4|(\mathcal{R})_{1/NH}|/\delta)}{3N} + \frac{4}{N},$$

$$\frac{1}{K}\sum_{k=1}^{K}\left(\widehat{V}_{1;P^\star;\widetilde{r}}^{\pi^k} - V_{1;P^\star;\widetilde{r}}^{\pi^k}\right) + \left(V_{1;P^\star;r^k}^{\pi^k} - \widehat{V}_{1;P^\star;r^k}^{\pi^k}\right)$$

$$\le 2\sqrt{\frac{2}{K}\sum_{k=1}^{K}(\max_{r\in\mathcal{R}}\mathrm{VaR}_{\pi^k;r} + \frac{1}{K^2})\log(4|(\mathcal{R})_{1/KH}|/\delta)} + \frac{4\log(4|(\mathcal{R})_{1/KH}|/\delta)}{3K} + \frac{4}{K}$$

Combining two high-probability inequalities, let $\rho_K := 1/KH$ and $\rho_N := 1/NH$, with probability $1-\delta$:

$$\text{T4} \le 2\sqrt{\frac{2}{N}\left(\max_{r\in\mathcal{R}}\mathrm{VaR}_{\pi^E;r} + \frac{1}{N^2}\right)\log(4|(\mathcal{R})_{\rho_N}|/\delta)} + \frac{4\log(4|(\mathcal{R})_{\rho_N}|/\delta)}{3N} + \frac{4}{N}$$

$$+ 2\sqrt{\frac{2}{K}\sum_{k=1}^{K}(\max_{r\in\mathcal{R}}\mathrm{VaR}_{\pi^k;r} + \frac{1}{K^2})\log(4|(\mathcal{R})_{\rho_K}|/\delta)} + \frac{4\log(4|(\mathcal{R})_{\rho_K}|/\delta)}{3K} + \frac{4}{K}$$

$$\lesssim \mathcal{O}\left( \frac{\log(|(\mathcal{R})_{\rho_K}|/\delta)}{K} + \frac{\log(|(\mathcal{R})_{\rho_N}|/\delta)}{N} + \sqrt{\frac{1}{K} \sum_{k=1}^{K} (\max_{r \in \mathcal{R}} \mathrm{VaR}_{\pi^k;r} + \frac{1}{K^2}) \log(|(\mathcal{R})_{\rho_K}|/\delta)} \right.$$
$$\left. + \sqrt{\frac{1}{N} \left( \max_{r \in \mathcal{R}} \mathrm{VaR}_{\pi^E;r} + \frac{1}{N^2} \right) \log(|(\mathcal{R})_{\rho_N}|/\delta)} \right)$$

$$\lesssim \mathcal{O}\left( \frac{\log(\mathcal{N}_{\rho_K}(\mathcal{R})/\delta)}{K} + \frac{\log(\mathcal{N}_{\rho_N}(\mathcal{R})/\delta)}{N} + \sqrt{\frac{1}{K} \sum_{k=1}^{K} (\max_{r \in \mathcal{R}} \mathrm{VaR}_{\pi^k;r} + \frac{1}{K^2}) \log(\mathcal{N}_{\rho_K}(\mathcal{R})/\delta)} \right.$$
$$\left. + \sqrt{\frac{1}{N} \left( \max_{r \in \mathcal{R}} \mathrm{VaR}_{\pi^E;r} + \frac{1}{N^2} \right) \log(\mathcal{N}_{\rho_N}(\mathcal{R})/\delta)} \right)$$

When we incorporate FTRL as the no-regret algorithm for reward optimization, providing an $O(1/\sqrt{K})$ bound for $\epsilon_{\mathrm{opt}}^r$ and keeping only the dominating terms, we can finally upper bound the reward error as:

$$\mathrm{T1} \lesssim \mathcal{O}\left( \frac{1}{\sqrt{K}} + \frac{\log(\mathcal{N}_{\rho_K}(\mathcal{R})/\delta)}{K} + \frac{\log(\mathcal{N}_{\rho_N}(\mathcal{R})/\delta)}{N} + \sqrt{\frac{1}{K} \sum_{k=1}^{K} \max_{r \in \mathcal{R}} \mathrm{VaR}_{\pi^k;r} \log(\mathcal{N}_{\rho_K}(\mathcal{R})/\delta)} \right.$$
$$\left. + \sqrt{\frac{1}{N} \max_{r \in \mathcal{R}} \mathrm{VaR}_{\pi^E;r} \log(\mathcal{N}_{\rho_N}(\mathcal{R})/\delta)} \right) \tag{C.11}$$

Further upper bounding the estimation error with $\mathcal{N}_{\mathcal{R}} = \max\{\mathcal{N}_{\rho_K}(\mathcal{R}), \mathcal{N}_{\rho_N}(\mathcal{R})\}$ concludes the proof. $\square$

### C.5.2 UPPER BOUND FOR THE POLICY ERROR

*Proof of Lemma C.15.* For the policy error, by following the approach in Wang et al. (2024), and applying Lemma C.12 together with the proof steps in Appendix C.4, we obtain a policy error bound under general function approximation using bracketing number for model class covering. Specifically, with

$$\beta = 7 \log\left( \frac{K \mathcal{N}_{[]}\left( (KH|\mathcal{S}|)^{-1}, \mathcal{P}, \|\cdot\|_\infty \right)}{\delta} \right),$$

the bound holds with probability at least probability $1 - \delta$:

$$\mathrm{T2} \lesssim \mathcal{O}\left( \frac{1}{K} \sqrt{\sum_{k=1}^{K} \max_{r \in \mathcal{R}} \mathrm{VaR}_{\pi^k,r} \cdot d_E \log(KH\mathcal{N}_{[]}((KH|\mathcal{S}|)^{-1}, \mathcal{P}, \|\cdot\|_\infty)/\delta) \log(KH)} \right.$$
$$\left. + \frac{1}{K} d_E \cdot \log(KH\mathcal{N}_{[]}((KH|\mathcal{S}|)^{-1}, \mathcal{P}, \|\cdot\|_\infty)/\delta) \log(KH) \right),$$

where $d_E = DE_1(\mathcal{G}, \mathcal{S} \times \mathcal{A}, 1/KH)$. $\square$

## D PROOF OF THE LOWER BOUNDS

### D.1 DETAILS OF THE HARD INSTANCE STRUCTURE AND PROOF SKETCH

In this subsection, we define the transition dynamics, the reward function of the constructed hard instance (Figure 1), and the policy class used in the lower bound analysis. Formally speaking, the transition dynamics and reward function is written by:

$$P(\mathbf{s}' = \boxed{\mathbf{s}, 1} \| \mathbf{s} = \boxed{\mathbf{s}, 0}, \mathbf{a}) = \tfrac{1}{2} + \epsilon_P \langle \boldsymbol{\mu}, \mathbf{a}_P \rangle, \quad P(\mathbf{s}' = \boxed{\mathbf{s}_0, 1} \| \mathbf{s} = \boxed{\mathbf{s}, 0}, \mathbf{a}) = \tfrac{1}{2} - \epsilon_P \langle \boldsymbol{\mu}, \mathbf{a}_P \rangle,$$
$$P(\mathbf{s}' = \boxed{\mathbf{s}, 1} \| \mathbf{s} = \boxed{\mathbf{s}, 1}, \mathbf{a}) = P(\mathbf{s}' = \boxed{\mathbf{s}_0, 1} \| \mathbf{s} = \boxed{\mathbf{s}_0, 1}, \mathbf{a}) = 1, \; \forall \mathbf{a} \in \mathcal{A} = \mathbb{B}^{d_P+1},$$
$$r(\mathbf{s} = \boxed{\mathbf{s}, 1}, \mathbf{a}) = \tfrac{\tau}{H}(\tfrac{1}{2} + \tfrac{a_R}{2d_R} \langle \boldsymbol{\theta}, \mathbf{s} \rangle) \quad P(\mathbf{s}'|\mathbf{s}, \mathbf{a}) = 0, r(\mathbf{s}, \mathbf{a}) = 0 \text{ otherwise if not stated.}$$

It's easy to verify that the reward class $\log|\mathcal{R}| = \Theta(d_R)$ and the model class $\log|\mathcal{P}| = \Theta(d_P)$ in this setting, while $\log|\mathcal{S}| = \Theta(d_R)$ and $\log|\mathcal{A}| = \Theta(d_P)$.

**Policy class.** We define a stable stochastic policy class parameterized by $(\widehat{\boldsymbol{\mu}}, \widehat{\boldsymbol{\theta}})$. The policy $\pi$ outputs an action $\mathbf{a} \in \mathbb{B}^{d_P+1}$, sampled from a Rademacher distribution biased by $\widehat{\boldsymbol{\mu}}$ and $\widehat{\boldsymbol{\theta}}$:

$$\pi(\mathbf{a} \mid \mathbf{s}; \widehat{\boldsymbol{\mu}}, \widehat{\boldsymbol{\theta}}) = \text{Rademacher}\left( \left[ \tfrac{1}{2} \cdot \mathbf{1}_{d_P} + \epsilon_\pi d_R \widehat{\boldsymbol{\mu}} \right] \oplus \left[ \tfrac{1}{2} + \epsilon_\pi \langle \mathbf{s}, \widehat{\boldsymbol{\theta}} \rangle \right] \right), \tag{D.1}$$

where $\oplus$ denotes concatenation, $\widehat{\boldsymbol{\mu}} \in \mathbb{B}^{d_P}$ and $\widehat{\boldsymbol{\theta}} \in \mathbb{M}^{d_R}$. For an action $\mathbf{a}$ generated by the policy $\pi(\cdot \mid \mathbf{s}; \widehat{\boldsymbol{\mu}}, \widehat{\boldsymbol{\theta}})$, we can find that the first component $\mathbf{a}_P$ is governed by $\widehat{\boldsymbol{\mu}}$, while the second component $a_R$ is determined by both the state $\mathbf{s}$ and the parameter $\widehat{\boldsymbol{\theta}}$. For any instance defined by parameters $(\boldsymbol{\mu}^*, \boldsymbol{\theta}^*)$, the offline expert demonstrations are generated by the expert policy $\pi^E = \pi(\mathbf{a} \mid \mathbf{s}; \boldsymbol{\mu}^*, \boldsymbol{\theta}^*)$.

The high-level idea behind the proof of Theorem 5.8 is summarized as follows:

*Proof Sketch of Theorem 5.8.* We first relates the variance $\sigma^2$ with parameter $\tau$ in the hard instance and the Fano's inequality. We would like to highlight that the lower bound is built on two cases from the hard instances where $(\epsilon_P, \epsilon_\pi) = (\epsilon, 1/(4d_R))$ when the policy is hard-to-learn and $(\epsilon_P, \epsilon_\pi) = (1/(4d_P), \epsilon)$ when the model is hard-to-learn. Then for all algorithm that seeks a uniform performance on all hard instances, the final lower bound is obtained by taking the maximum of these two cases. $\square$

### D.2 TECHNICAL LEMMAS

To construct the proof of the lower bounds, we first present a few technical lemmas that will be used during the proof. We first provide a lemma for the existence and property regarding the reward parameter space $\mathbb{M}^{d_R} \subseteq \mathbb{B}^{d_R}$:

**Lemma D.1.** Given $\gamma \in (0,1)$, there exists a $\mathbb{M}^{d_R} \subset \mathbb{B}^{d_R}$ such that for any two different vectors $\mathbf{x}, \mathbf{x}' \in \mathbb{M}^{d_R}$ and $\langle \mathbf{x}, \mathbf{x}' \rangle \leq d_R \gamma$, and the log-cardinality of the proposed set is bounded as $\log|\mathbb{M}^{d_R}| < d_R \gamma^2/4$.

*Proof of Lemma D.1.* To begin with, we assume that $\mathbf{x} \sim \text{Unif}(\mathbb{B}^{d_R})$, i.e. $[\mathbf{x}]_i \sim \{-1, 1\}$. Thus given any $\mathbf{x}, \mathbf{x}' \sim \text{Unif}(\mathbb{B}^{d_R})$, we have

$$\mathbb{P}(\langle \mathbf{x}, \mathbf{x}' \rangle \geq d_R \gamma) = \mathbb{P}_{z_i \sim \text{Unif}\{-1,1\}}\left( \sum_{i=1}^{d_R} z_i \geq d_R \gamma \right)$$

$$= \mathbb{P}_{z_i \sim \text{Unif}\{-1,1\}}\left( \frac{1}{d_R} \sum_{i=1}^{d_R} z_i \geq \gamma \right)$$

$$\leq \exp(-d_R \gamma^2/2),$$

The last inequality follows from the Azuma-Hoeffding inequality, noting that $z_i \sim \text{Unif}\{-1, 1\}$ is a bounded random variable. Consider a set $\mathbb{M}^{d_R}$ of cardinality $|\mathbb{M}^{d_R}|$; there are at most $|\mathbb{M}^{d_R}|^2$ pairs $(\mathbf{x}, \mathbf{x}')$. Applying a union bound over all such vector pairs, we obtain

$$\mathbb{P}(\exists \mathbf{x}, \mathbf{x}' \in \mathbb{M}^{d_R}, \mathbf{x} \neq \mathbf{x}', \langle \mathbf{x}, \mathbf{x}' \rangle \geq d_R \gamma) \leq |\mathbb{M}^{d_R}|^2 \exp(-d_R \gamma^2/2),$$

thus

$$\mathbb{P}(\forall \mathbf{x}, \mathbf{x}' \in \mathbb{M}^{d_R}, \mathbf{x} \neq \mathbf{x}', \langle \mathbf{x}, \mathbf{x}' \rangle \leq d_R \gamma) \geq 1 - |\mathbb{M}^{d_R}|^2 \exp(-d_R \gamma^2/2),$$

Once we have that $|\mathbb{M}^{d_R}|^2 \exp(-d_R \gamma^2/2) < 1$, there exists a set $\mathbb{M}^{d_R}$ such that for any two different vector $\mathbf{x}, \mathbf{x}' \in \mathbb{M}^{d_R}, \langle \mathbf{x}, \mathbf{x}' \rangle \leq d_R \gamma$. $\square$

By Lemma D.1, for any $\mathbf{x}, \mathbf{x}' \in \mathbb{M}^{d_R}$, we have $\|\mathbf{x} - \mathbf{x}'\|_1 \geq d_R \mathbb{1}[\mathbf{x} \neq \mathbf{x}']$, since at least one coordinate differs between the two vectors, contributing at least $d_R$ to the $\ell_1$ distance. By Lemma D.1, we can construct a parameter set $\mathbb{M}^{d_R}$ with $|\mathbb{M}^{d_R}| = \lceil \exp(d_R \gamma^2/4) \rceil - 1$ and $\gamma = \frac{1}{2}$. The constructed set indeed satisfies Lemma D.1, and thus we obtain the following lower bound on $\log|\mathbb{M}^{d_R}|$:

**Lemma D.2.** Given $\gamma = \frac{1}{2}$, for $d_R > 48$, we have the following lower bound on $\log |\mathbb{M}^{d_R}|$, where $|\mathbb{M}^{d_R}| = \lceil \exp(d_R \gamma^2 / 4) \rceil - 1$:

$$\log |\mathbb{M}^{d_R}| \geq \frac{d_R \gamma^2}{4} - 3$$

*Proof of Lemma D.2.*

$$\log |\mathbb{M}^{d_R}| \geq \log(\exp(d_R \gamma^2 / 4) - 1)$$
$$= \frac{d_R \gamma^2}{4} + \log\left(1 - \exp\left(-\frac{d_R \gamma^2}{4}\right)\right)$$
$$\geq \frac{d_R \gamma^2}{4} - 3,$$

where the last inequality holds by applying $\gamma = 1/2$ and $d_R > 48$. $\qquad\square$

We then provide a useful lemma for $\mathbb{M}^{d_R}$ to assist the proof of Lemma D.7:

**Lemma D.3.** Consider $\mathbf{x}, \mathbf{y}, \mathbf{y}' \in \mathbb{M}^{d_R}$, given $\gamma = \frac{1}{2}$, we have the following lower bound:

$$\max_{\mathbf{x} \in \mathbb{M}^{d_R}} \mathbf{x}^\top (\mathbf{y} - \mathbf{y}') \geq \frac{1}{2} d_R \, \mathbb{1}[\mathbf{y} \neq \mathbf{y}']$$

*Proof of Lemma D.3.* Since we can write the LHS of the inequality as:

$$\max_{\mathbf{x} \in \mathbb{M}^{d_R}} \mathbf{x}^\top (\mathbf{y} - \mathbf{y}') = \mathbf{y}^\top (\mathbf{y} - \mathbf{y}') = \|\mathbf{y} - \mathbf{y}'\|_1.$$

Consider two cases: (i): $\mathbf{y} = \mathbf{y}'$ and (ii): $\mathbf{y} \neq \mathbf{y}'$. For the first case, it's easy to verify that $\|\mathbf{y} - \mathbf{y}'\|_1 = \mathbb{1}[\mathbf{y} \neq \mathbf{y}'] = 0$, which makes LHS = RHS. For the second case, since we have:

$$\|\mathbf{y} - \mathbf{y}'\|_1 = d_R - \mathbf{y}^\top \mathbf{y}' \geq d_R(1 - \gamma) \, \mathbb{1}[\mathbf{y} \neq \mathbf{y}'] = \frac{1}{2} d_R \, \mathbb{1}[\mathbf{y} \neq \mathbf{y}'],$$

where the inequality follows from $\langle \mathbf{y}, \mathbf{y}' \rangle \leq d_R \gamma$ and $\mathbb{1}[\mathbf{y} \neq \mathbf{y}'] = 1$, with $\gamma = \frac{1}{2}$ applied in the final step of the proof. Combining two cases, we conclude the proof. $\qquad\square$

We finally present some inequalities that are instrumental in our lower bound analysis:

**Lemma D.4** (KL divergence for Rademacher distribution). Let $P, Q$ be two Rademacher distributions with probability $\frac{1}{2} + \epsilon$ and $\frac{1}{2} - \epsilon$, for $\epsilon \leq 1/4$, the KL divergence of $P$ and $Q$ can be upper bounded as:

$$\mathrm{KL}(P, Q) \leq 16\epsilon^2$$

*Proof of Lemma D.4.*

$$\mathrm{KL}(P, Q) = \mathbb{E}_P \log(P/Q)$$
$$= (\frac{1}{2} + \epsilon) \log\left(\frac{\frac{1}{2} + \epsilon}{\frac{1}{2} - \epsilon}\right) + (\frac{1}{2} - \epsilon) \log\left(\frac{\frac{1}{2} - \epsilon}{\frac{1}{2} + \epsilon}\right)$$
$$= 2\epsilon \, \log\left(\frac{1 + 2\epsilon}{1 - 2\epsilon}\right)$$
$$\leq 16\epsilon^2,$$

where the last inequality leverages the property $x \log(1 + x)/(1 - x) \leq 4x^2$ when $x \leq \frac{1}{2}$. $\qquad\square$

**Lemma D.5** (Fano's inequality, Fano 1961)**.** Consider a finite set of probability measures $\{\mathbb{P}_{\mathbf{u}} : \mathbf{u} \in \mathcal{U}\}$ on a measurable space $\Omega$. Let $\widehat{\mathbf{u}}$ be any estimator based on samples drawn from $\mathbb{P}_{\mathbf{u}}$. Then, for any reference distribution $\mathbb{P}_0$ on $\Omega$, the average probability of error satisfies:

$$\frac{1}{|\mathcal{U}|} \sum_{\mathbf{u} \in \mathcal{U}} \mathbb{P}_{\mathbf{u}}[\widehat{\mathbf{u}} \neq \mathbf{u}] \geq 1 - \frac{\log 2 + \frac{1}{|\mathcal{U}|} \sum_{\mathbf{u}} \mathrm{KL}(\mathbb{P}_{\mathbf{u}}, \mathbb{P}_0)}{\log |\mathcal{U}|}.$$

**Lemma D.6** (Bretagnolle–Huber inequality, Bretagnolle & Huber 1979)**.** Let $P$ and $Q$ be two probability measures on the same measurable space, and let $A$ be any measurable event. Then

$$P(A) + Q(A^c) \geq \tfrac{1}{2} \exp\left(-\mathrm{KL}(P \,\|\, Q)\right),$$

where $\mathrm{KL}(P \,\|\, Q)$ denotes the KL divergence between $P$ and $Q$.

### D.3 AIL GAP LOWER BOUND

In this section, we provide a lemma that draws the connection between the performance loss and the estimation error of the parameter $\boldsymbol{\mu}$ and $\boldsymbol{\theta}$.

**Lemma D.7.** Consider the model parameter $(\boldsymbol{\mu}^*, \boldsymbol{\theta}^*)$ and the corresponding expert policy $\pi^E$ described by $\pi(\mathbf{a} \mid \mathbf{s}; \boldsymbol{\mu}^*, \boldsymbol{\theta}^*)$, for any $\pi \in \Pi$ with parameter $(\widehat{\boldsymbol{\mu}}, \widehat{\boldsymbol{\theta}})$, the AIL risk is bounded by:

$$\max_{r \in \mathcal{R}} \mathbb{E}[V_{1;P^\star;r}^{\pi^E}(\mathbf{s}) - V_{1;P^\star;r}^{\pi}(\mathbf{s})] \geq \tfrac{\tau \epsilon_P \epsilon_\pi d_R}{2} \|\boldsymbol{\mu}^* - \widehat{\boldsymbol{\mu}}\|_1 + \tfrac{1}{2} \epsilon_\pi \tau \, \mathbb{1}[\boldsymbol{\theta}^* \neq \widehat{\boldsymbol{\theta}}],$$

where the expectation is taken over the uniform distribution over the state space.

*Proof of Lemma D.7.* To begin with, for any reward function parameter $\widetilde{\boldsymbol{\theta}} \in \mathbb{M}^{d_R}$ and policy $\pi$ with parameter $\widehat{\boldsymbol{\mu}}, \widehat{\boldsymbol{\theta}}$, the value function $V_{2;P^\star;r}^{\pi}(\mathbf{s})$ can be calculated by

$$V_{2;P^\star;r}^{\pi}(\mathbf{s}) = \mathbb{E}_{\mathbf{a}}\left[\sum_{h=1}^{H} \tfrac{1}{2H}\tau + \tfrac{a_R \tau}{2H d_R}\langle \widetilde{\boldsymbol{\theta}}, \mathbf{s}\rangle\right] = \tfrac{1}{2}\tau + \tfrac{\tau}{2d_R}\langle \boldsymbol{\theta}, \mathbf{s}\rangle \mathbb{E}_{\mathbf{a} \sim \pi(\cdot|\mathbf{s})}[a_R] = \tfrac{1}{2}\tau + \tfrac{\tau \epsilon_\pi}{d_R}\langle \widetilde{\boldsymbol{\theta}}, \mathbf{s}\rangle\langle \widehat{\boldsymbol{\theta}}, \mathbf{s}\rangle,$$

whee the second last equation is due to the implementation of a stable policy and the last equation is from the definition of the policy class (D.1). The transition probability from the initial state $\boxed{\mathbf{s}, 0}$ to the state $\boxed{\mathbf{s}, 1}$ is:

$$P(\mathbf{s}' = \boxed{\mathbf{s}, 1} | \mathbf{s} = \boxed{\mathbf{s}, 0}, \pi(\widehat{\boldsymbol{\theta}}, \widehat{\boldsymbol{\mu}})) = \mathbb{E}_{\mathbf{a}}\left[\tfrac{1}{2} + \epsilon_P \langle \boldsymbol{\mu}^*, \mathbf{a}_P\rangle\right] = \tfrac{1}{2} + \epsilon_P \epsilon_\pi d_R \langle \boldsymbol{\mu}^*, \widehat{\boldsymbol{\mu}}\rangle, \tag{D.2}$$

where the expectation is taken over the action $\mathbf{a}$ defined in the policy class. Since $V_{2;P^\star;r}^{\pi}(\mathbf{s} = \boxed{\mathbf{s}_0, 1})$ is always zero, applying Bellman's equation yields:

$$V_{1;P^\star;r}^{\pi}(\mathbf{s}) = \left(\tfrac{1}{2} + \epsilon_\pi \epsilon_P d_R \langle \boldsymbol{\mu}^*, \widehat{\boldsymbol{\mu}}\rangle\right) \cdot \left(\tfrac{1}{2}\tau + \tfrac{\epsilon_\pi \tau}{d_R}\langle \widetilde{\boldsymbol{\theta}}, \mathbf{s}\rangle\langle \widehat{\boldsymbol{\theta}}, \mathbf{s}\rangle\right)$$

$$= \tfrac{1}{4}\tau + \tfrac{\tau \epsilon_\pi \epsilon_P d_R \langle \boldsymbol{\mu}^*, \widehat{\boldsymbol{\mu}}\rangle}{2} + \tfrac{\epsilon_\pi}{d_R}\left(\tfrac{1}{2}\tau + \tau \epsilon_P \epsilon_\pi d_R \langle \boldsymbol{\mu}^*, \widehat{\boldsymbol{\mu}}\rangle\right)\langle \widetilde{\boldsymbol{\theta}}, \mathbf{s}\rangle\langle \widehat{\boldsymbol{\theta}}, \mathbf{s}\rangle. \tag{D.3}$$

Repeating the calculation presented in (D.3) for $\pi^E$ parameterized by $\boldsymbol{\mu}^*, \boldsymbol{\theta}^*$, the performance gap between any policy $\pi$ and the expert policy $\pi^E$ can be written by:

$$V_{1;P^\star;r}^{\pi^E}(\mathbf{s}) - V_{1;P^\star;r}^{\pi}(\mathbf{s}) = \tfrac{\tau \epsilon_P \epsilon_\pi d_R \langle \boldsymbol{\mu}^*, \boldsymbol{\mu}^* - \widehat{\boldsymbol{\mu}}\rangle}{2} + \tfrac{\epsilon_\pi \tau}{2d_R}\langle \widetilde{\boldsymbol{\theta}}, \mathbf{s}\rangle \cdot \langle \boldsymbol{\theta}^* - \widehat{\boldsymbol{\theta}}, \mathbf{s}\rangle$$

$$+ \epsilon_\pi^2 \epsilon_P \tau \left(\langle \boldsymbol{\mu}^*, \boldsymbol{\mu}^*\rangle\langle \widetilde{\boldsymbol{\theta}}, \mathbf{s}\rangle\langle \boldsymbol{\theta}^*, \mathbf{s}\rangle - \langle \boldsymbol{\mu}^*, \widehat{\boldsymbol{\mu}}\rangle\langle \widetilde{\boldsymbol{\theta}}, \mathbf{s}\rangle\langle \widehat{\boldsymbol{\theta}}, \mathbf{s}\rangle\right)$$

$$\geq \tfrac{\tau d_R \epsilon_P \epsilon_\pi \langle \boldsymbol{\mu}^*, \boldsymbol{\mu}^* - \widehat{\boldsymbol{\mu}}\rangle}{2} + \tfrac{\epsilon_\pi \tau}{2d_R}\langle \widetilde{\boldsymbol{\theta}}, \mathbf{s}\rangle \cdot \langle \boldsymbol{\theta}^* - \widehat{\boldsymbol{\theta}}, \mathbf{s}\rangle + \tau \epsilon_\pi^2 \epsilon_P d_P \langle \widetilde{\boldsymbol{\theta}}, \mathbf{s}\rangle \cdot \langle \boldsymbol{\theta}^* - \widehat{\boldsymbol{\theta}}, \mathbf{s}\rangle, \tag{D.4}$$

where the last inequality uses the fact that $\langle \boldsymbol{\mu}^*, \widehat{\boldsymbol{\mu}}\rangle \leq \langle \boldsymbol{\mu}^*, \widehat{\boldsymbol{\mu}}^*\rangle = d_P$. Therefore, taking the adversarial reward by maximizing this gap over all possible $\widetilde{\boldsymbol{\theta}} \in \mathbb{M}^{d_R}$, (D.4) and taking an expectation over the state space yields

$$\max_{r \in \mathcal{R}} \mathbb{E}_{\mathbf{s}}[V_{1;P^\star;r}^{\pi^E}(\mathbf{s}) - V_{1;P^\star;r}^{\pi}(\mathbf{s})]$$

$$\geq \max_{\widetilde{\boldsymbol{\theta}}} \mathbb{E}_{\mathbf{s}} \tfrac{\tau d_R \epsilon_P \epsilon_\pi \langle \boldsymbol{\mu}^*, \boldsymbol{\mu}^* - \widehat{\boldsymbol{\mu}}\rangle}{2} + \tfrac{\epsilon_\pi \tau}{2d_R}\widetilde{\boldsymbol{\theta}}^\top \mathbf{s}\mathbf{s}^\top(\boldsymbol{\theta}^* - \widehat{\boldsymbol{\theta}}) + \tau \epsilon_\pi^2 \epsilon_P d_P \widetilde{\boldsymbol{\theta}}^\top \mathbf{s}\mathbf{s}^\top(\boldsymbol{\theta}^* - \widehat{\boldsymbol{\theta}})$$

$$\geq \tfrac{\tau d_R \epsilon_P \epsilon_\pi}{2} \|\boldsymbol{\mu}^* - \widehat{\boldsymbol{\mu}}\|_1 + \tfrac{1}{2}(\tau \epsilon_\pi + \tau \epsilon_\pi^2 \epsilon_P d_P) \, \mathbb{1}[\boldsymbol{\theta}^* \neq \widehat{\boldsymbol{\theta}}]$$

$$\geq \tfrac{\tau d_R \epsilon_P \epsilon_\pi}{2} \|\boldsymbol{\mu}^* - \widehat{\boldsymbol{\mu}}\|_1 + \tfrac{1}{2}\epsilon_\pi \tau \, \mathbb{1}[\boldsymbol{\theta}^* \neq \widehat{\boldsymbol{\theta}}], \tag{D.5}$$

where we use the property $\mathbb{E}_{\mathbf{s}}\mathbf{s}\mathbf{s}^\top = \mathbf{I}$, Lemma D.1 and D.3 in the second inequality. $\square$

## D.4 Lower Bound for Reward Accuracy

In order to lower bound the second term in Lemma D.7 related to the reward parameter $\boldsymbol{\theta}$, we introduce the following lemma:

**Lemma D.8** (Lower Bound for Reward Accuracy). Given the defined MDP structure, reward function, and policy class, for $d_R > 48$, any (offline or online) imitation learning algorithm must suffer from the reward estimation error by:

$$\mathbb{E}_{\boldsymbol{\theta} \sim \mathbb{M}^{d_R}} \left[ \mathbb{P}_{\boldsymbol{\theta}}[\widehat{\boldsymbol{\theta}} \neq \boldsymbol{\theta}] \right] \geq 1 - \left( 1 + \left( \frac{16}{d_R - 48} \right) \right) \log 2 - 16 N \epsilon_\pi^2,$$

where the expectation is taken over the uniform distribution in $\boldsymbol{\theta} \in \mathbb{M}^{d_R}$ and $P_{\boldsymbol{\theta}}$ measures the probabilities when the reward model is $\boldsymbol{\theta}$.

For the proof of Lemma D.8, we start from leveraging the Fano's inequality as in Lemma D.5. We are then ready for the proof.

*Proof of Lemma D.8.* We start from the Fano's inequality with $\mathbf{u} \leftarrow \boldsymbol{\theta}$ thus $\log |\mathcal{U}| = \log |\mathbb{M}^{d_R}| \leq d_R \gamma^2/4$. The reference distribution is defined by $\mathbb{P}_0 = \frac{1}{|\mathcal{U}|} \sum_{\mathbf{v}} \mathbb{P}_{\mathbf{v}}$. Since the agent can only observe the reward at step $h = H$, we denote $x = \{a_R^H\}_n$ over $n \in [N]$ offline demonstrations. Then for any $\mathbf{v} \in \mathcal{U}$, the KL divergence $\mathrm{KL}(\mathbb{P}_{\mathbf{u}}, \mathbb{P}_0)$ is bounded by:

$$\mathrm{KL}(\mathbb{P}_{\mathbf{u}}, \mathbb{P}_0) = \mathbb{E}_{x \sim \mathbb{P}_{\mathbf{u}}} \left[ \log \frac{\mathbb{P}_{\mathbf{u}}(x)}{2^{-\log |\mathbb{M}^{d_R}|} \sum_{\mathbf{v} \in \mathcal{U}} \mathbb{P}_{\mathbf{v}}(x)} \right]$$

$$\leq \log |\mathbb{M}^{d_R}| \log 2 + \mathbb{E}_{x \sim \mathbb{P}_{\mathbf{u}}} \left[ \log \frac{\mathbb{P}_{\mathbf{u}}(x)}{\mathbb{P}_{\mathbf{v}}(x)} \right]$$

$$= \log |\mathbb{M}^{d_R}| \log 2 + \mathrm{KL}(\mathbb{P}_{\mathbf{u}}, \mathbb{P}_{\mathbf{v}}). \tag{D.6}$$

Then consider $\mathbf{u} \leftarrow \boldsymbol{\theta}_0$, let $\mathbf{v} \leftarrow \boldsymbol{\theta}_1 \in \mathbb{M}^{d_R}$ as a vector that only differs in one coordinate, according to the definition of the policy class, over the action sequence $\{a_R^H\}_{1:N}$ collected over the offline demonstration with it's length as $N$, for $\epsilon_\pi \leq 1/(4 d_R)$ and $d_R \geq 1$, we have:

$$\mathrm{KL}(\mathbb{P}(\{a_R^H\}_{1:N} | \boldsymbol{\theta}_0), \mathbb{P}(\{a_R^H\}_{1:N} | \boldsymbol{\theta}_1)) = N \cdot \mathbb{E}_{\mathbf{s}}[\mathrm{KL}(\mathbb{P}(a_R | \boldsymbol{\theta}_0), \mathbb{P}(a_R | \boldsymbol{\theta}_1))]$$

$$= N d_R \mathrm{KL}(\mathrm{Rademacher}(\tfrac{1}{2} + \epsilon_\pi), \mathrm{Rademacher}(\tfrac{1}{2} - \epsilon_\pi))$$

$$= 2 N d_R \epsilon_\pi \log \frac{1 + 2\epsilon_\pi}{1 - 2\epsilon_\pi} \leq 16 N d_R \epsilon_\pi^2, \tag{D.7}$$

where $\epsilon_\pi \leq 1$, and the last inequality is because of Lemma D.4. Plugging (D.7) and (D.6) into the statement of Lemma D.5 yields:

$$\mathbb{E}_{\boldsymbol{\theta} \sim \mathbb{M}^{d_R}} \left[ \mathbb{P}_{\boldsymbol{\theta}}[\widehat{\boldsymbol{\theta}} \neq \boldsymbol{\theta}] \right] \geq 1 - \log^{-1} |\mathbb{M}^{d_R}| \left( \log 2 + \tfrac{1}{|\mathcal{U}|} \sum_{\mathbf{u}} \mathrm{KL}(\mathbb{P}_{\mathbf{u}}, \mathbb{P}_0) \right)$$

$$\geq 1 - \log^{-1} |\mathbb{M}^{d_R}| \left( \log 2 + \log^{-1} |\mathbb{M}^{d_R}| \log 2 + \mathrm{KL}(\mathbb{P}_{\boldsymbol{\theta}_0}, \mathbb{P}_{\boldsymbol{\theta}_1}) \right)$$

$$\geq 1 - (1 + \log^{-1} |\mathbb{M}^{d_R}|) \log 2 - 16 N \epsilon_\pi^2$$

$$\geq 1 - \left( 1 + \frac{16}{d_R - 48} \right) \log 2 - 16 N \epsilon_\pi^2, \tag{D.8}$$

where the last inequality leverages Lemma D.2 with $\gamma = \frac{1}{2}$ according to the construction of $\mathbb{M}^{d_R}$ for $d_R > 48$. $\qquad \square$

## D.5 Lower Bound for Model Accuracy

In order to lower bound the first term in Lemma D.7 related to the model parameter $\boldsymbol{\mu} \in \mathbb{B}^{d_P}$, we introduce the following lemma:

**Lemma D.9** (Lower Bound for Model Accuracy). Given the defined MDP structure, reward function, and policy class, any (offline or online) imitation learning algorithm must suffer from the model estimation error by

$$\sum_{i=1}^{d_P} \mathbb{P} \left( \sum_{k=1}^{K} \mathbb{1}[\widehat{\boldsymbol{\mu}}_i \neq \boldsymbol{\mu}_i^*] > \frac{K}{2} \right) \geq \frac{d_P}{4} \exp \left( -16 - N(16 \epsilon_\pi^2 + \tfrac{16}{K}) \right),$$

The proof of the estimation error for $\boldsymbol{\mu}$ follows from an analysis based on the Bretagnolle–Huber inequality, as detailed in Lemma D.6.

*Proof of Lemma D.9.* We start from the Bretagnolle–Huber inequality with $\mathbf{u} \leftarrow \boldsymbol{\mu}$ thus $|\mathcal{U}| = 2^{d_P}$. The agent can infer the model $\boldsymbol{\mu}$ from two streams: first, the expert demonstration $\{\mathbf{a}_1^n\}$ contains the model information as $\mathbf{a}_P$; second, from both the expert demonstration and online exploration, the agent can observe $\{\boxed{\mathbf{s},1}, \boxed{\mathbf{s}_0,1}\}$. Therefore, we denote the $\mathbb{P}(x)$ as the joint distribution of $\{\mathbf{a}_P^1, \boxed{\mathbf{s},1}\}_{n \in [N]} \cup \{\mathbf{a}_P^1, \boxed{\mathbf{s},1}\}_{k \in [K]}$. Consider $\mathbf{u} \leftarrow \boldsymbol{\mu}_0$ and $\mathbf{v} \leftarrow \boldsymbol{\mu}_1 \in \mathbb{B}^{d_P}$ as vectors that differ only one dimension. According to the chain rule of the KL divergence, let $P_k(\boldsymbol{\mu}) := P(\boxed{\mathbf{s},1}_k \| \boxed{\mathbf{s},0}_k, \{\mathbf{a}^1\}_k, \boldsymbol{\mu})$ and $P_n(\boldsymbol{\mu}) := P(\boxed{\mathbf{s},1}_n \| \boxed{\mathbf{s},0}_n, \{\mathbf{a}^1\}_n, \boldsymbol{\mu})$, we have that

$$
\begin{aligned}
\mathrm{KL}(\mathbb{P}_\mathbf{u}, \mathbb{P}_\mathbf{v}) &= N \cdot \mathrm{KL}(\mathbb{P}_\mathbf{u}(\{\mathbf{a}_P^1\}_n, \boxed{\mathbf{s},1}_n\}), \mathbb{P}_\mathbf{v}(\{\mathbf{a}_P^1\}_n, \boxed{\mathbf{s},1}_n\})) \\
&\quad + K \cdot \mathrm{KL}(\mathbb{P}_\mathbf{u}(\{\mathbf{a}_P^1\}_k, \boxed{\mathbf{s},1}_k\}), \mathbb{P}_\mathbf{v}(\{\mathbf{a}_P^1\}_n, \boxed{\mathbf{s},1}_k\})) \\
&= N \cdot \left( \mathrm{KL}\left( \pi_{\boldsymbol{\mu}_0}^E(\{\mathbf{a}_P^1\}_n), \pi_{\boldsymbol{\mu}_1}^E(\{\mathbf{a}_P^1\}_n) \right) + \mathrm{KL}\left( P_n(\boldsymbol{\mu}_0), P_n(\boldsymbol{\mu}_1) \right) \right) \\
&\quad + K \cdot \left( \mathrm{KL}\left( \pi^k(\{\mathbf{a}_P^1\}_k), \pi^k(\{\mathbf{a}_P^1\}_k) \right) + \mathrm{KL}\left( P_k(\boldsymbol{\mu}_0), P_k(\boldsymbol{\mu}_1) \right) \right) \\
&= N \cdot \left( \mathrm{KL}\left( \pi_{\boldsymbol{\mu}_0}^E(\{\mathbf{a}^1\}_n), \pi_{\boldsymbol{\mu}_1}^E(\{\mathbf{a}^1\}_n) \right) + \mathrm{KL}\left( P_n(\boldsymbol{\mu}_0), P_n(\boldsymbol{\mu}_1) \right) \right) + K \cdot \mathrm{KL}\left( P_k(\boldsymbol{\mu}_0), P_k(\boldsymbol{\mu}_1) \right),
\end{aligned}
$$

where the last equation drops the term $\mathrm{KL}\left( \pi^k(\{\mathbf{a}^1\}_k), \pi^k(\{\mathbf{a}^1\}_k) \right)$ since the online policy is not affected by the offline demonstration, in terms of the model-related part. According to the definition of the expert policy and model, by Lemma D.4, for $\epsilon_\pi \le 1/(4d_R)$, $\epsilon_P \le 1/(d_P)$ and $d_R, d_P \ge 1$, we have:

$$
\mathrm{KL}\left( \pi_{\boldsymbol{\mu}_0}^E(\{\mathbf{a}_P^1\}_n), \pi_{\boldsymbol{\mu}_1}^E(\{\mathbf{a}_P^1\}_n) \right) \le 16 d_R^2 \epsilon_\pi^2
$$

$$
\mathrm{KL}\left( P(\boxed{\mathbf{s},1}_n \| \boxed{\mathbf{s},0}_n, \{\mathbf{a}^1\}_n, \boldsymbol{\mu}_0), P(\boxed{\mathbf{s},1}_n \| \boxed{\mathbf{s},0}_n, \{\mathbf{a}^1\}_n, \boldsymbol{\mu}_1) \right) \le 16 \epsilon_P^2
$$

Therefore, we can bound the KL divergence:

$$
\mathrm{KL}(\mathbb{P}_\mathbf{u}, \mathbb{P}_\mathbf{v}) \le K \cdot 16 \epsilon_P^2 + N \cdot (16 d_R^2 \epsilon_\pi^2 + 16 \epsilon_P^2). \tag{D.9}
$$

For $i \in [d_P]$, we define:

$$
p_{\mathbf{u},i} = \mathbb{P}\left( \sum_{k=1}^K \mathbb{1}[\widehat{\boldsymbol{\mu}}_i \ne \boldsymbol{\mu}_i^*] > \frac{K}{2} \right).
$$

With Lemma D.6 and Eq. D.9, letting $\epsilon_P^2 = 1/K$, we can obtain:

$$
p_{\mathbf{u},i} + p_{\mathbf{v},i} \ge \frac{1}{2} \exp\left( -16 - N\left( 16 d_R^2 \epsilon_\pi^2 + \frac{16}{K} \right) \right).
$$

By averaging over the parameter space $\mathcal{U}$, it implies:

$$
\sum_{i=1}^{d_P} p_{\mathbf{u},i} \ge \frac{d_P}{4} \exp\left( -16 - N\left( 16 d_R^2 \epsilon_\pi^2 + \frac{16}{K} \right) \right).
$$

Therefore, we can obtain:

$$
\sum_{i=1}^{d_P} \mathbb{P}\left( \sum_{k=1}^K \mathbb{1}[\widehat{\boldsymbol{\mu}}_i \ne \boldsymbol{\mu}_i^*] > \frac{K}{2} \right) \ge \frac{d_P}{4} \exp\left( -16 - N\left( 16 d_R^2 \epsilon_\pi^2 + \frac{16}{K} \right) \right),
$$

which concludes the proof.

$\square$

## D.6 PROOF OF THEOREM 5.8

*Proof of Theorem 5.8.* Let the probability of learning an incorrect reward be $\mathbb{P}(\boldsymbol{\theta}^* \ne \widehat{\boldsymbol{\theta}}) = 1/2$, from Eq. D.8, for $d_R > 48$, it yields:

$$
N \ge \frac{1}{16 \epsilon_\pi^2} \left( \frac{1}{2} - \left( 1 + \frac{16}{d_R - 48} \right) \log 2 \right) \simeq \Omega(1/\epsilon_\pi^2).
$$

The suboptimality gap is lower bounded:

$$\max_{r \in \mathcal{R}} \mathbb{E}_{\mathbf{s}}[V_{1;P^\star;r}^{\pi^E}(\mathbf{s}) - V_{1;P^\star;r}^{\pi^k}(\mathbf{s})] \geq \tfrac{1}{2}\epsilon_\pi \tau \mathbb{P}(\boldsymbol{\theta}^* \neq \widehat{\boldsymbol{\theta}}),$$

which suggest that $\Omega(\tau^2/\epsilon_\pi^2)$ is the lower bound for $N$.

By Lemma D.7, Lemma D.9 and summing the suboptimality over the iteration, letting $\epsilon_P = 1/\sqrt{K}$ the regret of the imitation learning algorithm can be lower bounded as:

$$\mathrm{Regret}(K) = \max_{r \in \mathcal{R}} \sum_{k=1}^{K} \mathbb{E}_{\mathbf{s}}[V_{1;P^\star;r}^{\pi^E}(\mathbf{s}) - V_{1;P^\star;r}^{\pi^k}(\mathbf{s})]$$

$$\geq \frac{1}{2\sqrt{K}} \sum_{k=1}^{K} \tau \epsilon_\pi d_R \|\boldsymbol{\mu}^* - \widehat{\boldsymbol{\mu}}\|_1$$

$$\geq \frac{1}{2\sqrt{K}} \sum_{k=1}^{K} \sum_{i=1}^{d_P} \tau \epsilon_\pi d_R \, \mathbb{1}[\boldsymbol{\mu}_i^* \neq \widehat{\boldsymbol{\mu}}_i]_1$$

$$\geq \frac{\sqrt{K}}{4} \sum_{i=1}^{d_P} \tau \epsilon_\pi d_R \mathbb{P}\left( \sum_{k=1}^{K} \mathbb{1}[\widehat{\boldsymbol{\mu}}_i \neq \boldsymbol{\mu}_i^*] > \frac{K}{2} \right)$$

$$\geq \frac{d_P d_R \tau \epsilon_\pi \sqrt{K}}{16} \exp\left( -16 - N\left(16 d_R^2 \epsilon_\pi^2 + \frac{16}{K}\right) \right)$$

Let $(\epsilon_P, \epsilon_\pi)$ be a problem instance parameterized by two perturbations. For $\epsilon > 0$ and $\max_{r \in \mathcal{R}} \mathbb{E}_{\mathbf{s}}[V_{1;P^\star;r}^{\pi^E}(\mathbf{s}) - V_{1;P^\star;r}^{\pi}(\mathbf{s})] \leq \epsilon$, for two hard instances parameterized by $\epsilon$: $(1/(4d_P), \epsilon)$ and $(\epsilon, 1/(4d_R))$, the lower bound should be able to apply to both instances. For instance $(\epsilon, 1/(4d_R))$, the lower bound for $N$ and $K$ is as follows:

$$N \gtrsim \Omega\left(\tau^2 d_R^2\right), \quad K \gtrsim \Omega(\tau^2 d_P^2 \exp(-N)/\epsilon^2).$$

For instance $(1/(4d_P), \epsilon)$ the lower bound for $N$ and $K$ is as follows:

$$N \gtrsim \Omega\left(\tau^2/\epsilon^2\right), \quad K \gtrsim \Omega(\tau^2 \epsilon^2 d_R^2 d_P^4 \exp(-N/d_P^2)).$$

Combining two lower bounds above, the lower bound must hold for both hard instances:

$$N \gtrsim \Omega\left(\max\left\{\tau^2/\epsilon^2, \tau^2 d_R^2\right\}\right), K \gtrsim \Omega\left(\max\left\{\tau^2 d_P^2 \exp(-N)/\epsilon^2, \tau^2 \epsilon^2 d_R^2 d_P^4 \exp(-N/d_P^2)\right\}\right).$$

Consider the variance of the cumulative rewards:

$$\mathrm{VaR}_{\pi,r} = \mathbb{E}\left[ \left(\sum_{h=1}^{H} r(s_h, a_h)\right)^2 \right] - \left( \mathbb{E}[V_{1;P^\star;r}^{\pi}(s)] \right)^2$$

$$\leq \mathbb{E}\left[ P(\boxed{\mathbf{s},1}|\boxed{\mathbf{s},0}, \mathbf{a}) \tau^2 \mathbb{E}\left( \sum_{h=1}^{H} \frac{1}{2H} + \frac{a_R}{2 d_R H}\langle\widetilde{\boldsymbol{\theta}}, \mathbf{s}\rangle \right)^2 \right]$$

$$= \left( \tfrac{1}{2} + \epsilon_\pi \epsilon_P d_R \langle\widehat{\boldsymbol{\mu}}, \boldsymbol{\mu}^*\rangle \right) \tau^2 \mathbb{E}\left( \tfrac{1}{4} + \frac{a_R}{2 d_R}\langle\widetilde{\boldsymbol{\theta}}, \mathbf{s}\rangle + \frac{1}{4 d_R^2}\langle\widetilde{\boldsymbol{\theta}}, \mathbf{s}\rangle^2 \right),$$

where we use $a_R^2 = 1$ in the last line. Let $\epsilon_P \leq 1/(2 d_P)$ and $\epsilon_\pi \leq 1/d_R$, we have:

$$\mathrm{VaR}_{\pi,r} \leq \tau^2 \mathbb{E}\left( \tfrac{1}{4} + \frac{a_R}{2 d_R}\langle\widetilde{\boldsymbol{\theta}}, \mathbf{s}\rangle + \frac{1}{4 d_R^2}\langle\widetilde{\boldsymbol{\theta}}, \mathbf{s}\rangle^2 \right) \leq \tau^2.$$

Therefore, the lower bound for $N$ and $K$ can be reformulated using the variance of the accumulated rewards:

$$N \gtrsim \Omega\left(\max\left\{\mathrm{VaR}_{\pi,r}/\epsilon^2, \mathrm{VaR}_{\pi,r} d_R^2\right\}\right),$$
$$K \gtrsim \Omega\left(\max\left\{\mathrm{VaR}_{\pi,r} d_P^2 \exp(-N)/\epsilon^2, \mathrm{VaR}_{\pi,r} \epsilon^2 d_R^2 d_P^4 \exp(-N/d_P^2)\right\}\right).$$

Substituting $\mathrm{VaR}_{\pi,r}$ with $\sigma^2$, together with $d_R = \Theta(\log|\mathcal{R}|)$ and $d_P = \Theta(\log|\mathcal{P}|)$, completes the proof. $\qquad\square$

---

**Algorithm 2** Model-based AIL (Practical)

---

**Input:** Number of iterations $K$, discount factor $\gamma$, expert dataset $\mathcal{D}_E$, policy $\pi_\theta$, model ensemble $\{P_{\xi_i}\}_{i=0:D-1}$ with ensemble size $D$ and reward model $r_\psi$.
1: **for** $k = 1, 2, \ldots, K$ **do**
2:      Collect trajectories using current policy $\pi_\theta$ and populate behavioral buffer $\mathcal{B}^\pi$.
3:      Sample behavioral and expert batches $B^\pi, B^E \sim \mathcal{B}^\pi, \mathcal{B}^E$.
4:      Update $r_\psi$ using Eq. E.1.
5:      Update model ensemble $\{P_{\xi_i}\}_{i=0:D-1}$ using $B^\pi \cup B^E$ using Eq. E.2.
6:      Collect $D$ model rollouts $\{\tau_d\}_{d=0:D-1}$ using $\{P_{\xi_i}\}_{i=0:D-1}$ and $\pi_\theta$.
7:      Estimate the value of each rollout $V(\tau_d) = \sum_t \gamma^t r_\psi(s_t)$.
8:      Select the rollout with $\tau^\star = \arg\max V(\tau_d)$ and store it into the model buffer $\mathcal{B}^M$.
9:      Train the RL algorithm on the model buffer $\mathcal{B}^M$ using the reward model $r_\psi$ to update the policy $\pi_\theta$.
10: **end for**

---

## E   PRACTICAL IMPLEMENTATION

In this section, we present a practical implementation of MB-AIL, an optimization-based approach with neural network parameterizations for the policy $\pi_\theta$, reward $r_\psi$, and transition model ensemble $\{P_{\xi_i}\}_{i=0:D-1}$. For reward learning, the reward model is optimized using the following training objective:

$$\min_\psi \mathcal{L}_R(\psi) := \mathbb{E}_{(s,a)\sim\mathcal{B}^\pi} r_\psi(s, a) - \mathbb{E}_{(s,a)\sim\mathcal{B}^E} r_\psi(s, a) + \alpha\, \phi(r), \tag{E.1}$$

where $\mathcal{B}^\pi$ and $\mathcal{B}^E$ denote the replay buffers for the policy and the expert, respectively. The term $\phi(r)$ is a regularization function, for which we adopt the gradient penalty (Arjovsky et al., 2017). Then for each model $P_{\xi_i}$ in the ensemble $\{P_{\xi_i}\}_{i=0}^{D-1}$, we update it using the maximum likelihood objective conducted on data sampled jointly from the expert and behavioral replay buffer:

$$\min_{\xi_i} \mathcal{L}_P(\xi_i) = -\mathbb{E}_{(s,a,s')\sim\mathcal{B}^E\cup\mathcal{B}^\pi}\ \log P_{\xi_i}(s'|s, a). \tag{E.2}$$

The practical implementation of the algorithm is summarized in Algorithm 2. In Line 9, we employ Soft Actor-Critic (SAC) Haarnoja et al. (2018) as the RL algorithm.

## F   GRIDWORLD EXPERIMENTS

**Environment Setup** We conduct a toy experiment on a $9 \times 9$ GridWorld environment, illustrated in Figure 3. Rewards of value 1 are available only in the top-right corner of the grid. The state space corresponds to the $9 \times 9$ grid, and the action space is $\mathcal{A} = \{\text{up}, \text{down}, \text{left}, \text{right}\}$. The environment dynamics are stochastic: when an action is taken (e.g., *up*), the agent transitions to the intended neighboring cell with probability $p$; each of the other three neighboring cells is selected with probability $\frac{1}{3}(1 - p)$. Each episode lasts 20 steps, and performance is measured by the accumulated reward. We use a single expert trajectory with a total reward of 8.0 across all experiments. We initialize the agent in the area $\{(0, 0), (0, 1), (1, 0), (1, 1)\}$.

We evaluate behavioral cloning (offline) and online imitation learning in this environment in order to analyze the benefits of online exploration in the context of imitation learning. Both BC and online IL agents do not observe rewards directly but have access to the expert trajectory. For BC, the agent mimics expert actions in states encountered in the trajectory, and acts randomly elsewhere. For online IL, we maintain both a Q-table and a reward table, initialized with zeros. The Q-learning agent explores using $\epsilon$-greedy with actions chosen as $a = \arg\max Q(s, a)$. Whenever the agent encounters a state-action pair from the expert trajectory, it assigns $r(s, a) = 1$ in its reward table; otherwise, the reward is set to 0. The Q-table is updated from visited experiences using this reward table.

**Experiment Settings** We evaluate two settings in the GridWorld experiment. The first examines the effect of reward space size. Specifically, we partition the reward table into $n \times n$ regions, assigning the same reward value within each region. Thus, the reward space size scales as $|\mathcal{R}| = (\lceil N/n \rceil)^2$, where $N = 9$. We fix $p = 0.65$ and test $n = 1, 2, 3, 4$ to study the impact of varying reward space sizes.

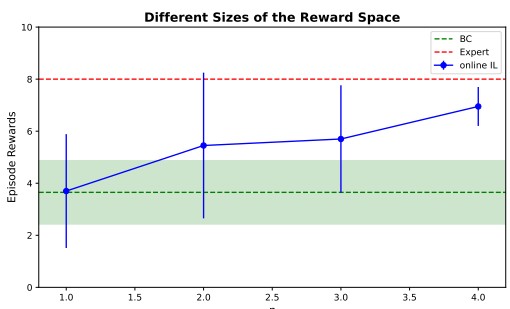 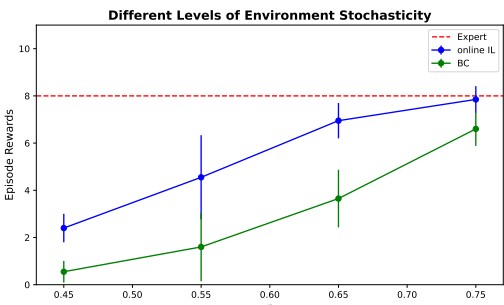

Figure 2: **Results for GridWorld Experiments.** We present results analyzing the impact of varying reward space sizes and different levels of environment stochasticity on adversarial imitation learning and behavioral cloning, reported over 5 random seeds.

The second setting investigates the effect of environment stochasticity on imitation learning performance. We fix $n = 4$ and control stochasticity by adjusting the transition probability $p$: a larger $p$ corresponds to a more deterministic environment. We experiment with $p = 0.45, 0.55, 0.65, 0.75$ to analyze this effect.

**Main Results** We present the empirical results of the two experiment settings in Figure 2. For each configuration, we evaluate 10 episodes for each of the 5 random seeds and report the mean and standard deviation. In the reward space experiment, when the reward space is large, online IL performs similarly to BC. However, as the reward space becomes smaller (larger $n$), online IL achieves significantly better performance, approaching the optimal value, which is consistent with our theoretical upper bound in Theorem 5.1. In the stochasticity experiment,

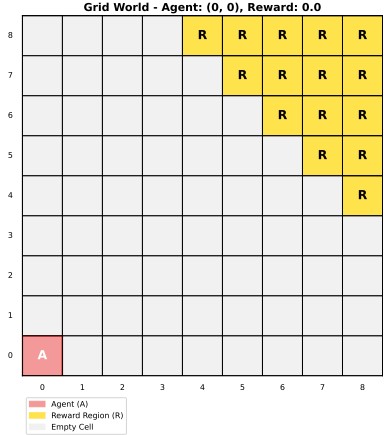

Figure 3: **GridWorld Illustration.** An illustration of the $9 \times 9$ GridWorld with the agent initialized at $(0, 0)$ is shown.

both online IL and BC improve as the environment becomes more deterministic. Across all stochasticity levels, online IL consistently outperforms BC.

**Results on the Model Misspecification** In order to evaluate the behavior of online IL when Assumption 3.5 is mildly violated, we conduct an experiments to evaluate the impact of model misspecification. Specifically, we randomly sample $m$ states in GridWorld as misspecified states. During Bellman updates, whenever a transition leads to one of these misspecified states, we manually overwrite the next state as $(0, 0)$, thereby introducing controlled transition model misspecification. The results show that the algorithm maintains strong performance when the proportion of misspecified states is small, but its performance degrades as the degree of misspecification increases. The detailed results are shown in Figure 4.

## G    DETAILS OF THE MUJOCO EXPERIMENTS

**Hyperparameter Setting** We summarize the hyperparameter settings used in our experiments below:

- Model ensemble size is 7 for all tasks.
- Model optimization: learning rate $3 \times 10^{-4}$, weight decay $5 \times 10^{-5}$, batch size 256.
- Policy parameterization: stochastic Gaussian policy.
- Discriminator: ensemble size 7, trained with batch size 4096 and learning rate $8 \times 10^{-4}$.
- SAC training: learning rate $3 \times 10^{-4}$, batch size 256, entropy coefficient $\alpha = 0.2$ for all tasks.

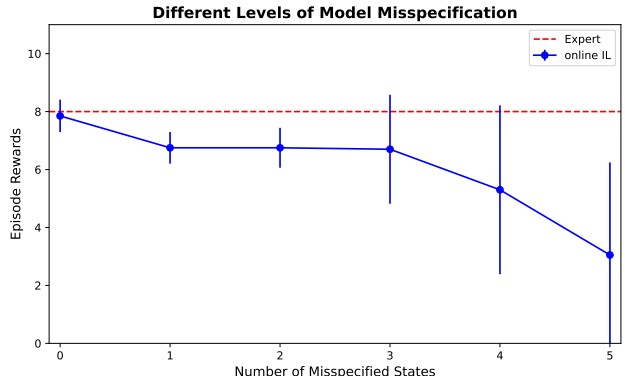

Figure 4: **Results on the Model Misspecification** We show the performance of online imitation learning when some of the states in the GridWorld will lead to misspecification. Results show that the algorithm maintains strong performance when the proportion of misspecified states is small, but its performance degrades as the degree of misspecification increases. The results are reported using 5 random seeds.

- Network architecture: hidden size 400 for the model network, and 512 for both the discriminator and policy networks.

**Environment Details** The specifications of the environments are summarized in Table 6.

| Environment | Observation Dimension | Action Dimension |
|---|---|---|
| Hopper | 11 | 3 |
| Walker2d | 17 | 6 |
| Humanoid | 45 | 17 |

Table 6: **Environment Details.** Observation and action dimensions for each environment. The Humanoid task features a higher-dimensional state and action space, making it significantly more challenging for imitation learning compared to the lower-dimensional Hopper and Walker2d tasks.

**Baseline Methods** For the OPT-AIL baseline (Xu et al., 2024), we adopt the official implementation provided in their repository. For the GAIL baseline (Ho & Ermon, 2016), we use the open-source implementation available at this repository. For the BC baseline, we train a two-layer MLP with a hidden dimension of 256 on the expert dataset for direct behavioral cloning.

## USE OF LARGE LANGUAGE MODELS

We used LLMs solely as a writing assistant for minor grammar and phrasing corrections during manuscript preparation. LLMs were not involved in research ideation, experiment design, data analysis, or result interpretation.

