# OpenReview forum: "Near-Optimal Second-Order Guarantees for Model-Based Adversarial Imitation Learning"
_ICLR.cc/2026/Conference — ICLR 2026 Poster_

### Official Review · Reviewer_PeTQ · 2025-10-15

**Soundness:** 1
**Presentation:** 3
**Contribution:** 3
**Rating:** 2
**Confidence:** 3

**Summary:**

The authors derive second order sample complexity bound in the context of imitation learning where the learner can collect reward free trajectories.
As a result the expert samples and MDP interaction complexity only scales with the maximum reward variance where the maximum is taken over both the reward and policy class and not on the horizon. These are very interesting results.

My low score at the moment is because there seems to me that there is a contradiction between the lower bound in theorem 5.9 and the BC upper bound, which scales only with the expert variance and not with the maximum variance over the policy and reward class.

**Strengths:**

The analysis relies on elegant techniques and it looks correct to me.

The results are important for the imitation learning literature.

The experiments are convincing.

**Weaknesses:**

Here are my main suggestions on how to improve the presentation of your work. I think that the most important one is weakness 5 since it identifies a potential technical flow. I am willing to raise my score significantly once I am sure that there is no technical issue there.

1) In Remark 5.6, the authors discuss that in environments where the expert is complex ( ie Pi contains many functions) than the BC bound in Foster et al 2024 becomes suboptimal compared to MB-AIL, claiming in this way that online interactions are beneficial. However, even in the offline case, it is possible to obtain bounds that are independent of $\Pi$ but that depend on the covering number of the $Q$ value class (see [1]). I suggest that the authors add a comparison with this result, and they suggest that it is reasonable to expect that $\abs{\mathcal{R}} \leq \abs{\mathcal{Q}}$ because for a fixed reward function, I can obtain many different $Q$ functions changing policy and dynamics.

2) In the context of Linear MDP, algorithms such as BRIG or FRA-IL[2] can work when observing only the reward features visited by the expert and not the chosen actions. This setting captures as an important example of state-only imitation learning. Similarly, in the context of general function approximation considered by your work, I think that it's enough to observe all the reward functions in the class $\mathcal{R}$ evaluated at the expert state action pairs. Observing the actions seems unnecessary. Pointing this out makes the work even stronger because it would make your algorithm applicable when BC is not (because BC needs to observe the actions).

3) I found Remark 5.5 a bit odd because if expert and dynamics are deterministic, then trivially only one trajectory is needed. Therefore, $1/\epsilon$ bound in this setting is disappointing. Noticed that Foster et al. 2024 attained $\mathcal{O}(1/\epsilon)$ assuming only a deterministic expert but not transitions. Also in Table 2, for the case deterministic expert and deterministic transitions (right bottom cell ), I think that one trajectory is enough for BC because simply copying the actions can guarantee remaining with probability one on the support of states visited by the expert. Can the same result be proven for MB-AIL ?

4) The expert sample complexity of MB-TAIL scales with the maximum variance over the policy class while the BC bound scales with the maximum variance in class. Can you improve the bound to match the variance dependent bound in BC ? Please fix this in Table 1 where this difference is not visible.

5) This also seems at odd with the lower bound in Theorem 5.9. The authors there state that any online or offline algorithm needs to pay the maximum variance in class. How is this possible considering that the bound for BC scales only with the expert variance. Moreover, how is it possible to prove a lower bound on $N$ for any offline algorithm since by definition an offline imitation learning algorithm uses $N=0$?

[1] Moulin et al. 2025, Inverse Q Learning Done Right: Offline Imitation Learning in $Q^\pi$ realizable MDPs
[2] Moulin et al. 2025, Optimistically Optimistic Exploration for Provably Efficient Infinite-Horizon Reinforcement and Imitation Learning

Minor

At line 358, I think it should $\Pi$ and not $\mathcal{R}$.

**Questions:**

The lower bound seems related to the construction used in [2] Section 5.3. Could you elaborate on the differences?

Related to weakness 2. Does MB-AIL need to observe the actions or only the values of the reward functions in $\mathcal{R}$ evaluated at the state actions in the expert dataset?

 In the algorithm, how to solve efficiently the problem at Step 7 where the maximization is over both policies and transition dynamics in the confidence set. Are you planning to use extended value iteration? Can this be done in a more computationally efficient way using corresponding exploration bonuses?

Please answer also to my questions raised in the Weaknesses Section.

---

> ### Author Response · Authors · 2025-11-19
>
> Thank you for your the constructive feedback and would like to clarify potential misunderstandings in this paper. Our lower bound results do **not** contradict the existing upper bound results for BC and it's **impossible** to conduct behavior cloning with only  one trajectory even under determinstic transition kernel. We address the your concern in detail below:
>
> **Q1.** Why the lower bound scales with the maximum variance but the upper bounds scales with the expert variance?
>
> **A1.** We would like to highlight that **there is no contradiction** between the maximum variance and the expert variance in the results. This is simply because the expert variance $\sigma_E^2$ and the maximum variance is essentially in the same order according to our hard-case construction. To be specific, by the definition of the variance we have
>
>
> $$\sigma^2_E=\max_r VaR_{\pi^E,r}=\max_r\mathbb{E}\left[\Big(\sum_{h=1}^H r(s_h, a_h)\Big)^2\right] - \Big(\mathbb{E}[\,V_{1;P^\star;r}^{\pi^E}(s)\,]\Big)^2,
> $$
> where by Eq. D.3 in the manuscript appendix and by the definition of the reward in Appendix D.1, let the parameterization of the reward be $\tilde\theta$ and the expert policy be $\pi(a|s;\mu^\star,\theta^\star)$, we have
> $$
> \mathbb{E}\left[\Big(\sum_{h=1}^H r(s_h, a_h)\Big)^2\right]=\left(\frac{1}{2}+\epsilon_\pi\epsilon_P d_R\langle\mu^\star,\mu^\star\rangle\right)\tau^2\mathbb{E}\left(\frac{1}{4}+\frac{a_R}{2d_R}\langle\tilde\theta,s\rangle+\frac{1}{4d^2_R}\langle\tilde\theta,s\rangle^2\right),
> $$
>
> $$
> \Big(\mathbb{E}[\,V_{1;P^\star;r}^\pi(s)\,]\Big)^2=\Big[\left(\frac{1}{2} + {\epsilon_\pi\epsilon_Pd_R\langle \mu^\star, \mu^\star\rangle}\right) \cdot \left(\frac{1}{2}\tau + \frac{\epsilon_\pi\tau}{d_R}\langle \tilde \theta, s \rangle \langle \theta^\star, s \rangle\right)\Big]^2.
> $$
> By straightforward algebraic manipulation (see Appendix B of the revised manuscript), we show that the expert variance $\sigma^2_E$ admits a lower bound that depends on the same parameter $\tau$.
>
> Therefore, lower bounding the sampling complexity using the maximum variance over **all policies** leads to the same order as the variance of the **expert policy** in our setting, and there is no contradiction with the results in Foster et. al. [5]. We have added this into Appendix B for a detailed verificaiton.
>
> ---
>
>
>
> **Q2.** Add the comparison with [1] depending on the $Q$ function class?
>
> **A2.** Thank you for pointing out that literature. We have added the comparison into related works. In particular, as you acknowledged, we would expect $|Q| \approx |\Pi|$ since they are all generated by the cross-product of reward class $R$ and transition kernel $P$. Therefore the contribution of our results remains unchanged since it suggests that the expert demonstrations can be reduced with online interaction when $|R| \le |Q|$.
>
> ---
>
> **Q3.** How is the results being applied to the state-only imitation learning setting such as BRIG and FRA-IL [2]?
>
> **A3.** Thanks for pointing out this state-only imitation learning setting. We have added these works as additional related works and offered discussions in Section 2. We would like to highlight that though applying the current techniques to the state-only imitation learning when the reward $R$ only depends on the state is possible, the focus of this paper is still to understand the online interaction for regular imitation learning setting. We view this combination and extension as a primising direction for future research.
>
> ---
>
>
>  **Q4.** Could we use one trajectory for BC when the transition kernel is determinstic?
>
>  **A4.** We would like to clarify that even both the policy and transition kernel are determinstic, it is **impossible** to learn from **one trajectory** because of the two following reasons. **First**, as proved in Theorem 2.2 (Lower bound for deterministic experts) in [5], for *any* horizon $H \in \mathbb{N}$, ..., the suboptimality gap is lower bounded by $\mathbb{E}[J(\pi^\star)-J(\pi)]\geq c\cdot\tfrac{H}{N}$. Obviously, this result applies to the case with $H = 1$, where there is *no transition kernel*, the required sample complexity is still $\Omega(\tfrac{1}{\epsilon})$. **Second**, the $\Omega(\tfrac{1}{\epsilon})$ barrier comes from the randomness of initial state, where only one trajectory starting from $s$ will be not enough for learning the policies from other initial state $s'$. Such a hardness can also be validated by the construction of [5] (see Proof of Theorem G.1, pp.58, [5]). We would like to further highlight that even in their $\Omega(\tfrac{1}{\epsilon})$ proof the transition kernel is also determinstic i.e. $P(s' | s, a) = 1[s' = s]$. In conclusion, we emphesis that this $\Omega(\tfrac{1}{\epsilon})$ can **not** be lifted by the determinstic transition kernel and we respectfully disagree your one-trajectory hypothesis.

---

> ### Author Response · Authors · 2025-11-19
>
> **Q5.** Can we obtain a second-order bound matches the expert variance in BC? Also highlight this in Table 1.
>
> **A5.** We would like to clarify that our upper bound for the *expert demonstrations*, as shown in Corollary 5.2, is indeed scales the variance of the expert policy, which **matches** with the variance term in [5]. We are not aware of any results providing similar results for online interaction that only depends on the variance of the expert policy. We hypothesis that this second-order expert-dependent results is hard to obtain since we found that [4] also depends on $\text{VaR} _{\pi^K}$ instead of $\text{VaR} _{\pi^*}$. In Table 1 we have reflected this results and it's clear that the expert sample complexity for both ours and [5] depends on the variance of the expert policy.
>
>
> ---
>
> **Q6.** How will the lower bound be extended to the purely offline setting?
>
> **A6.** Thank you for the thoughtful comment. We first emphasize that $K$ indexes the policies (number of iterations), whereas the amount of real data used corresponds to $K - 1$, since $\pi_K$ is learned from the offline dataset together with the previous $K - 1$ rounds of interaction. Although $K$ and $K - 1$ are of the same order when $K$ is large, the purely offline setting corresponds to $K = 1$, not $K = 0$. We then restate the conclusion of Theorem 5.9: to achieve an $\epsilon$-suboptimality gap
> $$
> \text{SubOpt}(\pi^{K})
> = \max_{r \in R} \mathbb{E}\left[ V_{1;P^\star;r}^{\pi^E}(s) - V_{1;P^\star;r}^{\pi^K}(s) \right],
> $$
> our theorem establishes that when $\epsilon$ is sufficiently small, one needs at least $N = \Omega(\sigma^2 \epsilon^{-2})$ expert demonstrations and $K = \Omega\left( \sigma^2 \log^2|\mathcal{P}| \exp(-N)\epsilon^{-2} \right)$
> online interaction episodes. A key observation is that the lower bound on $K$ contains an $\exp(-N)$ factor. When $N = \Theta(\sigma^2 \epsilon^{-2})$ and $\epsilon$ is small, we have $\sigma^2 \log^2|\mathcal{P}| \exp(-N)\epsilon^{-2} = o(1),$ so the lower bound is satisfied with $K = 1$, which corresponds to **zero** online interactions.
>
> Thus, our theorem implies a lower bound of $\Omega(\sigma^2 \epsilon^{-2})$ expert demonstrations and zero interactions, which is fully consistent with Foster et al. (2024) [5].  We have updated this remark to Appendix B.
>
> ---
>
> **Q7.** How does the lower bound construction differs from [2]?
>
> **A7.** We would like to highlight that our lower bound construction differs fundamentally from [2] in many aspects, to the best of our understanding. First, construction in [2] and ours is that [2] employs a unit-vector representation for the state embeddings, whereas we use a binary-vector construction. In addition, their MDP consists of two states, while our construction involves three states.
>
> ---
>
>
> **Q8.** How to efficiently solve Line 7 in the algorithm practically?
>
>
> **A8.** In the practical implementation of our algorithm, we approximate the version-space construction using a model ensemble, following a strategy also adopted in prior work [3]. For policy updates, we select model rollouts optimistically by choosing those that yield the highest estimated value. The computationally efficient practical variant of our method is provided in Algorithm 2 of the original manuscript.
>
> [1] Moulin et al. 2025, Inverse Q Learning Done Right: Offline Imitation Learning in $Q^\pi$ realizable MDPs
>
> [2] Moulin et al. 2025, Optimistically Optimistic Exploration for Provably Efficient Infinite-Horizon Reinforcement and Imitation Learning
>
> [3] Ye, C., Yang, R., Gu, Q., & Zhang, T. (2023). Corruption-robust offline reinforcement learning with general function approximation. Advances in Neural Information Processing Systems, 36, 36208-36221.
>
> [4] Wang, Z., Zhou, D., Lui, J., & Sun, W. (2024). Model-based rl as a minimalist approach to horizon-free and second-order bounds. arXiv preprint arXiv:2408.08994.
>
> [5] Foster, D. J., Block, A., & Misra, D. (2024). Is behavior cloning all you need? understanding horizon in imitation learning. Advances in Neural Information Processing Systems, 37, 120602-120666.

---

> > ### Comment · Reviewer_PeTQ · 2025-11-20
> >
> > Thanks a lot for your comments!
> >
> > My impression of the paper improved a lot !
> > Thanks in particular to explain the importance of the factor exp(-N) in the lower bound. This is a very interesting result!
> >
> > However, I think that the lower bound result is reported in a misleading way in Table 1 since the factor exp(-N) does not appear there.
> >
> > Moreover, I would like the authors to state the lower bound in terms of expert variance. I understand that in your construction learner and expert variance are of the same order but this does not exclude the existence of another instance where the maximum possible variance is much larger than the expert one.
> >
> > Once these changes is done I will increase my score!
> >
> > Finally sorry to misinterpret your deterministic case , I missed that the initial state could still be stochastic,

---

> > > ### Author Response · Authors · 2025-11-20
> > >
> > > Thank you for your positive feedback and we are glad to see our response have resolved your concerns and questions.
> > >
> > > In response to your suggestion. We have updated the papers following these two points:
> > >
> > > **Q9.** Adding the $\exp(-N)$ factor into Table 1.
> > >
> > > **A9.** We have added the $\exp(-N)$ factor into Table 1 with a footnote saying $N$ can be small (which would be enough to learn the reward when the reward class $\mathcal R$ is simple). So the order on $\sigma^2\epsilon^{-2}$ remains unchanged.
> > >
> > > ---
> > >
> > > **Q10.** Presenting the lower bound in terms of expert variance $\sigma_E^2$.
> > >
> > > **A10.** We have revised the presentation of Theorem 5.8 as well as Table 1 to present the lower bound for the expert demonstrations in $\sigma_E^2$. To reflect the message that $\sigma_E$ and $\sigma$ are esentailly the same order, in the presentation of Theorem 5.8, we sandwiched it by $\tfrac{1}{2}\sigma^2 \le \sigma_E^2 \le \sigma^2$ as we have discussed in **A1** in our previous response to avoid further confusion. In addition, we choose to still present the number of iterations $K$ with $\sigma^2$ since the online trajectories are not collected by the expert policy. Despite this, we do agree that writing it as $\sigma_E^2$ is still technically correct since $\sigma_E^2$ and, $\sigma^2$ are in the same order.
> > >
> > > We sincerely thank you again for your suggestions and please let us know if furtuer revisions are needed.

---

> > > > ### Comment · Reviewer_PeTQ · 2025-11-20
> > > >
> > > > Thanks a lot for addressing my comments!
> > > >
> > > > I am happy to increase my score!
> > > >
> > > > Best,
> > > > Reviewer

---

> > > > > ### Author Response · Authors · 2025-11-20
> > > > >
> > > > > We sincerely appreciate your kind review and all the suggestions.
> > > > >
> > > > > Best regards,
> > > > > Authors

---

### Official Review · Reviewer_J4QT · 2025-10-29

**Soundness:** 4
**Presentation:** 3
**Contribution:** 3
**Rating:** 8
**Confidence:** 3

**Summary:**

This work takes a theoretical perspective on adversarial imitation learning (AIL), providing results that clarify the benefits of leveraging online interactions and the role of stochasticity in both the expert policy and the environment’s transition dynamics.

**Strengths:**

This is a well-presented and solid theoretical contribution to AIL. The paper formulates an interesting in-depth analysis, leading to theoretical results that, to my knowledge, were not previously established in the literature.

**Weaknesses:**

1. It is not fully clear what new practical insights the theoretical analysis provides for the design of MB-AIL. While the bounds are technically interesting, the paper does not convincingly articulate how these results translate into concrete algorithmic guidance or practical improvements.

2. The presented algorithm MB-AIL is derived by decomposing the regret into the error of reward and error of the model. Other studies, not mentioned in the Related Work, have previously leveraged this division in their analysis. Refer to:

[1] Chen Y, Giammarino V, Queeney J, Paschalidis IC. Provably efficient off-policy adversarial imitation learning with convergence guarantees. arXiv preprint arXiv:2405.16668. 2024 May 26.

[2] Shani L, Zahavy T, Mannor S. Online apprenticeship learning. InProceedings of the AAAI conference on artificial intelligence 2022 Jun 28 (Vol. 36, No. 8, pp. 8240-8248).

Minor:

A few typos and grammar errors: “stragety” in Line 71, “as an practical example” Line 277, “Despite of this difference” in Line 282.

**Questions:**

1. What new practical insights do the theoretical results provide for the design of MB-AIL? Are there concrete guidelines that practitioners can draw from the analysis?

---

> ### Author Response · Authors · 2025-11-19
>
> Thank you for your supportive feedback. In the following we addressed your questions and concerns.
>
> **Q1.** How is the theoretical insights can be translated into algorithmic guidance or practical improvements.
>
>
> **A1.** We highlight several practical insights from our analysis:
>
> 1. **Benefit of online interactions:** As shown in Theorem 5.9, when the number of expert demonstrations is limited, online interactions improve imitation learning performance compared to purely offline behavioral cloning.
>
> 2. **Policy determinism vs. transition stochasticity:** As revealed in Table 2, when the policy is highly deterministic but the transition dynamics are highly stochastic, skipping transition model learning and performing policy cloning may outperform model-based approaches. In most other scenarios, model-based approaches are preferable.
>
> 3. **Advantage of AIL over BC in complex policies:** As discussed in Remark 5.6, adversarial imitation learning (AIL) is advantageous when rewards are simple but the policy class is complex. This is because the upper bound of AIL depends on the reward class $\mathcal{R}$ rather than the policy class $\Pi$, whereas behavioral cloning (BC) depends on $\Pi$.
>
> In addition, as we presented in Appendix E, MB-AIL with these guidance does improve the performance compared with the current models.
>
> ---
>
> **Q2.** Discussions on related works [1, 2]
>
>
> **A2.** We thank the reviewer for pointing out. In particular, [2] provide an $O(\sqrt{K})$ regret upper bound for on-policy AIL, while [1] provide a convergence guarantee for off-policy AIL in tabular MDPs. We have added that in the revision (see sentences in Section 2 highlighted in blue).
>
> [1] Chen Y, Giammarino V, Queeney J, Paschalidis IC. Provably efficient off-policy adversarial imitation learning with convergence guarantees. arXiv preprint arXiv:2405.16668. 2024 May 26.
>
> [2] Shani L, Zahavy T, Mannor S. Online apprenticeship learning. InProceedings of the AAAI conference on artificial intelligence 2022 Jun 28 (Vol. 36, No. 8, pp. 8240-8248).

---

### Official Review · Reviewer_7CkW · 2025-11-01

**Soundness:** 2
**Presentation:** 3
**Contribution:** 2
**Rating:** 4
**Confidence:** 4

**Summary:**

The paper studies how an agent can learn to imitate expert behavior using both expert demonstrations and online interactions, without access to reward signals. The authors propose a new algorithm called Model-Based Adversarial Imitation Learning (MB-AIL). It combines model-based reinforcement learning with adversarial imitation learning by separately learning a reward function from expert data and a world model from online interactions.

The paper provides strong theoretical results. It proves that MB-AIL achieves horizon-free, second-order sample complexity bounds, meaning its efficiency depends on the variance of returns and improves in more deterministic settings. It also establishes information-theoretic lower bounds showing that MB-AIL is nearly minimax-optimal—it uses about as few samples as theoretically possible. Experiments on MuJoCo and GridWorld tasks confirm the theory, showing that MB-AIL matches or surpasses some imitation learning methods in sample efficiency.

**Strengths:**

1. The paper derives horizon-free, second-order upper bounds for adversarial imitation learning under general function approximation. This is technically meaningful and advances the theoretical understanding of how stochasticity affects AIL.
2. The idea of learning transition models in AIL is natural and well-motivated. Splitting the learning into adversarial reward estimation + model estimation + optimistic planning is elegant and leverages the strengths of model-based RL in a principled way.

**Weaknesses:**

1. As stated in the introduction, the paper's main goal is to provide a tight characterization of the benefits of online interaction in imitation learning. However, the result is unsatisfying and does not clarify when adversarial imitation learning outperforms BC. The paper claims that online interactions are helpful only when $\log (N_{R}) < \log (|\Pi|)$. In practice, however, it is difficult to compare the complexity of the policy class and the reward class, which are typically neural networks.
2. The lower bound result (Theorem 5.9) appears to be incorrect. If I understand correctly, Theorem 5.9 claims that achieving an approximately optimal policy requires both $\Omega (\sigma^2 \epsilon^{-2})$ expert demonstrations and $\Omega\left(\sigma^2 \cdot \log ^2|\mathcal{P}| \exp (-N) \epsilon^{-2}\right)$ online interactions. However, as shown in Table 1, [Foster et al., 2024] proved that BC requires $\Omega (\sigma^2 \epsilon^{-2})$ expert demonstrations but zero online interactions, which conflicts with Theorem 5.9.
3. The paper provides a practical version of MB-AIL, but this version differs significantly from the original algorithm, creating a notable gap between theory and practice. The original algorithm constructs a version space over transition models and solves a joint optimization problem over both the policy and transition model. The practical version ignores this design entirely—it simply learns transition models using maximum likelihood estimation and then runs RL on the learned model. From this view, such a model-based adversarial imitation learning has been proposed in [1].
4. The experimental validation is insufficient to substantiate the paper’s claims that MB-AIL matches or surpasses the sample efficiency of existing methods. The empirical study is limited to a single toy GridWorld environment and three MuJoCo tasks, which do not provide enough diversity or complexity to convincingly demonstrate the generality of the proposed approach. Moreover, the comparisons are restricted to only two adversarial imitation learning baselines, omitting several important and state-of-the-art methods (e.g., [1, 2]). Including these stronger baselines is essential for a fair and comprehensive evaluation. Without broader empirical coverage and more competitive baselines, the experimental evidence remains too weak to support the strong performance claims made in the paper.

References:

[1] Juntao Ren, Gokul Swamy, Zhiwei Steven Wu, J Andrew Bagnell, and Sanjiban Choudhury. Hybrid inverse reinforcement learning.

[2] Gokul Swamy, David Wu, Sanjiban Choudhury, Drew Bagnell, and Steven Wu. Inverse reinforcement learning without reinforcement learning.

**Questions:**

1. The analysis assumes that the true reward function, transition dynamics, and optimal policy are contained within the function classes (Assumption 3.5). How sensitive is MB-AIL to mild violations of this assumption?
2. Remark 5.6 states that AIL is better than BC when $\log (N_{R}) < \log (|\Pi|)$. Could the authors empirically validate this claim by controlling the complexity of the policy networks and reward networks?
3. The established theory shows that the variance term plays a key role in AIL. However, the experiments do not validate the role of variance in theory. Could the authors show experiments where the environment or policy stochasticity is systematically varied?

---

> ### Author Response · Authors · 2025-11-19
>
> Thank you for your constructive feedback. We address your concerns below:
>
>
> **Q1.** How to compare the size of the reward class $\log(N_R)$ and policy class $\log(|\Pi|)$？
>
> **A1.** We would like to emphasize that in most practical scenarios, the reward class is significantly smaller than the policy class. In RL for LLM post-training, prior work has made a similar assumption (see Hypothesis 6 in [4]). Likewise, in MuJoCo environments [5], rewards are typically modeled as linear combinations of a few physical parameters (e.g., velocity, position), which is intuitively much simpler than the policy class.
>
> ---
>
>
> **Q2.** The lower bound proposed in Theorem 5.9 seems contradictive to Foster et al., 2024's result, which has shown that $O(\sigma^2 \epsilon^{-2})$ number of expert demonstrations and 0 number of online interactions is enough to obtain a $\epsilon$-optimal policy.
>
>
> **A2.** Thank you for the thoughtful comment. We first emphasize that $K$ indexes the policies (number of iterations), whereas the amount of real data used corresponds to $K - 1$, since $\pi_K$ is learned from the offline dataset together with the previous $K - 1$ rounds of interaction. Although $K$ and $K - 1$ are of the same order when $K$ is large, the purely offline setting corresponds to $K = 1$, not $K = 0$. We then restate the conclusion of Theorem 5.9: to achieve an $\epsilon$-suboptimality gap
> $$
> \text{SubOpt}(\pi^{K})
> = \max_{r \in R} \mathbb{E}[ V_{1;P^\star;r}^{\pi^E}(s) - V_{1;P^\star;r}^{\pi^{K}}(s) ],
> $$
> our theorem establishes that when $\epsilon$ is sufficiently small, one needs at least $N = \Omega(\sigma^2 \epsilon^{-2})$ expert demonstrations and $K = \Omega\left( \sigma^2 \log^2|\mathcal{P}| \exp(-N)\, \epsilon^{-2} \right)$
> online interaction episodes. A key observation is that the lower bound on $K$ contains an $\exp(-N)$ factor. When $N = \Theta(\sigma^2 \epsilon^{-2})$ and $\epsilon$ is small, we have $\sigma^2 \log^2|\mathcal{P}| \exp(-N)\, \epsilon^{-2} = O(1),$ so the lower bound is satisfied with $K = 1$, which corresponds to **zero** online interactions (since $\pi^1$ only depends on the offline expert demonstrations)
>
> Thus, our theorem implies a lower bound of $\Omega(\sigma^2 \epsilon^{-2})$ expert demonstrations and zero interactions, which is fully consistent with Foster et al. (2024) [6]. We have updated this remark to Appendix B.
>
>
>
>
>  ---
>  **Q3.** How is the practical implementation of the algorithm and how does it compare with [1]
>
>  **A3.** In our practical implementation, we use a model ensemble to approximate the version space construction, a practice also adopted in previous works [3]. The comparison between our approach and the model-based version in [1] can be summarized in two aspects:
>
> 1. **Update scheme:** We perform off-policy updates on both the reward and transition models. In contrast, [1] updates the reward and transition models using no-regret learning on the expert samples and the samples from the *current* policy $\pi_t$, which is on-policy.
>
> 2. **Optimistic update:** We leverage a model ensemble and select the model rollout that maximizes the value for policy updates, introducing an optimistic way for updates.
>
> In conclusion, we believe our practical implementation is aligned with the theoretical justifications based on existing literatures and the methods are different with the implementation in [1].
>
>
>
> ---
>
> **Q4.** Could the authors add more experiments especailly the comparison with [1, 2]?
>
>
> **A4.** We include additional comparison results with existing state-of-the-art methods [1, 2]. The results are summarized below:
>
> **Episode Rewards:**
>
> | Environments | FILTER        | HyPE          | HyPER         | MB-AIL (Ours)   |
> |---------------|---------------|---------------|---------------|-----------------|
> | Hopper        | 3012.2 ± 89.6 | 3396.2 ± 34.4 | 3408.3 ± 18.6 | **3451.3 ± 15.5**   |
> | Walker2D      | 4181.2 ± 23.3 | **4489.5 ± 34.1** | 4043.1 ± 41.2 | 4169.7 ± 48.3   |
> | Humanoid      | 5228.9 ± 45.4 | 5779.4 ± 23.8 | 5804.7 ± 22.1 | **5816.4 ± 15.2**   |
>
> **Sampling Complexity:**
>
> | Environments | FILTER        | HyPE          | HyPER         | MB-AIL (Ours)   |
> |---------------|---------------|---------------|---------------|-----------------|
> | Hopper        | ~520K | ~800K | ~80K | ~60K   |
> | Walker2D      | ~1000K | ~820K | ~100K | ~120K   |
> | Humanoid      | ~550K | ~240K | ~100K | ~90K   |
>
> The results demonstrate that our practical algorithm matches the performance of existing state‑of‑the‑art methods.

---

> ### Author Response · Authors · 2025-11-19
>
> **Q5.** The theoretical results reply on the realizablity assumption. How sensitive is MB-AIL to mild violations of this assumption?
>
> **A5.** This is related to the misspecified situation. In fact, our GridWorld experiment already includes a study on misspecification error, as shown in Appendix E.1 of the original manuscript. Regarding the effect of the reward space size on performance, when the reward space shrinks as $n$ increases (left plot of Figure 2), the ground-truth reward function falls outside the reward class. This occurs because we partition the reward table into $n \times n$ regions and assign a single reward value within each region. Interestingly, the results in Figure 2 show that even under reward misspecification, the algorithm can achieve better performance as the reward class becomes smaller.
>
> Moreover, we conduct additional experiments to evaluate the impact of model misspecification. Specifically, we randomly sample $m$ states in GridWorld as misspecified states. During Bellman updates, whenever a transition leads to one of these misspecified states, we manually overwrite the next state as (0,0), thereby introducing controlled transition model misspecification. The results show that the algorithm maintains strong performance when the proportion of misspecified states is small, but its performance degrades as the degree of misspecification increases:
>
> | Misspecified Portion ($m/N$) | Episode Rewards   |
> |----------------------|-------------------|
> | 0/81                 | 7.85 ± 0.56       |
> | 1/81                 | 6.75 ± 0.55       |
> | 2/81                 | 6.75 ± 0.69       |
> | 3/81                 | 6.70 ± 1.88       |
> | 4/81                 | 5.30 ± 2.91       |
> | 5/81                 | 3.05 ± 3.20       |
>
> The results are reported using 5 random seeds.
>
> **Q6.** Can you validate the relationship between $\log|\Pi|$ and $\log |\mathcal{R}|$ empirically?
>
> **A6.**  In response to your question, we empirically validate this claim by controlling the complexity of the policy networks and reward networks in the following tables:
>
> **Fix policy network to be a 2-layer MLP with 1024 hidden units**
> | Reward Network            | Episode Rewards    |
> |---------------------------|--------------------|
> | 2-layer, 256 hidden units | 4169.7 ± 48.3      |
> | 2-layer, 128 hidden units | 4109.5 ± 39.6      |
> | 1-layer, 256 hidden units | 4125.1 ± 28.2      |
>
> **Fix reward network to be a 2-layer MLP with 256 hidden units**
> | Policy Network             | Episode Rewards    |
> |----------------------------|--------------------|
> | 2-layer, 1024 hidden units | 4169.7 ± 48.3      |
> | 2-layer, 256 hidden units  | 704.9 ± 112.5      |
>
>
> As a result for this ablation study, the reward function can be learned effectively with a smaller network, whereas simplifying the policy network causes a substantial drop in performance. We also report the same results in Appendix H of our revised manuscript.
>
>
> **Q7.** How is the variance-dependent theoretical results being validated empirically
> **A7.** We control the variance of the expert policy (which is a Gaussian policy) manually during expert data sampling and use this expert data to run our MB-AIL algorithm in Walker2D environment. Results indeed show that when the expert variance is increased,  the performance of the algorithm degrades, which matches the theoretical results shown in Theorem 5.1:
>
> | std ($\sigma_E$) | Episode Rewards     |
> |--------------------|----------------------|
> | 0.00               | 4169.7 ± 48.3        |
> | 0.01               | 4088.1 ± 52.2        |
> | 0.1                | 3887.9 ± 135.4       |
> | 0.5                | 1561.2 ± 243.8       |
>
> [1] Juntao Ren, Gokul Swamy, Zhiwei Steven Wu, J Andrew Bagnell, and Sanjiban Choudhury. Hybrid inverse reinforcement learning.
>
> [2] Gokul Swamy, David Wu, Sanjiban Choudhury, Drew Bagnell, and Steven Wu. Inverse reinforcement learning without reinforcement learning.
>
> [3] Ye, C., Yang, R., Gu, Q., & Zhang, T. (2023). Corruption-robust offline reinforcement learning with general function approximation. Advances in Neural Information Processing Systems, 36, 36208-36221.
>
> [4] Swamy, G., Choudhury, S., Sun, W., Wu, Z. S., & Bagnell, J. A. (2025). All roads lead to likelihood: The value of reinforcement learning in fine-tuning. arXiv preprint arXiv:2503.01067.
>
> [5] Brockman, G., Cheung, V., Pettersson, L., Schneider, J., Schulman, J., Tang, J., & Zaremba, W. (2016). Openai gym. arXiv preprint arXiv:1606.01540.
>
> [6] Foster, D. J., Block, A., & Misra, D. (2024). Is behavior cloning all you need? understanding horizon in imitation learning. Advances in Neural Information Processing Systems, 37, 120602-120666.

---

> > ### Author Response · Authors · 2025-11-25
> >
> > We would like to kindly follow up on our previous response as the discussion deadline approaches. We have addressed your concerns as summarized below:
> >
> > - **Reward vs. policy class size:** In most practical settings, the reward class is significantly smaller than the policy class (e.g., in LLM post-training and MuJoCo, rewards are simple parameterizations compared to complex policies).
> >
> > - **No contradiction with Foster et al. (2024):** Purely offline learning corresponds to $K = 1$, not $K = 0$. When $N = \Theta(\sigma^2 \epsilon^{-2})$, the lower bound permits zero online interactions, fully consistent with Foster et al.
> >
> > - **Practical algorithm design:** We use a model ensemble to approximate the version space, perform off-policy updates, and apply optimistic model selection, which differs from the on-policy, non-optimistic updates in [1].
> >
> > - **Additional experiments:** Newly added comparisons with FILTER, HyPE, and HyPER show that MB-AIL is competitive in episode return and more sample-efficient.
> >
> > - **Robustness to misspecification:** We evaluate the impact of mild model misspecification in GridWorld setting, with graceful performance degradation as misspecification increases.
> >
> > - **Empirical validation of $\log|\Pi| > \log|\mathcal{R}|$:** Smaller reward networks preserve performance, while simplifying the policy network causes substantial degradation, supporting our assumption.
> >
> > - **Variance dependence confirmed:** Increasing expert policy variance consistently degrades performance, in alignment with our variance-dependent theoretical results.
> >
> > We would like to know if there is any other comments and suggestions for our work. Thank you again for your thoughtful feedback.

---

### Official Review · Reviewer_yf8s · 2025-11-04

**Soundness:** 3
**Presentation:** 3
**Contribution:** 4
**Rating:** 8
**Confidence:** 4

**Summary:**

This paper has proposed MB-AIL, a model-based adversarial imitation learning algorithm (Algorithm 1). Second-order regret/sample complexity upper bounds are provided in Section 5.1 (see Theorem 5.1 and Corollary 5.2), and minmax lower bounds are provided in Section 5.2 (Theorem 5.9). Preliminary experimental results are provided in the appendices.

**Strengths:**

- Overall, this is a very strong theoretical machine learning paper. To the best of my knowledge, the proposed MB-AIL algorithm is novel. This paper has established both upper regret/sample complexity bounds for MB-AIL and minmax lower bounds. They together justify that MB-AIL is near-optimal in both its use of online interaction and its dependence on expert demonstrations.

- This paper has done a very good job of literature review, as summarized in Table 1.

- Extensive and rigorous discussions on the theoretical results are provided in Section 5. Many of such discussions are thought-provoking.

Overall, I recommend accepting this paper.

**Weaknesses:**

- It is not clear to me why the authors do not include **any** experimental results in the main body of the paper. Of course, there is a page limit, but I think the authors can rewrite the paper to include at least one experimental result. I think this will make the paper more "balanced" between the theoretical results and the experimental results.

- My understanding is that the realizability assumption (Assumption 3.5) is a major weakness of this paper. In particular, with function approximation, this assumption rarely holds in practical problems. I fully understand that this is a standard assumption made in theoretical papers to ensure the analyses are tractable, and analyzing the proposed algorithm without this assumption can be very difficult. However, I recommend that the authors add some experimental results when Assumption 3.5 is (mildly) violated to justify that the proposed algorithm is robust to (mild) model misspecification. (It is not clear to me if such experimental results already exist in the appendices; if so, please explain.)

- In Theorem 5.1, the regret bound depends on three notions of complexities: the eluder dimension $d_E$, the bracketing number for the model class $\mathcal{N}_{\mathcal{P}}$, and the covering number for the reward class $\mathcal{N}_{\mathcal{R}}$. Please better explain why we need three different notions of complexities to establish this regret bound. The readers should be able to understand the intuitions without reading the proof.

- I recommend accepting this paper, but I do feel that this paper is too "dense" for a conference paper. I recommend that the authors further polish the writing to make it more readable.

- There are still minor typos in the paper. Please make another pass and fix them. Examples:
  - In the first equation on Page 4, $a_{h'}$ should be $a_t$
  - In equations 3.1 and 3.2, the expectation over the initial state is missing.

**Questions:**

Please try to address the weaknesses listed above.

---

> ### Author Response · Authors · 2025-11-19
>
> Thank you for your positive feedback and we appreciate your strong support for the completeness of our work. We address your concerns in the following response:
>
>
> **Q1.** Can the authors incorporate more empirical results in the main paper?
>
> **A1.** We thank the reviewer for this suggestion. The majority of the empirical results were presented in Appendix E and F originally. We will extend Section 6 in the cameara ready revision given additional page allowance.
>
> ---
>
> **Q2.** How is the model robust to model misspfication empirically when the realizability assumption does not holds? Can you justify it through experiments?
>
> **A2.** In fact, our GridWorld experiment already includes a study on misspecification error, as shown in Appendix E.1 of the original manuscript. Regarding the effect of the reward space size on performance, when the reward space shrinks as $n$ increases (left plot of Figure 2), the ground-truth reward function falls outside the reward class. This occurs because we partition the reward table into $n \times n$ regions and assign a single reward value within each region. Interestingly, the results in Figure 2 show that even under reward misspecification, the algorithm can achieve better performance as the reward class becomes smaller.
>
> Moreover, we conduct additional experiments to evaluate the impact of model misspecification. Specifically, we randomly sample $m$ states in GridWorld as misspecified states. During Bellman updates, whenever a transition leads to one of these misspecified states, we manually overwrite the next state as (0,0), thereby introducing controlled transition model misspecification. The results show that the algorithm maintains strong performance when the proportion of misspecified states is small, but its performance degrades as the degree of misspecification increases:
>
> | Misspecified Portion ($m/N$) | Episode Rewards   |
> |----------------------|-------------------|
> | 0/81                 | 7.85 ± 0.56       |
> | 1/81                 | 6.75 ± 0.55       |
> | 2/81                 | 6.75 ± 0.69       |
> | 3/81                 | 6.70 ± 1.88       |
> | 4/81                 | 5.30 ± 2.91       |
> | 5/81                 | 3.05 ± 3.20       |
>
> The results are reported using 5 random seeds.
>
> ---
> **Q3.** What is the intuition behind the eluder dimension $d_E$, the bracketing number for the model class $N_{P}$ and the covering number for the reward class $N_{R}$?
>
> **A3.** The eluder dimension $d_E$ characterizes the complexity of learning in a sequential decision-making setting, capturing how difficult it is to resolve uncertainty about the value or transition functions through interaction, as in [1]. The bracketing number for the model class $N_P$ and the covering number for the reward class $N_R$ quantify the richness of these function classes, as discussed in [2]. Especially, when these classes are finite, $N_P$ and $N_R$ can be seen as the cardinality of these classes separately.
>
> ---
>
> [1] Russo, D., & Van Roy, B. (2013). Eluder Dimension and the Sample Complexity of Optimistic Exploration. Neural Information Processing Systems.
>
> [2] Wang, Z., Zhou, D., Lui, J., & Sun, W. (2024). Model-based rl as a minimalist approach to horizon-free and second-order bounds. arXiv preprint arXiv:2408.08994.

---

### Author Response · Authors · 2025-11-19

We sincerely thank all reviewers for their constructive feedback. We appreciate the reviewers' recommendation for the thought-provoking discussions (Reviewer yf8s), well motivation and novelty (Reviewer 7CkW and J4QT) and the importance of this work (Reviewer J4QT and PeTQ).

Below we summarize the main revisions made in our updated manuscript (highlighted in **blue** in the revised version):

**1.** In lines 106–136 and 150–153, we incorporated additional discussions of related works, addressing the concerns raised by **Reviewer J4QT** and **Reviewer PeTQ**.

**2.** In lines 116–118, we clarified the distinction between the variance of the expert policy and the variance over all policies in Table 1, reducing potential ambiguity as noted by **Reviewer PeTQ**.

**3.** In **Remark 5.6**, we added further discussion on offline imitation learning results based on the $Q$-function class, addressing comments from **Reviewer PeTQ**.

**4.** In **Theorem 5.8**, **Remark 5.9**, and **Appendix B**, we clarified the lower bound in the offline setting in response to **Reviewer 7CkW** and **Reviewer PeTQ**.

**5.** In **Appendix B**, we expanded the discussion on variance dependence, explaining why our results do not contradict prior work that depends solely on expert-policy variance, as pointed out by **Reviewer PeTQ**.

**6.** In lines 1975–1981, we added new experimental results on model misspecification, addressing feedback from **Reviewer yf8s** and **Reviewer 7CkW**.

**7.** In **Appendix H**, we included additional ablation studies on network sizes for both policy and reward parameterizations, addressing suggestions from **Reviewer 7CkW**.

---

### Author Response · Authors · 2025-11-24

We sincerely thank all reviewers for their constructive feedback. Noticing that the 10-th page is allowed for revision and camera ready, we have submitted a second revision of the paper and, following Reviewer yf8s's suggestion, moved part of the experimental results from the appendix to the main text to improve balance and clarity in the presentation. All changes in the revised manuscript are highlighted in blue.

We would also like to follow up with all reviewers and ask if there are any additional comments or questions. We would be more than happy to discuss!

Best regards,
The Authors

---

### Meta-Review · Area_Chair_Uj9t · 2026-01-07

**Summary:**

reviewers generally agreed the theoretical contribution is strong, offering novel horizon-free second-order bounds and a near-optimal algorithm. primary concerns included clarity of the lower bound's relation to prior work, the theory-practice gap, limited experiments, and the strong realizability assumption. the authors addressed these in rebuttal by adding experiments, clarifying the lower bound, and discussing robustness.

**Reviewer Concerns:**

addressed: the lower bound contradiction (petq, 7ckw) was clarified: the exp(-n) factor reconciles it with offline bc. the practical insight question (j4qt) was answered with guidelines on when online interaction helps. requests for more experiments (yf8s, 7ckw) were met with additional comparisons and ablation studies.

outstanding: the theory-practice gap remains: the practical algorithm uses ensembles not version spaces. the realizability assumption is still strong, and while misspecification tests were added, more extensive evaluation would help.

**Reviewer Scores:**

yf8s: likely maintains 8, as their concerns were mostly addressed.

7ckw: might still hesitate due to the theory-practice gap.

j4qt: likely stays at 8.

petq: might increase from 2 to 5.

---

### Decision · Program_Chairs · 2026-01-26

Accept (Poster)